# Learning and Generalization in Univariate Overparameterized Normalizing Flows

## Abstract

In supervised learning, it is known that overparameterized neural networks with one hidden layer provably and efficiently learn and generalize, when trained using Stochastic Gradient Descent (SGD). In contrast, the benefit of overparameterization in unsupervised learning is not well understood. Normalizing flows (NFs) learn to map complex real-world distributions into simple base distributions and constitute an important class of models in unsupervised learning for sampling and density estimation. In this paper, we theoretically and empirically analyze these models when the underlying neural network is one hidden layer overparametrized network. On the one hand, we provide evidence that for a class of NFs, overparametrization hurts training. On the other hand, we prove that another class of NFs, with similar underlying networks, can efficiently learn any reasonable data distribution under minimal assumptions. We extend theoretical ideas on learning and generalization from overparameterized neural networks in supervised learning to overparameterized normalizing flows in unsupervised learning. We also provide experimental validation to support our theoretical analysis in practice.

## 1 Introduction

Neural network models trained using simple first-order iterative algorithms have been very effective in both supervised and unsupervised learning. Theoretical reasoning of this phenomenon requires one to consider simple but quintessential formulations, where this can be demonstrated by mathematical proof, along with experimental evidence for the underlying intuition. First, the minimization of training loss is typically a non-smooth and non-convex optimization over the parameters of neural networks, so it is surprising that neural networks can be trained efficiently by first-order iterative algorithms. Second, even large neural networks whose number parameters are more than the size of training data often generalize well with a small loss on the unseen test data, instead of overfitting the seen training data. Recent work in supervised learning attempts to provide theoretical justification for why overparameterized neural networks can train and generalize efficiently in the above sense.

In supervised learning, the empirical risk minimization with quadratic loss is a non-convex optimization problem even for a fully connected neural network with one hidden layer of neurons with ReLU activations. Around 2018, it was realized that when the hidden layer size is large compared to the dataset size or compared to some measure of complexity of the data, one can provably show efficient training and generalization for these networks, e.g. Jacot et al. (2018); Li & Liang (2018); Du et al. (2018); Allen-Zhu et al. (2019); Arora et al. (2019). Of these, Allen-Zhu et al. (2019) is directly relevant to our paper and will be discussed later.

The role of overparameterization, and provable training and generalization guarantees for neural networks are less well understood in unsupervised learning. Generative models or learning a data distribution from given samples is an important problem in unsupervised learning. Popular generative models based on neural networks include Generative Adversarial Networks (GANs) (e.g., Goodfellow et al. (2014)), Variational AutoEncoders (VAEs) (e.g., Kingma & Welling (2014)), and Normalizing Flows (e.g., Rezende & Mohamed (2015)). GANs and VAEs have shown impressive capability to generate samples of photo-realistic images but they cannot give probability density estimates for new data points. Training of GANs and VAEs has various additional challenges such as mode collapse, posterior collapse, vanishing gradients, training instability, etc. as shown in e.g. Bowman et al. (2016); Salimans et al. (2016); Arora et al. (2018); Lucic et al. (2018).

In contrast to the generative models such as GANs and VAEs, when normalizing flows learn distributions, they can do both sampling and density estimation, leading to wide-ranging applications as mentioned in the surveys by Kobyzev et al. (2020) and Papamakarios et al. (2019). Theoretical understanding of learning and generalization in normalizing flows (more generally, generative models and unsupervised learning) is a natural and important open question, and our main technical contribution is to extend known techniques from supervised learning to make progress towards answering this question. In this paper, we study learning and generalization in the case of univariate overparameterized normalizing flows. Restriction to the univariate case is technically non-trivial and interesting in its own right: univariate ReLU networks have been studied in recent supervised learning literature (e.g., Savarese et al. (2019), Williams et al. (2019), Sahs et al. (2020) and Daubechies et al. (2019)). Multidimensional flows are qualitatively more complex and our 1D analysis sheds some light on them (see Sec. 4). Before stating our contributions, we briefly introduce normalizing flows; details appear in Section 2.

**Normalizing Flows.** We work with one-dimensional probability distributions with continuous density. The general idea behind normalizing flows (NFs), restricted to 1D can be summarized as follows: Let $X \in \mathbb{R}$ be a random variable denoting the data distribution. We also fix a *base* distribution with associated random variable $Z$ which is typically standard Gaussian, though in this paper we will work with the exponential distribution as well. Given i.i.d. samples of $X$, the goal is to learn a continuous strictly monotone increasing map $f_X : \mathbb{R} \to \mathbb{R}$ that transports the distribution of $X$ to the distribution of $Z$: in other words, the distribution of $f_X^{-1}(Z)$ is that of $X$. The learning of $f_X$ is done by representing it by a neural network and setting up an appropriate loss function.

The monotonicity requirement on $f$ which makes $f$ invertible, while not essential, greatly simplifies the problem and is present in all the works we are aware of. It is not clear how to set up a tractable optimization problem without this requirement. Since the function represented by standard neural networks are not necessarily monotone, the design of the neural net is altered to make it monotone. For our 1D situation, one-hidden layer networks are of the form $N(x) = \sum_{i=1}^{m} a_i \sigma(w_i x + b_i)$, where $m$ is the size of the hidden layer and the $a_i, w_i, b_i$ are the parameters of the network.

We will assume that the activation functions used are monotone. Here we distinguish between two such alterations: (1) Changing the parametrization of the neural network. This can be done in multiple ways: instead of $a_i, w_i$ we use $a_i^2, w_i^2$ (or other functions, such as the exponential function, of $a_i, w_i$ that take on only positive values) (Huang et al., 2018; Cao et al., 2019). This approach appears to be the most popular. In this paper, we also suggest another related alteration: we simply restrict the parameters $a_i, w_i$ to be positive. This is achieved by enforcing this constraint during training. (2) Instead of using $N(x)$ for $f(x)$ we use $\phi(N(x))$ for $f'(x) = \frac{\mathrm{d}f}{\mathrm{d}x}$, where $\phi : \mathbb{R} \to \mathbb{R}^+$ takes on only positive values. Positivity of $f'$ implies monotonicity of $f$. Note that no restrictions on the parameters are required; however, because we parametrize $f'$, the function $f$ needs to be reconstructed using numerical quadrature. This approach is used by Wehenkel & Louppe (2019).

We will refer to the models in the first class as *constrained normalizing flows* (CNFs) and those in the second class as *unconstrained normalizing flows* (UNFs).

**Our Contributions.** In this paper, we study both constrained and unconstrained univariate NFs theoretically as well as empirically. The existing analyses for overparametrized neural networks in the supervised setting work with a linear approximation of the neural network, termed *pseudo network* in Allen-Zhu et al. (2019). They show that (1) there is a pseudo network with weights close to the initial ones approximating the target function, (2) the loss surfaces of the neural network and the pseudo network are close and moreover the latter is convex for convex loss functions. This allows for proof of the convergence of the training of neural network to global optima. One can try to adapt the approach of using a linear approximation of the neural network to analyze training of NFs. However, one immediately encounters some new roadblocks: the loss surface of the pseudo networks is non-convex in both CNFs and UNFs.

In both cases, we identify novel variations that make the optimization problem for associated pseudo network convex: For CNFs, instead of using $a_i^2, w_i^2$ as parameters, we simply impose the constraints $a_i \geq \epsilon$ and $w_i \geq \epsilon$ for some small constant $\epsilon$. The optimization algorithm now is projected SGD, which in this case incurs essentially no extra cost over SGD due to the simplicity of the positivity constraints. Apart from making the optimization problem convex, in experiments this variation

slightly *improves* the training of NFs compared to the reparametrization approaches, and may be useful in practical settings.

Similarly, for UNFs we identify two changes from the model of Wehenkel & Louppe (2019) that make the associated optimization problem convex, while still retaining empirical effectiveness: (1) Instead of Clenshaw–Curtis quadrature employed in Wehenkel & Louppe (2019) which uses positive and negative coefficients, we use the simple rectangle quadrature which uses only positive coefficients. This change makes the model somewhat slow (it uses twice as many samples and time to get similar performance on the examples we tried). (2) Instead of the standard Gaussian distribution as the base distribution, we use the exponential distribution. In experiments, this does not cause much change.

Our results point to a dichotomy between these two classes of NFs: our variant of UNFs can be theoretically analyzed when the networks are overparametrized to prove that the UNF indeed learns the data distribution. To our knowledge, this is the first "end-to-end" analysis of an NF model, and a neural generative model using gradient-based algorithms used in practice. This proof, while following the high-level scheme of Allen-Zhu et al. (2019) proof, has a number of differences, conceptual as well as technical, due to different settings. E.g., our loss function involves a function and its integral estimated by quadrature.

On the other hand, for CNFs, our empirical and theoretical findings provide evidence that over-parametrization makes training slower to the extent that models of similar size which learn the data distribution well for UNFs, fail to do so for CNFs. We also analyze CNFs theoretically in the overparametrized setting and point to potential sources of the difficulty. The case of moderate-sized networks, where training and generalization do take place empirically, is likely to be difficult to analyze theoretically as presently this setting is open for the simpler supervised learning case. We hope that our results will pave the way for further progress. We make some remarks on the multidimensional case in Sec. 4. In summary, our contributions include:

- To our knowledge, first efficient training and generalization proof for NFs (in 1D).
- Identification of architectural variants of UNFs that admit analysis via overparametrization.
- Identification of "barriers" to the analysis of CNFs.

**Related Work.** Previous work on normalizing flows has studied different variants such as planar and radial flows in Rezende & Mohamed (2015), Sylvester flow in van den Berg et al. (2018), Householder flow in Tomczak & Welling (2016), masked autoregressive flow in Papamakarios et al. (2017). Most variants of normalizing flows are specific to certain applications, and the expressive power (i.e., which base and data distributions they can map between) and complexity of normalizing flow models have been studied recently, e.g. Kong & Chaudhuri (2020) and Teshima et al. (2020). Invertible transformations defined by monotonic neural networks can be combined into autoregressive flows that are universal density approximators of continuous probability distributions; see Masked Autoregressive Flows (MAF) Papamakarios et al. (2017), UNMM-MAF by Wehenkel & Louppe (2019), Neural Autoregressive Flows (NAF) by Huang et al. (2018), Block Neural Autoregressive Flow (B-NAF) by Cao et al. (2019). Unconstrained Monotonic Neural Network (UMNN) models proposed by Wehenkel & Louppe (2019) are particularly relevant to the technical part of our paper.

Lei et al. (2020) show that when the generator is a two-layer tanh, sigmoid or leaky ReLU network, Wasserstein GAN trained with stochastic gradient descent-ascent converges to a global solution with polynomial time and sample complexity. Using the moments method and a learning algorithm motivated by tensor decomposition, Li & Dou (2020) show that GANs can efficiently learn a large class of distributions including those generated by two-layer networks. Nguyen et al. (2019b) show that two-layer autoencoders with ReLU or threshold activations can be trained with normalized gradient descent over the reconstruction loss to provably learn the parameters of any generative bilinear model (e.g., mixture of Gaussians, sparse coding model). Nguyen et al. (2019a) extend the work of Du et al. (2018) on supervised learning mentioned earlier to study weakly-trained (i.e., only encoder is trained) and jointly-trained (i.e., both encoder and decoder are trained) two-layer autoencoders, and show joint training requires less overparameterization and converges to a global optimum. The effect of overparameterization in unsupervised learning has also been of recent interest. Buhai et al. (2020) do an empirical study to show that across a variety of latent variable models and training algorithms, overparameterization can significantly increase the number of recovered ground truth latent variables. Radhakrishnan et al. (2020) show that overparameterized autoencoders

and sequence encoders essentially implement associative memory by storing training samples as attractors in a dynamical system.

**Outline.** A brief outline of our paper is as follows. Section 2 contains preliminaries and an overview of our results about constrained and unconstrained normalizing flows. Appendix B shows the existence of a pseudo network whose loss closely approximates the loss of the target function. Appendix C shows the coupling or closeness of their gradients over random initialization. Appendices D and E contain complete proofs of our optimization and generalization results, respectively. Section 3 and Appendix G contain our empirical studies towards validating our theoretical results.

## 2 PRELIMINARIES AND OVERVIEW OF RESULTS

We confine our discussion to the 1D case which is the focus of the present paper. The goal of NF is to learn a probability distribution given via i.i.d. samples data. We will work with distributions whose densities have finite support, and assumed to be $[-1, 1]$, without loss of generality. Let $X$ be the random variable corresponding to the data distribution we want to learn. We denote the probability density (we often just say density) of $X$ at $u \in \mathbb{R}$ by $p_X(u)$. Let $Z$ be a random variable with either standard Gaussian or the exponential distribution with $\lambda = 1$ (which we call standard exponential). Recall that the density of the standard exponential distribution at $u \in \mathbb{R}$ is given by $e^{-u}$ for $u \geq 0$ and 0 for $u < 0$.

Let $f : \mathbb{R} \to \mathbb{R}$ be a strictly increasing continuous function. Thus, $f$ is invertible. We use $f'(x) = \frac{\mathrm{d}f}{\mathrm{d}x}$ to denote the derivative. Let $p_{f,Z}(\cdot)$ be the density of the random variable $f^{-1}(Z)$. Let $x = f^{-1}(z)$, for $z \in \mathbb{R}$. Then by the standard change of density formula using the monotonicity of $f$ gives

$$p_{f,Z}(x) = p_Z(z)f'(x). \tag{2.1}$$

We would like to choose $f$ so that $p_{f,Z} = p_X$, the true data density. It is known that such an $f$ always exists and is unique; see e.g. Chapter 2 of Santambrogio (2015). We will refer to the distribution of $Z$ as the *base* distribution. Note that if we can find $f$, then we can generate samples of $X$ using $f^{-1}(Z)$ since generating the samples of $Z$ is easy. Similarly, we can evaluate $p_X(x) = p_Z(f^{-1}(z))f'(x)$ using (2.1). To find $f$ from the data, we set up the maximum log-likelihood objective:

$$\max_f \frac{1}{n} \sum_{i=1}^n \log p_{f,Z}(x_i) = \max_f \frac{1}{n} \left[ \sum_{i=1}^n \log p_Z(f(x_i)) + \sum_{i=1}^n \log f'(x_i) \right], \tag{2.2}$$

where $S = \{x_1, \ldots, x_n\} \subset \mathbb{R}$ contains i.i.d. samples of $X$, and the maximum is over continuous strictly increasing functions. When $Z$ is standard exponential, the optimization problem (2.2) becomes

$$\min_f L(f, S), \quad \text{where } L(f, S) = \frac{1}{n} \sum_{x \in S} L(f, x) \text{ and } L(f, x) = f(x) - \log f'(x). \tag{2.3}$$

A similar expression, with $f(x)^2/2$ replacing $f(x)$, holds for the standard Gaussian. We denote the loss for standard Gaussian as $L_G(f, x)$.

Informally, one would expect that as $n \to \infty$, for the optimum $f$ in the above optimization problems $p_{f,Z} \to p_X$. To make the above optimization problem tractable, instead of $f$ we use a neural network $N$. We consider one-hidden layer neural networks with the following basic form which will then be modified according to whether we are constraining the parameters or the output.

$$N(x) = \sum_{r=1}^m a_{r0} \, \rho \left( (w_{r0} + w_r) \, x + (b_r + b_{r0}) \right). \tag{2.4}$$

Here $m$ is the size of the hidden layer, $\rho : \mathbb{R} \to \mathbb{R}$ is a monotonically increasing activation function, the weights $a_{r0}, w_{r0}, b_{r0}$ are the initial weights chosen at random according to some distribution, and $w_r, b_r$ are offsets from the initial weights. We will only train the $w_r, b_r$ and the $a_{r0}$ will remain frozen to their initial values.

Let $\theta = (W, B) \in \mathbb{R}^{2m}$ denote the parameters $W = (w_1, w_2, ..., w_m) \in \mathbb{R}^m$ and $B = (b_1, b_2, ..., b_m) \in \mathbb{R}^m$ of the neural network. We use Stochastic Gradient Descent (SGD) to update the parameters of neural networks. Denote by $\theta^t = (W^t, B^t)$ with $W^t = (w_1^t, w_2^t, ..., w_m^t)$ and

$B^t = (b_1^t, b_2^t, ..., b_m^t)$ the parameters at time step $t = 1, 2, \ldots$, and the corresponding network by $N_t(x)$. The SGD updates are given by $\theta^{t+1} = \theta^t - \eta \nabla_\theta L_s(N_t, x^t)$ where $\eta > 0$ is learning rate, and $L_s(N_t, x^t)$ is a loss function, and $x^t \in S$ is chosen uniformly randomly at each time step. For supervised learning where we are given labeled data $\{(x_1, y_1), \ldots, (x_n, y_n)\}$, one often works with the mean square loss $L_s(N_t) = \frac{1}{n} \sum_{i=1}^n L_s(N_t, x_i)$ with $L_s(N_t, x_i) = (N_t(x_i) - y_i)^2$.

We now very briefly outline the proof technique of Allen-Zhu et al. (2019) for analyzing training and generalization for one-hidden layer neural networks for supervised learning. (While they work in a general agnostic learning setting, for simplicity, we restrict the discussion to the realizable setting.) In their setting, the data $x \in \mathbb{R}^d$ is generated by some distribution $D$ and the labels $y = h(x)$ are generated by some unknown function $h : \mathbb{R}^d \to \mathbb{R}$. The function $h$ is assumed to have small "complexity" $C_h$ which in this case measures the required size of neural network with smooth activations to approximate $h$.

The problem of optimizing the square loss is non-convex even for one-hidden layer networks. Allen-Zhu et al. (2019) instead work with *pseudo network*, $P(x)$ which is the linear approximation of $N(x)$ given by the first-order Taylor expansion of the activation:

$$P(x) = \sum_{r=1}^m a_{r0} \left( \sigma(w_{r0}x + b_{r0}) + \sigma'(w_{r0}x + b_{r0})(w_r x + b_r) \right). \qquad (2.5)$$

Similarly to $N_t$ we can also define $P_t$ with parameters $\theta^t$. They observe that when the network is highly overparameterized, i.e. the network size $m$ is sufficiently large compared to $C_h$, and the learning rate is small, i.e. $\eta = O(1/m)$, SGD iterates when applied to $L(N_t)$ and $L(P_t)$ remain close throughout. Moreover, the problem of optimizing $L(P)$ is a convex problem in $\theta$ and thus can be analyzed with existing methods. They also show an *approximation* theorem stating that with high probability there are neural network parameters $\theta^*$ close to the initial parameters $\theta^0$ such that the pseudo network with parameters $\theta^*$ is close to the target function. This together with the analysis of SGD shows that the pseudo network, and hence the neural network too, achieves small training loss. Then by a Rademacher complexity argument they show that the neural network after $T = O(C_h/\epsilon^2)$ time steps has population loss within $\epsilon$ of the optimal loss, thus obtaining a generalization result.

We will now describe how to obtain neural networks representing monotonically increasing functions using the two different methods mentioned earlier, namely CNFs and UNFs.

## 2.1 Constrained Normalizing Flow

Note that if we have $a_{r0} \geq 0$, $w_{r0} + w_r \geq 0$ for all $r$, then the function represented by the neural network is monotonically increasing. We can ensure this positivity constraint by replacing $a_{r0}$ and $w_{r0} + w_r$ by their functions that take on only positive values. For example, the function $x \mapsto x^2$ would give us the neural network $N(x) = \sum_{r=1}^m a_{r0}^2 \rho((w_{r0} + w_r)^2 x + b_{r0} + b_r)$. Note that $a_{r0}$, $w_{r0} + w_r$ and $b_{r0} + b_r$ have no constraints, and so this network can be trained using standard gradient-based algorithms. But first we need to specify the (monotone) activation $\rho$. Let $\sigma(x) = x \, \mathbb{I}[x \geq 0]$ denote the ReLU activation. If we choose $\rho = \sigma$, then note that in (2.3) we have

$$\log f'(x) = \log \frac{\partial N(x)}{\partial x} = \log \left( \sum_{r=1}^m a_{r0}^2 (w_{r0} + w_r)^2 \mathbb{I}\left[ (w_{r0} + w_r)^2 x + b_{r0} + b_r \geq 0 \right] \right).$$

This is a discontinuous function in $x$ as well as in $w_r$ and $b_r$. Gradient-based optimization algorithms are not applicable to problems with discontinuous objectives, and indeed this is reflected in experimental failure of such models in learning the distribution. By the same argument, any activation that has a discontinuous derivative is not admissible. Activations which have continuous derivative but are convex (e.g. $\mathsf{ELU}(x)$ given by $e^x - 1$ for $x < 0$ and $x$ for $x \geq 0$)) also cannot be used because then $N(x)$ is also a convex function of $x$, which need not be the case for the optimal $f$. The oft-used activation $\tanh$ does not suffer from either of these defects. Pseudo network with activation $\tanh$ is given by

$$P(x) = \sum_{r=1}^m a_{r0}^2 \left( \tanh(w_{r0}^2 x + b_{r0}) + \tanh'(w_{r0}^2 x + b_{r0}) \left( (w_r^2 + 2w_{r0}w_r) x + b_r \right) \right).$$

Note that $P(x)$ is not linear in the parameters $\theta$. Hence, it is not obvious that the loss function for the pseudo network will remain convex in parameters; indeed, non-convexity can be confirmed in experiments. A similar situation arises for exponential parameterization instead of square.

To overcome the non-convexity issue, we propose another formulation for constrained normalizing flows. Here we retain the form of the neural network as in (2.4), but ensure the constraints $a_{r0} \geq 0$ and $w_{r0} \geq 0$ by the choice of the initialization distribution and $w_{r0} + w_r \geq 0$ by using projected gradient descent for optimization.

$$N(x) = \sum_{r=1}^{m} a_{r0} \, \tanh\left((w_{r0} + w_r)\, x + (b_r + b_{r0})\right), \text{ with constraints } w_{r0} + w_r \geq \epsilon, \text{ for all } r.$$

Here, $\epsilon > 0$ is a small constant ensuring strict monotonicity of $N(x)$. Note that constraints in the formulation are simple and easy to use in practice. The pseudo network in this formulation will be

$$P(x) = \sum_{r=1}^{m} a_{r0} \left(\tanh(w_{r0}x + b_{r0}) + \tanh'(w_{r0}x + b_{r0})\, (w_r x + b_r)\right),$$

with constraints $w_{r0} + w_r \geq \epsilon$, for all $r$. $P(x)$ is linear in $\theta$, therefore the objective function is also convex in $\theta$. Note that $P(x)$ need not be forced to remain monotone using constraints: if $N(x)$ and $P(x)$ are sufficiently close and $N(x)$ is strictly monotone with not too small $\min_x \frac{\partial N(x)}{\partial x}$, then we will get monotonicity of $P(x)$. Next, we point out that this formulation has a problem in approximation of any target function by a pseudo network. We decompose $P(x)$ into two parts: $P(x) = P_c(x) + P_\ell(x)$, where

$$P_c(x) = \sum_{r=1}^{m} a_{r0} \left(\tanh(w_{r0}x + b_{r0})\right) \quad \text{and} \quad P_\ell(x) = \sum_{r=1}^{m} a_{r0} \left(\tanh'(w_{r0}x + b_{r0})\, (w_r x + b_r)\right).$$

Note that $P_c(x)$ only depends upon initialization and does not depend on $w_r$ and $b_r$. Hence, it can not approximate the target function after the training, therefore $P_\ell(x)$ needs to approximate target function with $P_c(x)$ subtracted. Now, we will show that $P_\ell(x)$ can not approximate "sufficiently non-linear" functions. The initialization distribution for $w_{r0}$ is half-normal distribution with zero-mean and variance$=\frac{1}{m}$ of normal distribution, i.e. $w_{r0} = |X|$ where $X$ has normal distribution with the same parameters. The bias term $b_{r0}$ follows normal distribution with 0 mean and $\frac{1}{m}$ variance. Using the initialization, we can say that $w_{r0}$ and $|b_{r0}|$ are $O\left(\frac{\sqrt{\log m}}{\sqrt{m}}\right)$ with high probability; therefore, $|w_{r0}x + b_{r0}|$ is $O\left(\frac{\sqrt{\log m}}{\sqrt{m}}\right)$. Using the fact that $\tanh'(y) \approx 1$ for small $y$, we get that $\tanh'(w_{r0}x + b_{r0}) \approx 1$ for sufficient large $m$. In such cases, $P_\ell(x)$ becomes linear function in $x$ and won't be able to approximate sufficiently non-linear function.

Note that this issue does not arise in pseudo network with ReLU activation because the derivative of ReLU is discontinuous at 0 but as described earlier, for CNFs activations need to have continuous derivative. The same issue in approximation arises for all activations with continuous derivative. Using other variance of initializations leads to problem in other parts of the proof. This problem remains if we use normal distribution initialization of $w_{r0}$ and $b_{r0}$ with variance $o\left(\frac{1}{\log m}\right)$. For normal distribution initialization of $w_{r0}$ and $b_{r0}$ with variance $\Omega\left(\frac{1}{\log m}\right)$ and $O(1)$, successfully training of CNFs to small training error can lose coupling between neural network $N(x)$ and pseudo network $P(x)$. Please see Appendix F for more details. A generalization argument for activations with continuous derivatives is not known even in the supervised case, therefore we do not work with constrained normalizing flow. However, we show the effect of overparameterization for constrained normalizing flow with $\tanh$ activation in experiments (Section 3).

## 2.2 UNCONSTRAINED NORMALIZING FLOW

Unlike the constrained case, where we modeled $f(x)$ using a neural network $N(x)$, here we model $f'(x)$ using a neural network. Then we have $f(x) = \int_{-1}^{x} f'(u)\, du$. While this cannot be computed exactly, good approximation can be obtained via numerical integration also known as numerical quadrature of $f'(x)$. The strict monotonicity of $f$ is achieved by ensuring that $f'(x)$ is always

positive. To this end a suitable nonlinearity is applied on top of the neural network: $f'(x) = \phi(N(x))$, where $N(x)$ is as in (2.4) with $\rho = \sigma = \text{ReLU}$, and $\phi$ is the function $\text{ELU} + 1$ given by $\phi(x) = e^x \, \mathbb{I}[x < 0] + (x + 1) \, \mathbb{I}[x \geq 0]$. Thus $\phi(x) > 0$, for all $x \in \mathbb{R}$, which means that $f'(x) > 0$ for all $x$. Although this was the only property of $\text{ELU} + 1$ mentioned by Wehenkel & Louppe (2019), it turns out to have several other properties which we will exploit in our proof: it is 1-Lipschitz monotone increasing; its derivative is bounded from above by 1.

We denote by $\tilde{f}(x)$ the estimate of $f(x) = \int_{-1}^{x} f'(u) \, du$ obtained from $f'(x)$ via quadrature $\tilde{f}(x) = \sum_{i=1}^{Q} q_i f'(\tau_i(x))$. Here $Q$ is the number of quadrature points $\tau_1(x), \ldots, \tau_Q(x)$, and the $q_1, \ldots, q_Q \in \mathbb{R}$ are the corresponding coefficients. Wehenkel & Louppe (2019) use Clenshaw–Curtis quadrature where the coefficients $q_i$ can be negative.

We will use simple rectangle quadrature, which arises in Riemann integration, and uses only positive coefficients: $\tilde{f}(x) = \Delta_x \big[ f'(-1 + \Delta_x) + f'(-1 + 2\Delta_x) \ldots + f'(x) \big]$, where $\Delta_x = \frac{x+1}{Q}$. It is known (see e.g. Chapter 5 in Atkinson (1989) for related results) that

$$\left| \tilde{f}(x) - f(x) \right| \leq \frac{M''(x+1)^2}{2Q}, \quad \text{where } M'' = \max_{u \in [-1, x]} |f''(u)|.$$

Compared to Clenshaw–Curtis quadrature, the rectangle quadrature requires more points for similar accuracy (in our experiments this was about double). However, we use it because all the coefficients are positive which helps make the problem of minimizing the loss a convex optimization problem.

Instead of using $f$, to which we do not have access, we use $\tilde{f}$ in the loss function, denoting it $\hat{L}(f', x)$ for the standard exponential as the base distribution to write $\hat{L}(f', x) = \tilde{f}(x) - \log f'(x)$ and $\hat{L}(f', S) = \frac{1}{n} \sum_{x \in S} \hat{L}(f', x)$. The loss $\hat{L}_G(f', x)$ for the standard Gaussian as the base distribution is defined similarly.

Let $X$ be a random variable with density supported on $[-1, 1]$. Let the base distribution be the standard exponential, and so $Z$ will be a random variable with the standard exponential distribution. And let $F^* : \mathbb{R} \to \mathbb{R}$ be continuous monotone increasing such that $F^{*-1}(Z)$ has the same distribution as $X$. Let $S = \{x_1, \ldots, x_n\}$ be a set of i.i.d. samples of $X$. Following Allen-Zhu et al. (2019), we initialize $a_{r0} \sim \mathcal{N}(0, \epsilon_a^2)$, $w_{r0} \sim \mathcal{N}\left(0, \frac{1}{m}\right)$ and $b_{r0} \sim \mathcal{N}\left(0, \frac{1}{m}\right)$, where $\epsilon_a > 0$ is a small constant to be set later. The SGD updates are given by $\theta^{t+1} = \theta^t - \eta \nabla_\theta \hat{L}(f_t', x^t)$ where $f_t'(x) = \phi(N_t(x))$, and $x^t \in S$ is chosen uniformly at random at each step. We can now state our main result.

**Theorem 2.1** (informal statement of Theorem E.1). *(loss function is close to optimal) For any $\epsilon > 0$ and for any target function $F^*$ with finite second order derivative, hidden layer size $m \geq \frac{C_1(F^{*\prime})}{\epsilon^2}$, the number of samples $n \geq \frac{C_2(F^{*\prime})}{\epsilon^2}$ and the number of quadrature points $Q \geq \frac{C_3(F^{*\prime})}{\epsilon}$, where $C_1(\cdot), C_2(\cdot), C_3(\cdot)$ are complexity measures, with probability at least $0.9$, we have*

$$\mathbb{E}_{\text{sgd}} \left[ \frac{1}{T} \sum_{t=0}^{T-1} \mathbb{E}_{x \sim \mathcal{D}} L(f_t, x) \right] - \mathbb{E}_{x \sim \mathcal{D}} \left[ L(F^*, x) \right] = O(\epsilon).$$

The complexity functions in the above statement have natural interpretations in terms of how fast the function oscillates. Now recall that $\text{KL}(p_{F^*, Z} || p_{f_t, Z}) = \mathbb{E}_X \log \frac{p_{F^*, Z}(X)}{p_{f_t, Z}(X)}$, which gives $\mathbb{E}_{\text{sgd}} \left[ \frac{1}{T} \sum_{t=0}^{T-1} \text{KL}(p_{F^*, Z} || p_{f_t, Z}) \right] = O(\epsilon)$. Recall that $p_{f, Z}(x)$ is the probability density of $f^{-1}(Z)$. Using Pinsker's inequality, we can also bound the total variation distance between the learned and data distributions $p_{f_t, Z}$ and $p_{F^*, Z}$.

Define pseudo network $g'(x)$, which acts as proxy for $f'(x)$, as $g'(x) = \phi(P(x))$. Note that our definition of pseudo network is not the most straightforward version: $g'(x)$ is not a linear approximation of $f'(x)$. As in Allen-Zhu et al. (2019), we begin by showing the existence of a pseudo network close to the target function. However, for this we cannot use the approximation lemma in Allen-Zhu et al. (2019) as it seems to require dimension at least 2. We use the recent result of Ji et al. (2020) instead (Lemma B.1). The presence of both $f'$ and $\tilde{f}$ and other differences in the loss function leads to new difficulties in the analysis compared to the supervised case. We refer to the full proof due to the lack of space.

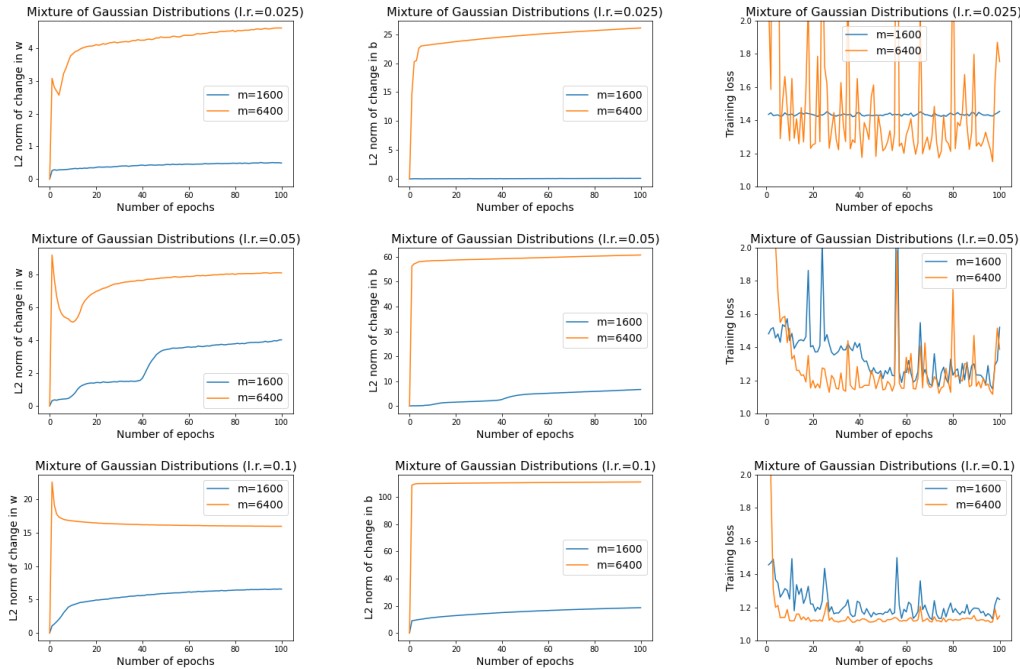

Figure 1: Effect of over-parameterization on training of constrained normalizing flow on mixture of Gaussian dataset for number of hidden layers $m = 1600, 6400$

## 3 EXPERIMENTS

Full details of experimental setup and additional results on constrained normalizing flow as well as results on unconstrained normalizing flow are given in appendix G.

### 3.1 RESULTS FOR CONSTRAINED NORMALIZING FLOW

In Sec. 2.1, we suggested that high overparameterization may adversely affect training for constrained normalizing flows. We now give experimental evidence for this. In Figs. 1, we see that as we increase the learning rate, training becomes more stable for larger $m$. Note that for learning rate 0.025, constrained normalizing flow with $m = 1600$ doesn't learn anything due to small learning rate. We observe that the $L_2$-norms of $W^t$ and $B^t$ for $m = 6400$ are at least as large as those of $m = 1600$. On both datasets, as we increase the learning rate, $L_2$-norm of $B^t$ increases and learning of constrained normalizing flow becomes more stable. These observations support our claim in Sec.2.1 that for learning and approximation of overparameterized constrained normalizing flow, neural networks need large $L_2$-norms of $W^t$ and $B^t$.

## 4 CONCLUSION

In this paper, we gave the first theoretical analysis of normalizing flows in the simple but instructive univariate case. We gave empirical and theoretical evidence that overparametrized networks are unlikely to be useful for CNFs. By contrast, for UNFs, overparametrization does not hurt and we can adapt techniques from supervised learning to analyze two-layer (or one hidden layer) networks. Our technical adaptations and NF variants may find use in future work.

Our work raises a number of open problems: (1) We made two changes to the unconstrained flow architecture of Wehenkel & Louppe (2019). An obvious open problem is an analysis of the original architecture or with at most one change. While the exponential distribution works well as the base distribution, can we also analyze the Gaussian distribution? Similarly, Clenshaw-Curtis quadrature instead of simple rectangle quadrature? These problems seem tractable but also likely

to require interesting new techniques as the optimization becomes non-convex. That would get us one step closer to the architectures used in practice. (2) Analysis of constrained normalizing flows. It is likely to be difficult because, as our results suggest, one needs networks that are not highly overparametrized—this regime is not well-understood even in the supervised case. (3) Finally, analysis of normalizing flows for the multidimensional case. Our 1D result brings into focus potential difficulties: All unconstrained architectures seem to require more than one hidden layer, which poses difficult challenges even in the supervised case. For CNFs, it is possible to design an architecture with one hidden layer, but as we have seen in our analysis of CNFs, that is challenging too.

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

## A  Notations

We denote $(\boldsymbol{\alpha}, \boldsymbol{\beta})$ as a concatenation of 2 vectors $\boldsymbol{\alpha}$ and $\boldsymbol{\beta}$. For any 2 vectors $\boldsymbol{\alpha}$ and $\boldsymbol{\beta}$, $\boldsymbol{\alpha} \odot \boldsymbol{\beta}$ denotes element wise multiplication of $\boldsymbol{\alpha}$ and $\boldsymbol{\beta}$ vector. We denote the parameters of neural network $\theta \in \mathbb{R}^{2m}$ is concatenation of $W = (w_1, w_2, ..., w_m) \in \mathbb{R}^m$ and $B = (b_1, b_2, ..., b_m) \in \mathbb{R}^m$ (i.e. $\theta = (W, B)$). Similarly, $\theta^t = (W^t, B^t)$ where $W^t = (w_1^t, w_2^t, ..., w_m^t)$ and $B^t = (b_1^t, b_2^t, ..., b_m^t)$. Similarly, $A_0 = (a_{10}, a_{20}, \ldots, a_{r0}, \ldots, a_{m0})$. We denote $\mathbf{1} = (1, 1, \ldots, 1) \in \mathbb{R}^m$. We use Big-$O$ notation to hide constants. We use $\log$ to denote natural logarithm. $[n]$ denotes set $\{1, 2, \ldots, n\}$

## B  Existence

This section contains a proof that shows existence of a pseudo network whose loss closely approximates the loss of the target function.

**Lemma B.1.** *For every positive function $F^{*\prime}$, for every $x$ in the radius of 1 (i.e. $|x| \leq 1$), there exist a function $h(w_{r0}, b_{r0}) : \mathbb{R}^2 \to [-U_h, U_h]$ such that*

$$\left| \phi^{-1}\left(F^{*\prime}(x)\right) - \mathbb{E}_{w_{r0}, b_{r0} \sim \mathcal{N}(0,1)}\left[h(w_{r0}, b_{r0})\mathbb{I}\left[w_{r0}x + b_{r0} \geq 0\right]\right] \right| \leq \omega_{\phi^{-1}(F^{*\prime})}(\delta)$$

*where $U_h$ is given by*

$$U_h = \tilde{O}\left( \frac{\| \left(\phi^{-1}\left(F^{*\prime}\right)\right)_{|\delta} \|_{L_1}^5}{\delta^{10}(\omega_{\phi^{-1}(F^{*\prime})}(\delta))^4} \right) \tag{B.1}$$

*Proof.* We use a result from Ji et al. (2020) to prove the lemma.

**Result B.1.** *(One-dimensional version of Theorem 4.3 from Ji et al. (2020)) Let $\psi : \mathbb{R} \to \mathbb{R}$ and $\delta > 0$ be given, and define*

$$\omega_\psi(\delta) = \sup\{\psi(x) - \psi(x') : \max\{|x|, |x'|\} \leq 1 + \delta, |x - x'| \leq \delta\}$$
$$\psi_{|\delta}(x) := \psi(x)\mathbb{I}\left[|x| \leq 1 + \delta\right]$$
$$\psi_{|\delta, \alpha} := \psi_{|\delta} * G_\alpha$$
$$\alpha := \frac{\delta}{1 + \sqrt{2\log(2M/\omega_\psi(\delta))}} = \tilde{O}(\delta)$$
$$M := \sup_{|x| \leq 1 + \delta} |\psi(x)|$$
$$\beta := \frac{1}{2\pi\alpha^2}$$

$$\mathcal{T}_r(w_{r0}, b_{r0}) := 2\left[ \psi_{|\delta, \alpha}(0) + \int \left| \hat{\psi}_{|\delta, \alpha}(v) \right| \cos\left(2\pi\left(\theta_{\psi_{|\delta, \alpha}}(v) - \|v\|\right)\right) dv \right]$$
$$+ 2\pi\left(2\pi\beta^2\right) \left| \hat{\psi}_{|\delta}(\beta w_{r0}) \right| e^{\frac{(b_{r0})^2}{2}} \sin\left(2\pi\left(\theta_{\psi_{|\delta, \alpha}}(\beta w_{r0}) - b_{r0}\right)\right) \mathbb{I}\left[|b_{r0}| \leq \|w_{r0}\| \leq r\right]$$

*where $*$ denotes convolution operation, $G_\alpha$ denotes Gaussian with mean 0 and variance $\alpha^2$. Note that $\tilde{O}$ hides logarithmic dependency of complexity measure of function $\psi$. $\left| \hat{\psi}_{|\delta, \alpha} \right|$ denotes magnitude of fourier transform of $\psi_{|\delta, \alpha}$ and $\theta_{\psi_{|\delta, \alpha}}$ denotes phase of fourier transform. Then,*

$$\sup_{|x| \leq 1} \left| \psi(x) - \mathbb{E}_{w_{r0}, b_{r0} \sim \mathcal{N}(0,1)}\left[\mathcal{T}_r(w_{r0}, b_{r0})\mathbb{I}\left[w_{r0}x + b_{r0} \geq 0\right]\right] \right| \leq \omega_\psi(\delta) \tag{B.2}$$

*The upper bound of $\mathcal{T}_r(w_{r0}, b_{r0})$ is given by*

$$\sup_{w_{r0}, b_{r0}} \|\mathcal{T}_r(w_{r0}, b_{r0})\| = \tilde{O}\left( \frac{\|\psi_{|\delta}\|_{L_1}^5}{\delta^{10}(\omega_\psi(\delta))^4} \right) = U_\mathcal{T} \tag{B.3}$$

Using Result B.1 for $\phi^{-1}(F^{*\prime}(x))$ function, denoting $\mathcal{T}_r(w_{r0}, b_{r0})$ for $\phi^{-1}(F^{*\prime}(x))$ function as $h(w_{r0}, b_{r0})$, we get

$$\left| \phi^{-1}(F^{*\prime}(x)) - \mathbb{E}_{w_{r0}, b_{r0} \sim \mathcal{N}(0,1)}\left[h(w_{r0}, b_{r0})\mathbb{I}\left[w_{r0}x + b_{r0} \geq 0\right]\right] \right| \leq \omega_{\phi^{-1}(F^{*\prime})}(\delta)$$

with following upper bound on $h(w_{r0}, b_{r0})$.

$$\sup_{w_{r0}, b_{r0}} h(w_{r0}, b_{r0}) \leq \tilde{O}\left( \frac{\| \left( \phi^{-1}\left( F^{*\prime} \right) \right)_{|\delta} \|_{L_1}^5}{\delta^{10}(\omega_{\phi^{-1}(F^{*\prime})}(\delta))^4} \right) = U_h$$

$\square$

Divide pseudo network $P(x)$ into 2 parts: $P_c(x)$, first part of pseudo network is constant and time-independent and $P_\ell(x)$, second part of pseudo network is linear in $w_r$ and $b_r$

$$P(x) = P_c(x) + P_\ell(x)$$

where

$$P_c(x) = \sum_{r=1}^{m} a_{r0} \left( w_{r0}x + b_{r0} \right) \mathbb{I}\left[ w_{r0}x + b_{r0} \geq 0 \right]$$

$$P_\ell(x) = \sum_{r=1}^{m} a_{r0} \left( w_r x + b_r \right) \mathbb{I}\left[ w_{r0}x + b_{r0} \geq 0 \right]$$

**Lemma B.2.** *(Approximating target function using $P_\ell(x)$) For every positive function $F^{*\prime}$ and for every $\epsilon \in (0, 1)$, with at least $1 - \frac{1}{c_1} - \exp\left( -\frac{\epsilon^2 m}{128 c_1^2 U_h^2 \log m} \right)$ probability over random initialization, there exist $\theta^*$ such that we get following inequality for all $x \in [-1, 1]$ and some fixed positive constant $c_1 > 1$.*

$$|\phi(P_\ell^*(x)) - F^{*\prime}(x)| \leq \omega_{\phi^{-1}(F^{*\prime})}(\delta) + \epsilon$$

*and upper bound $L_\infty$ norm of parameters is given by*

$$\|\theta^*\|_\infty \leq \frac{U_h \sqrt{\pi}}{\sqrt{2} m \epsilon_a}$$

*Proof.* Define $w_r^*$ and $b_r^*$ as

$$w_r^* = 0$$
$$b_r^* = \frac{\text{sign}(a_{r0}) \sqrt{\pi}}{m \epsilon_a \sqrt{2}} h(\sqrt{m} w_{r0}, \sqrt{m} b_{r0}) \tag{B.4}$$

Using $w_r^*$ and $b_r^*$,

$$\mathbb{E}_{a_{r0} \sim \mathcal{N}(0, \epsilon_a^2), w_{r0} \sim \mathcal{N}\left(0, \frac{1}{m}\right), b_{r0} \sim \mathcal{N}\left(0, \frac{1}{m}\right)} \left[ P_\ell^*(x) \right]$$

$$= \mathbb{E}_{a_{r0} \sim \mathcal{N}(0, \epsilon_a^2), w_{r0} \sim \mathcal{N}\left(0, \frac{1}{m}\right), b_{r0} \sim \mathcal{N}\left(0, \frac{1}{m}\right)} \left[ \sum_{r=1}^{m} a_{r0}(w_r^* x + b_r^*) \mathbb{I}\left[ w_{r0}x + b_{r0} \geq 0 \right] \right]$$

$$= \mathbb{E}_{a_{r0} \sim \mathcal{N}(0, \epsilon_a^2), w_{r0} \sim \mathcal{N}\left(0, \frac{1}{m}\right), b_{r0} \sim \mathcal{N}\left(0, \frac{1}{m}\right)} \left[ \frac{a_{r0}\text{sign}(a_{r0}) \sqrt{\pi}}{\epsilon_a \sqrt{2}} h(\sqrt{m} w_{r0}, \sqrt{m} b_{r0}) \mathbb{I}\left[ w_{r0}x + b_{r0} \geq 0 \right] \right]$$

$$\overset{(i)}{=} \mathbb{E}_{w_{r0} \sim \mathcal{N}\left(0, \frac{1}{m}\right), b_{r0} \sim \mathcal{N}\left(0, \frac{1}{m}\right)} \left[ h(\sqrt{m} w_{r0}, \sqrt{m} b_{r0}) \mathbb{I}\left[ \sqrt{m}\left( w_{r0}x + b_{r0} \right) \geq 0 \right] \right]$$

where equality (i) follows from Fact H.2 and homogeneity of indicator function. Using Lemma B.1,

$$\left| \mathbb{E}_{a_{r0} \sim \mathcal{N}(0, \epsilon_a^2), w_{r0} \sim \mathcal{N}\left(0, \frac{1}{m}\right), b_{r0} \sim \mathcal{N}\left(0, \frac{1}{m}\right)} \left[ P_\ell^*(x) \right] - \phi^{-1}\left( F^{*\prime}(x) \right) \right|$$

$$= \left| \mathbb{E}_{w_{r0} \sim \mathcal{N}\left(0, \frac{1}{m}\right), b_{r0} \sim \mathcal{N}\left(0, \frac{1}{m}\right)} \left[ h(\sqrt{m} w_{r0}, \sqrt{m} b_{r0}) \mathbb{I}\left[ \sqrt{m}\left( w_{r0}x + b_{r0} \right) \geq 0 \right] \right] - \phi^{-1}\left( F^{*\prime}(x) \right) \right|$$

$$\leq \omega_{\phi^{-1}(F^{*\prime})}(\delta) \tag{B.5}$$

Using technique from Yehudai & Shamir (2019), we define

$$h = h\left( (a_{10}, w_{10}, b_{10}), \ldots, (a_{r0}, w_{r0}, b_{r0}), \ldots, (a_{10}, w_{m0}, b_{m0}) \right) = \sup_{x \in [-1, 1]} |P_\ell^*(x) - \mathbb{E}_{a_{r0}, w_{r0}, b_{r0}} \left[ P_\ell^*(x) \right]|$$

We will use McDiarmid's inequality to bound $h$.

$$\left| h\left((a_{10}, w_{10}, b_{10}), \ldots, (a_{r0}, w_{r0}, b_{r0}), \ldots, (a_{10}, w_{m0}, b_{m0})\right) - h\left((a_{10}, w_{10}, b_{10}), \ldots, (a'_{r0}, w'_{r0}, b_{r0})', \ldots, (a_{10}, w_{m0}, b_{m0})\right) \right|$$

$$\leq \frac{4c_1 U_h \sqrt{2\log m}}{m}$$

Using Lemma 26.2 from Shalev-Shwartz & Ben-David (2014), we get

$$\mathbb{E}[h] = \frac{2}{m} \mathbb{E}_{a_{r0}, w_{r0}, b_{r0}, \xi_r} \left[ \sup_x m \left| \sum_{r=1}^{m} \xi_i \left(w_r^* x + b_r^*\right) \mathbb{I}\left[w_{r0} x + b_{r0} \geq 0\right] \right| \right]$$

where $\xi_1, \xi_2, \ldots, \xi_m$ are independent Rademacher random variables.

$$\mathbb{E}_{a_{r0}, w_{r0}, b_{r0}}[h] \leq \frac{2}{m} \mathbb{E}_{a_{r0}, w_{r0}, b_{r0}, \xi_r} \left[ \sup_x m \left| \sum_{r=1}^{m} \xi_i a_{r0} \left(w_r^* x + b_r^*\right) \mathbb{I}\left[w_{r0} x + b_{r0} \geq 0\right] \right| \right]$$

$$\leq \frac{2}{m} \mathbb{E}_{a_{r0}, w_{r0}, b_{r0}, \xi_r} \left[ \sup_x m \left| \sum_{r=1}^{m} \xi_i a_{r0} \left(w_r^* x + b_r^*\right) \mathbb{I}\left[w_{r0} x + b_{r0} \geq 0\right] \right| \right]$$

$$\leq \frac{8 c_1 \sqrt{\log m} U_h}{m} \mathbb{E}_{a_{r0}, w_{r0}, b_{r0}, \xi_r} \left[ \sup_x \left| \sum_{r=1}^{m} \xi_i \mathbb{I}\left[w_{r0} x + b_{r0} \geq 0\right] \right| \right]$$

One can show that

$$\frac{1}{m} \mathbb{E}_{a_{r0}, w_{r0}, b_{r0}, \xi_r} \left[ \sup_x \left| \sum_{r=1}^{m} \xi_i \mathbb{I}\left[w_{r0} x + b_{r0} \geq 0\right] \right| \right] \leq 2\sqrt{\frac{\log m}{m}}$$

Using this relation, we get

$$\mathbb{E}_{a_{r0}, w_{r0}, b_{r0}}[h] \leq \frac{16 c_1 U_h \log m}{\sqrt{m}}$$

Using Mcdiarmid's inequality, with at least $1 - \frac{1}{c_1} - \exp\left(-\frac{\epsilon^2 m}{128 c_1^2 U_h^2 \log m}\right)$, we have

$$\left| P_\ell^*(x) - \mathbb{E}_{a_{r0}, w_{r0}, b_{r0}}[P_\ell^*(x)] \right| = h = \leq \frac{\epsilon}{2} + \frac{16 c_1 U_h \log m}{\sqrt{m}} \overset{(i)}{\leq} \epsilon \tag{B.6}$$

where inequality (i) follows from our choice of $m$ in lemma D.2. Using eq.(B.5), we get

$$\left| P_\ell^*(x) - \phi^{-1}\left(F^{*\prime}(x)\right) \right| \leq \omega_{\phi^{-1}(F^{*\prime})}(\delta) + \epsilon \tag{B.7}$$

Using 1-Lipschitzness of $\phi$, we get

$$\left| \phi(P_\ell^*(x)) - F^{*\prime}(x) \right| = \left| \phi(P_\ell^*(x)) - \phi\left(\phi^{-1}\left(F^{*\prime}(x)\right)\right) \right|$$

$$\leq \left| P_\ell^*(x) - \phi^{-1}\left(F^{*\prime}(x)\right) \right|$$

$$\leq \omega_{\phi^{-1}(F^{*\prime})}(\delta) + \epsilon$$

The upper bound on norm of $\|\theta^*\|_\infty$ is given by the following equation.

$$\|\theta^*\|_\infty \leq \frac{U_h \sqrt{\pi}}{\sqrt{2} m \epsilon_a}$$

$\square$

**Corollary B.1.** *(Approximating target network using $P(x)$) For every positive function $F^{*\prime}$ and for every $\epsilon \in (0, 1)$, with at least $0.99 - \frac{1}{c_1} - \frac{1}{c_6} - \frac{1}{c_7} - \exp\left(-\frac{\epsilon^2 m}{128 c_1^2 U_h^2 \log m}\right)$ probability over random initialization, there exists $\theta^*$ such that we have following inequality for all $x \in [-1, 1]$ and some fixed positive constants $c_1 > 1, c_6 > 1$ and $c_7 > 1$.*

$$\left| \phi\left(P^*(x)\right) - F^{*\prime}(x) \right| \leq 16 c_1 \left(c_6 + c_7\right) \epsilon_a \log m + \omega_{\phi^{-1}(F^{*\prime})}(\delta) + \epsilon$$

*and upper bound on $L_\infty$ norm of parameters $\theta^*$ is given by*

$$\|\theta^*\|_\infty \leq \frac{U_h \sqrt{\pi}}{\sqrt{2} m \epsilon_a}$$

*Proof.* Using Lipschitz continuity of $\phi$ function, we get

$$|\phi\left(P_\ell^*(x)\right) - \phi(P^*(x))| \leq |P_\ell^*(x) - P^*(x)|$$

$$\leq \left| \sum_{r=1}^{m} a_{r0} \left(w_{r0}x + b_{r0}\right) \mathbb{I}\left[w_{r0}x + b_{r0} \geq 0\right] \right|$$

Now, there are at most $m$ break points of indicator $\mathbb{I}\left[w_{r0}x + b_{r0} \geq 0\right]$ where value of $\mathbb{I}\left[w_{r0}x + b_{r0} \geq 0\right]$ changes. We can divide range of $x$ into at most $m+1$ subsets where in each subset, value of indicators $\mathbb{I}\left[w_{r0}x + b_{r0} \geq 0\right]$ is fixed for all $r$. Suppose there are $m'$ indicators with value 1 in a given subset. Without loss of generality, we can assume that indicators from $r = 1$ to $r = m'$ is 1. Then,

$$\left| \sum_{r=1}^{m} a_{r0} \left(w_{r0}x + b_{r0}\right) \mathbb{I}\left[w_{r0}x + b_{r0} \geq 0\right] \right| = \left| \sum_{r=1}^{m'} a_{r0} \left(w_{r0}x + b_{r0}\right) \right|$$

$$\leq \left| x \sum_{r=1}^{m'} a_{r0}w_{r0} + \sum_{r=1}^{m'} a_{r0}b_{r0} \right|$$

Now, applying Hoeffding's inequality for the sum in above equation, we get

$$\text{Pr}\left( \left| \sum_{r=1}^{m'} a_{r0}w_{r0} \right| \geq t \right) \leq \exp\left( -\frac{2t^2 m}{m'\left(2c_1\epsilon_a\sqrt{2\log m}\right)^2\left(2c_6\sqrt{2\log m}\right)^2} \right)$$

$$= \exp\left( -\frac{t^2}{32c_1^2 c_6^2 \epsilon_a^2 \left(\log m\right)^2} \right)$$

Taking $t = 16c_1 c_6 \epsilon_a \left(\log m\right)$, with at least probability $0.999 - \frac{1}{c_1} - \frac{1}{c_6}$, we have

$$\left| \sum_{r=1}^{m'} a_{r0}w_{r0} \right| \leq 16c_1 c_6 \epsilon_a \left(\log m\right)$$

and similarly, we will get that with at least $0.999 - \frac{1}{c_1} - \frac{1}{c_7}$ probability,

$$\left| \sum_{r=1}^{m'} a_{r0}w_{r0} \right| \leq 16c_1 c_7 \epsilon_a \left(\log m\right)$$

we will get that at least $0.999 - \frac{1}{c_1} - \frac{1}{c_6} - \frac{1}{c_7}$ probability, we have

$$\left| \sum_{r=1}^{m} a_{r0}w_{r0}\mathbb{I}\left[w_{r0}x + b_{r0} \geq 0\right] \right| \leq 16c_1 c_6 \epsilon_a \left(\log m\right) \tag{B.8}$$

$$\left| \sum_{r=1}^{m} a_{r0}b_{r0}\mathbb{I}\left[w_{r0}x + b_{r0} \geq 0\right] \right| \leq 16c_1 c_7 \epsilon_a \left(\log m\right)$$

Using these relations, we get that with at least $0.99 - \frac{1}{c_1} - \frac{1}{c_6} - \frac{1}{c_7}$ probability,

$$\left| \sum_{r=1}^{m} a_{r0} \left(w_{r0}x + b_{r0}\right) \mathbb{I}\left[w_{r0}x + b_{r0} \geq 0\right] \right| \leq 16c_1 \left(c_6 + c_7\right) \epsilon_a \log m \tag{B.9}$$

Using above inequality, we get

$$|\phi\left(P_\ell^*(x)\right) - \phi(P^*(x))| \leq |P_\ell^*(x) - P^*(x)| \leq 16c_1 \left(c_6 + c_7\right) \epsilon_a \log m$$

Using lemma B.2, with at least $0.99 - \frac{1}{c_1} - \frac{1}{c_6} - \frac{1}{c_7} - \exp\left(-\frac{\epsilon^2 m}{128 c_1^2 U_h^2 \log m}\right)$ probability,

$$|\phi(P^*(x)) - F^{*\prime}(x)| \le |\phi(P^*(x)) - \phi(P_\ell^*(x))| + |\phi(P_\ell^*(x)) - F^{*\prime}(x)|$$
$$\le 16 c_1 (c_6 + c_7) \epsilon_a \log m + \omega_{\phi^{-1}(F^{*\prime})}(\delta) + \epsilon$$

$\square$

**Lemma B.3.** *(Optimal loss) For every positive function $F^{*\prime}$ and for every $\epsilon \in (0,1)$, with at least $0.99 - \frac{1}{c_1} - \frac{1}{c_6} - \frac{1}{c_7} - \exp\left(-\frac{\epsilon^2 m}{128 c_1^2 U_h^2 \log m}\right)$ probability over random initialization, there exist $\theta^*$ such that loss of pseudo network with $\theta^*$ parameters is close to that of the target function for all $x \in [-1, 1]$ and for some fixed positive constants $c_1 > 1, c_6 > 1$ and $c_7 > 1$.*

$$\left| \hat{L}(\phi(P^*), x) - \hat{L}(F^{*\prime}, x) \right| \le 3\left(16 c_1 (c_6 + c_7) \epsilon_a \log m + \omega_{\phi^{-1}(F^{*\prime})}(\delta) + \epsilon\right)$$

*Proof.*

$$\left| \hat{L}(\phi(P^*), x) - \hat{L}(F^{*\prime}, x) \right| \le \left| \sum_{i=1}^{Q} \Delta_x \phi(P^*(\tau_i(x))) - \sum_{i=1}^{Q} \Delta_x F^{*\prime}(\tau_i(x)) \right|$$
$$+ \left| \log(\phi(P^*(x))) - \log(F^{*\prime}(x)) \right|$$
$$\overset{(i)}{\le} 2\left(16 c_1 (c_6 + c_7) \epsilon_a \log m + \omega_{\phi^{-1}(F^{*\prime})}(\delta) + \epsilon\right)$$
$$+ \left| P^*(x) - \phi^{-1}(F^{*\prime}(x)) \right|$$
$$\le 2\left(16 c_1 (c_6 + c_7) \epsilon_a \log m + \omega_{\phi^{-1}(F^{*\prime})}(\delta) + \epsilon\right)$$
$$+ \left| P_c^*(x) \right| + \left| P_\ell^*(x) - \phi^{-1}(F^{*\prime}(x)) \right|$$
$$\overset{(ii)}{\le} 3\left(16 c_1 (c_6 + c_7) \epsilon_a \log m + \omega_{\phi^{-1}(F^{*\prime})}(\delta) + \epsilon\right)$$

where inequality (i) follows from Corollary B.1 with at least $0.99 - \frac{1}{c_1} - \frac{1}{c_6} - \frac{1}{c_7} - \exp\left(-\frac{\epsilon^2 m}{128 c_1^2 U_h^2 \log m}\right)$ probability. Inequality (ii) uses Eq.(B.7) and Eq.(B.9). $\square$

## C  COUPLING

In this section, we prove that, for random initialization, the gradients of the loss of pseudo network closely approximate the gradients of the loss of the target function. In other words, we show coupling of their gradient-based optimizations. Define $\lambda_1$ as

$$\lambda_1 = \sup_{t \in [T], r \in [m], w_r^t, b_r^t, |x| \le 1} \frac{\phi'(N_t(x))}{\phi(N_t(x))} \tag{C.1}$$

We get following find upper bound on $\lambda_1$.

$$\lambda_1 = \sup_{t \in [T], r \in [m], w_r^t, b_r^t, |x| \le 1} \frac{\phi'(N_t(x))}{\phi(N_t(x))}$$
$$= \sup_{t \in [T], r \in [m], w_r^t, b_r^t, |x| \le 1} \frac{\exp(N_t(x)) \mathbb{I}[N_t(x) < 0] + \mathbb{I}[N_t(x) \ge 0]}{\exp(N_t(x)) \mathbb{I}[N_t(x) < 0] + (N_t(x) + 1) \mathbb{I}[N_t(x) \ge 0]}$$
$$= \sup_{t \in [T], r \in [m], w_r^t, b_r^t, |x| \le 1} \mathbb{I}[N_t(x) < 0] + \frac{\mathbb{I}[N_t(x) \ge 0]}{N_t(x) + 1}$$
$$= 1 \tag{C.2}$$

Define $\bar{\Delta}$ as

$$\bar{\Delta} = 6 c_1 \epsilon_a \sqrt{2 \log m} \tag{C.3}$$

for some positive constant $c_1 > 1$.

**Lemma C.1.** *(Bound in change in patterns) For every $x$ in 1 radius ($|x| \leq 1$) and for every time step $t \geq 1$, with probability at least $1 - \frac{1}{c_1} - \exp\left(-\frac{64(c_2-1)^2\eta^2m^2\bar{\Delta}^2t^2}{\pi}\right)$ over random initialization, for at most $c_2\frac{4\sqrt{2}\eta\sqrt{m}\bar{\Delta}t}{\sqrt{\pi}}$ fraction of $r \in [m]$*

$$\mathbb{I}\left[(w_{r0} + w_r^t)x + b_{r0} + b_r^t \geq 0\right] \neq \mathbb{I}\left[w_{r0}x + b_{r0} \geq 0\right]$$

*for some positive constant $c_1 > 1$ and $c_2 \geq 1$.*

*Proof.* Taking derivative of $\hat{L}(f', x)$ wrt $w_r$,

$$\left|\frac{\partial\hat{L}(f_t', x)}{\partial w_r}\right| = \left|\left(\sum_{i=1}^{Q}\Delta_x\phi'\left(N_t\left(\tau_i\left(x\right)\right)\right)a_{r0}\sigma'\left((w_{r0} + w_r^t)\tau_i\left(x\right) + b_{r0} + b_r^t\right)\tau_i\left(x\right)\right)\right|$$
$$+ \left|\frac{1}{\phi(N_t(x))}\left(\phi'(N_t(x))a_{r0}\sigma'\left((w_{r0} + w_r^t)x + b_{r0} + b_r^t\right)x\right)\right|$$
$$\leq \sum_{i=1}^{Q}\left|\Delta_x\phi'\left(N_t\left(\tau_i\left(x\right)\right)\right)a_{r0}\sigma'\left((w_{r0} + w_r^t)\tau_i\left(x\right) + b_{r0} + b_r^t\right)\tau_i\left(x\right)\right|$$
$$+ \left|\frac{\phi'(N_t(x))}{\phi(N_t(x))}\right|\left|\left(a_{r0}\sigma'\left((w_{r0} + w_r^t)x + b_{r0} + b_r^t\right)x\right)\right|$$

Using Eq.(C.2), $\Delta_x \leq \frac{2}{Q}$, $|x| \leq 1$ and $|\phi'\left(N(x)\right)| \leq 1$ for all $x \in [-1, 1]$, we get

$$\left|\frac{\partial\hat{L}(f_t', x)}{\partial w_r}\right| \leq 3|a_{r0}|$$

Using Lemma H.2, with at least $1 - \frac{1}{c_1}$ probability, we get

$$\left|\frac{\partial\hat{L}(f_t', x)}{\partial w_r}\right| \leq \bar{\Delta} \tag{C.4}$$

where $\bar{\Delta}$ is defined in Eq.(C.3). Using same procedure for $b_r$, we get

$$\left|\frac{\partial\hat{L}(f_t', x)}{\partial b_r}\right| = \left|\sum_{i=1}^{Q}\Delta_x\phi'\left(N_t\left(\tau_i\left(x\right)\right)\right)a_{r0}\sigma'\left((w_{r0} + w_r^t)\tau_i\left(x\right) + b_{r0} + b_r^t\right)\right|$$
$$+ \left|\frac{1}{\phi(N_t(x))}\left(\phi'(N_t(x))a_{r0}\sigma'\left((w_{r0} + w_r^t)x + b_{r0} + b_r^t\right)\right)\right|$$
$$\leq 3|a_{r0}|$$
$$= \bar{\Delta} \tag{C.5}$$

Using Eq.(C.4) and Eq.(C.5), we get

$$|w_r^t| \leq \eta\bar{\Delta}t$$
$$|b_r^t| \leq \eta\bar{\Delta}t \tag{C.6}$$

Define

$$\mathcal{H}_t = \{r \in [m] \mid |w_{r0}x + b_{r0}| \geq 4\eta\bar{\Delta}t\} \tag{C.7}$$

For every $x$ with $|x| \leq 1$ and for all $r \in [m]$, $|w_r^t x + b_r^t| \leq 2\eta\bar{\Delta}t$. For all $r \in \mathcal{H}_t$, we get $\mathbb{I}\left[(w_{r0} + w_r^t)x + b_{r0} + b_r^t \geq 0\right] = \mathbb{I}\left[w_{r0}x + b_{r0} \geq 0\right]$. Now, we need to bound the size of $\mathcal{H}_t$. We know that for all $x \in [-1, 1]$, $w_{r0}x + b_{r0}$ is Gaussian with $\mathbb{E}\left[w_{r0}x + b_{r0}\right] = 0$ and $\text{Var}\left[w_{r0}x + b_{r0}\right] \geq \frac{1}{m}$. Using Lemma H.3, we get

$$\Pr\left(|w_{r0}x + b_{r0}| \leq 4\eta\bar{\Delta}t\right) \leq \frac{4\sqrt{2}\eta\sqrt{m}\bar{\Delta}t}{\sqrt{\pi}}$$

Using Fact H.1 for $\mathcal{H}_t^c$ (where $\mathcal{H}_t^c = [m]/\mathcal{H}_t$) for some positive constant $c_2 \geq 1$, we get

$$\Pr\left(|\mathcal{H}_t^c| \geq c_2 m \frac{4\sqrt{2}\eta\sqrt{m}\bar{\Delta}t}{\sqrt{\pi}}\right) \leq \exp\left(-2m\left((c_2 - 1)\left(\frac{4\sqrt{2}\eta\sqrt{m}\bar{\Delta}t}{\sqrt{\pi}}\right)\right)^2\right)$$

$$\leq \exp\left(-\frac{64(c_2 - 1)^2\eta^2 m^2\bar{\Delta}^2 t^2}{\pi}\right)$$

$$\Pr\left(|\mathcal{H}_t^c| \leq c_2 m \frac{4\sqrt{2}\eta\sqrt{m}\bar{\Delta}t}{\sqrt{\pi}}\right) \geq 1 - \exp\left(-\frac{64(1 - c_2)^2\eta^2 m^2\bar{\Delta}^2 t^2}{\pi}\right)$$

$$\Pr\left(|\mathcal{H}_t| \geq m\left(1 - c_2\frac{4\sqrt{2}\eta\sqrt{m}\bar{\Delta}t}{\sqrt{\pi}}\right)\right) \geq 1 - \exp\left(-\frac{64(1 - c_2)^2\eta^2 m^2\bar{\Delta}^2 t^2}{\pi}\right)$$

where $|\mathcal{H}_t|$ denotes the cardinality of set $\mathcal{H}_t$ and similarly for $|\mathcal{H}_t^c|$. $\qquad\square$

**Lemma C.2.** *(Bound on difference of $f'$ and $g'$) For every $x$ in 1 radius ($|x| \leq 1$) and for every time step $t \geq 1$, with at least $1 - \frac{1}{c_1}$ probability, function with neural network and function with pseudo network are close for some positive constants $c_1 > 1$.*

$$|\phi(N_t(x)) - \phi(P_t(x))| \leq 24c_1\epsilon_a\eta\bar{\Delta}t\left|\mathcal{H}_c^t\right|\sqrt{2\log m}$$

*Proof.* We know that $\phi$ is 1-Lipschitz continuous. Using Lipschitz continuity of $\phi$, we get

$$|\phi(N_t(x)) - \phi(P_t(x))| \leq |N_t(x) - P_t(x)|$$

We bound $|N_t(x) - P_t(x)|$ as following.

$$|N_t(x) - P_t(x)| \leq \left|\sum_{r\in[m]} a_{r0}\left((w_{r0} + w_r^t)x + b_{r0} + b_r^t\right)\mathbb{I}\left[(w_{r0} + w_r^t)x + b_{r0} + b_r^t \geq 0\right]\right.$$

$$\left. - \sum_{r\in[m]} a_{r0}\left((w_{r0} + w_r^t)x + b_{r0} + b_r^t\right)\mathbb{I}\left[w_{r0}x + b_{r0} \geq 0\right]\right|$$

$$\leq \left|\sum_{r\notin\mathcal{H}_t} a_{r0}\left((w_{r0} + w_r^t)x + b_{r0} + b_r^t\right)\left(\mathbb{I}\left[(w_{r0} + w_r^t)x + b_{r0} + b_r^t \geq 0\right] - \mathbb{I}\left[w_{r0}x + b_{r0} \geq 0\right]\right)\right|$$

$$\overset{(i)}{\leq} \left|\mathcal{H}_c^t\right|\left(2c_1\epsilon_a\sqrt{2\log m}\right)\left(4\eta\bar{\Delta}t + 2\eta\bar{\Delta}t\right) \quad (2)$$

$$\leq 24c_1\epsilon_a\eta\bar{\Delta}t\left|\mathcal{H}_c^t\right|\sqrt{2\log m} \tag{C.8}$$

where inequality (i) uses Lemma H.2 with at least $1 - \frac{1}{c_1}$ probability. $\qquad\square$

**Corollary C.1.** *(Final bound on difference of $f'$ and $g'$) For every $x$ in 1 radius ($|x| \leq 1$) and for every time step $t \geq 1$, with at least $1 - \frac{1}{c_1} - \exp\left(-\frac{64(c_2-1)^2\eta^2 m^2\bar{\Delta}^2 t^2}{\pi}\right)$ probability over random initialization, function with neural network and function with pseudo network are close for some positive constants $c_1 > 1$ and $c_2 \geq 1$.*

$$|\phi(N_t(x)) - \phi(P_t(x))| \leq \frac{192\eta^2 m^{1.5}\bar{\Delta}^2 c_1 c_2\epsilon_a t^2\sqrt{\log m}}{\sqrt{\pi}} \tag{C.9}$$

*Proof.* Using Lemma C.1 and Lemma C.2, we get

$$
\begin{aligned}
|\phi(N_t(x)) - \phi(P_t(x))| &\leq 24c_1\epsilon_a\eta\bar{\Delta}t \left|\mathcal{H}_c^t\right| \sqrt{2\log m} \\
&\overset{(i)}{\leq} 24c_1\epsilon_a\eta\bar{\Delta}t \left(c_2m\frac{4\sqrt{2}\eta\sqrt{m}\bar{\Delta}t}{\sqrt{\pi}}\right) \sqrt{2\log m} \\
&\leq \left(\frac{192\eta m^{1.5}\bar{\Delta}c_1c_2\epsilon_a t\sqrt{\log m}}{\sqrt{\pi}}\right) (\eta\bar{\Delta}t) \\
&= \frac{192\eta^2 m^{1.5}\bar{\Delta}^2 c_1c_2\epsilon_a t^2\sqrt{\log m}}{\sqrt{\pi}} \\
&\leq O(\eta^2 m^{1.5}\bar{\Delta}^2\epsilon_a t^2\sqrt{\log m})
\end{aligned}
\tag{C.10}
$$

where inequality (i) uses Lemma C.1 and the inequality follows with at least $1 - \frac{1}{c_1} - \exp\left(-\frac{64(c_2-1)^2\eta^2 m^2\bar{\Delta}^2 t^2}{\pi}\right)$ probability. Define $\Delta_{np}^t$ as

$$
\Delta_{np}^t = \frac{192\eta^2 m^{1.5}\bar{\Delta}^2 c_1c_2\epsilon_a t^2\sqrt{\log m}}{\sqrt{\pi}}
\tag{C.11}
$$

$\square$

**Lemma C.3.** *(Coupling of loss functions) For all $x$ in 1 radius ($|x| \leq 1$) and for every time step $t \geq 1$, with probability at least $1 - \frac{1}{c_1} - \exp\left(-\frac{64(c_2-1)^2\eta^2 m^2\bar{\Delta}^2 t^2}{\pi}\right)$ over random initialization, loss function of neural network and pseudo network are close for some positive constant $c_1 > 1$ and $c_2 \geq 1$.*

$$
\left|\hat{L}\left(f_t', x\right) - \hat{L}\left(g_t', x\right)\right| \leq 3\Delta_{np}^t
$$

*Proof.*

$$
\begin{aligned}
\left|\hat{L}\left(f_t', x\right) - \hat{L}\left(g_t', x\right)\right| &\leq \left|\sum_{i=1}^{Q}\Delta_x f_t'(\tau_i(x)) - \sum_{i=1}^{Q}\Delta_x g_t'(\tau_i(x))\right| + \left|\log\left(f_t'(x)\right) - \log\left(g_t'(x)\right)\right| \\
&\overset{(i)}{\leq} 2\left(\sup_{i\in[Q]}|f_t'(\tau_i(x)) - g_t'(\tau_i(x))|\right) + |N_t(x) - P_t(x)| \\
&\overset{(ii)}{\leq} 3\Delta_{np}^t
\end{aligned}
$$

where inequality (i) follows from 1-Lipschitz continuity of $\log(\phi(N(x)))$ with respect to $N(x)$. Inequality (ii) uses Eq.(C.8) and Lemma C.2. $\square$

**Lemma C.4.** *(Coupling of gradient of functions) For all $x$ in 1 radius ($|x| \leq 1$) and for every time step $t \geq 1$, with at least $1 - \frac{1}{c_1}$ probability over random initialization, gradient of derivative of neural network function and derivative of pseudo network function with respect to parameters are close for some positive constant $c_1 > 1$.*

$$
\left\|\nabla_\theta f_t'(x) - \nabla_\theta g_t'(x)\right\|_1 \leq 4c_1\epsilon_a\left(m\Delta_{np}^t + 2\left|\mathcal{H}_t^c\right|\right)\sqrt{2\log m}
$$

*Proof.*

$$
\begin{aligned}
\left\|\nabla_\theta f_t'(x) - \nabla_\theta g_t'(x)\right\|_1 &\leq \left\|\phi'(N_t(x))\nabla_\theta N_t(x) - \phi'(P_t(x))\nabla_\theta P_t(x)\right\|_1 \\
&\leq \left\|\phi'(N_t(x))\nabla_\theta N_t(x) - \phi'(P_t(x))\nabla_\theta N_t(x)\right\|_1 + \left\|\phi'(P_t(x))\nabla_\theta N_t(x) - \phi'(P_t(x))\nabla_\theta P_t(x)\right\|_1 \\
&\leq |\phi'(N_t(x)) - \phi'(P_t(x))|\left\|\nabla_\theta N_t(x)\right\|_1 + |\phi'(P_t(x))|\left\|\nabla_\theta N_t(x) - \nabla_\theta P_t(x)\right\|_1 \\
&\leq |N_t(x) - P_t(x)|\left\|\nabla_\theta N_t(x)\right\|_1 + \left\|\nabla_\theta N_t(x) - \nabla_\theta P_t(x)\right\|_1
\end{aligned}
$$

where last inequality follows from 1-Lipschitzness of $\phi'$ function and $\phi'(x) \leq 1$ for all $x$ such that $|x| \leq 1, t \in [T]$. To upper bound $\left\|\nabla_\theta N_t(x) - \nabla_\theta P_t(x)\right\|_1$,

$$
\begin{aligned}
\left\|\nabla_\theta N_t(x) - \nabla_\theta P_t(x)\right\|_1 &\leq \left\|(A_0, A_0) \odot (\mathbf{1}x, \mathbf{1}) \odot (\mathbb{I}\left[(W_0 + W^t)x + B_0 + B^t \geq 0\right] - \mathbb{I}\left[W_0 x + B_0 \geq 0\right], \right. \\
&\quad \left. \mathbb{I}\left[(W_0 + W^t)x + B_0 + B^t \geq 0\right] - \mathbb{I}\left[W_0 x + B_0 \geq 0\right])\right\|_1 \\
&\overset{(i)}{\leq} \left(8c_1\epsilon_a\sqrt{2\log m}\right)|\mathcal{H}_t^c| \\
&\leq 8c_1\epsilon_a|\mathcal{H}_t^c|\sqrt{2\log m}
\end{aligned} \tag{C.12}
$$

The inequality (i) uses property of $\mathcal{H}_t$ that for all $r \in \mathcal{H}_t, \mathbb{I}\left[(w_{r0} + w_r^t)x + b_{r0} + b_r^t \geq 0\right] = \mathbb{I}\left[w_{r0}x + b_{r0} \geq 0\right]$. Using Eq.(C.11) and Eq.(C.12), we get

$$
\begin{aligned}
\left\|\nabla_\theta f_t'(x) - \nabla_\theta g_t'(x)\right\|_1 &\leq |N_t(x) - P_t(x)|\left\|(A_0, A_0) \odot (\mathbf{1}x, \mathbf{1}) \odot (\mathbb{I}\left[(W_0 + W^t)x + B_0 + B^t \geq 0\right], \right. \\
&\quad \left. \mathbb{I}\left[(W_0 + W^t)x + B_0 + B^t \geq 0\right])\right\|_1 + \left\|\nabla_\theta N_t(x) - \nabla_\theta P_t(x)\right\|_1 \\
&\leq 4c_1\epsilon_a m\Delta_{np}^t\sqrt{2\log m} + 8c_1\epsilon_a|\mathcal{H}_t^c|\sqrt{2\log m} \\
&= 4c_1\epsilon_a\left(m\Delta_{np}^t + 2|\mathcal{H}_t^c|\right)\sqrt{2\log m}
\end{aligned}
$$

$\square$

**Lemma C.5.** *(Coupling of gradient of loss) For all $x$ in 1 radius ($|x| \leq 1$) and for every time step $t \geq 1$, with probability at least $1 - \frac{1}{c_1} - \exp\left(-\frac{64(c_2-1)^2\eta^2 m^2\bar{\Delta}^2 t^2}{\pi}\right)$ over random initialization, gradient of loss function with neural network and loss function with pseudo network are close for some positive constant $c_1 > 1$ and $c_2 \geq 1$.*

$$
\left\|\nabla_\theta\hat{L}(f_t', x) - \nabla_\theta\hat{L}(g_t', x)\right\|_1 \leq \frac{192\eta m^{1.5}\bar{\Delta}c_1c_2\epsilon_a t\sqrt{\log m}}{\sqrt{\pi}} + 16c_1\epsilon_a m\Delta_{np}^t\sqrt{2\log m}
$$

*Proof.*

$$
\begin{aligned}
\left\|\nabla_\theta\hat{L}(f_t', x) - \nabla_\theta\hat{L}(g_t', x)\right\|_1 &\leq \left\|\sum_{i=1}^{Q}\Delta_x\nabla_\theta f_t'(\tau_i(x)) - \frac{\nabla_\theta f_t'(x)}{f_t'(x)} \right. \\
&\quad \left. - \sum_{i=1}^{Q}\Delta_x\nabla_\theta g_t'(\tau_i(x)) + \frac{\nabla_\theta g_t'(x)}{g_t'(x)}\right\|_1 \\
&\leq \underbrace{\left\|\sum_{i=1}^{Q}\Delta_x\nabla_\theta f_t'(\tau_i(x)) - \sum_{i=1}^{Q}\Delta_x\nabla_\theta g_t'(\tau_i(x))\right\|_1}_{\text{I}} \\
&\quad + \underbrace{\left\|\frac{\nabla_\theta g_t'(x)}{g_t'(x)} - \frac{\nabla_\theta f_t'(x)}{f_t'(x)}\right\|_1}_{\text{II}}
\end{aligned}
$$

Proving bound on I,

$$
\begin{aligned}
\text{I} &= \left\|\sum_{i=1}^{Q}\Delta_x\nabla_\theta f_t'(\tau_i(x)) - \sum_{i=1}^{Q}\Delta_x\nabla_\theta g_t'(\tau_i(x))\right\|_1 \\
&\leq \sum_{i=1}^{Q}\Delta_x\left\|\nabla_\theta f_t'(\tau_i(x)) - \nabla_\theta g_t'(\tau_i(x))\right\|_1 \\
&\overset{(i)}{\leq} 8c_1\epsilon_a\left(m\Delta_{np}^t + 2|\mathcal{H}_t^c|\right)\sqrt{2\log m}
\end{aligned}
$$

where inequality (i) follows from Lemma C.4. Now, we will bound II,

$$
\begin{aligned}
\text{II} =& \left\| \frac{\nabla_\theta g_t'(x)}{g_t'(x)} - \frac{\nabla_\theta f_t'(x)}{f_t'(x)} \right\|_1 \\
=& \left\| \frac{\exp\left(P_t(x)\right) \mathbb{I}\left[P_t(x) < 0\right] + \mathbb{I}\left[P_t(x) \geq 0\right]}{\exp\left(P_t(x)\right) \mathbb{I}\left[P_t(x) < 0\right] + \left(P_t(x) + 1\right) \mathbb{I}\left[P_t(x) \geq 0\right]} \nabla_\theta P_t(x) \right. \\
& \left. - \frac{\exp\left(N_t(x)\right) \mathbb{I}\left[N_t(x) < 0\right] + \mathbb{I}\left[N_t(x) \geq 0\right]}{\exp\left(N_t(x)\right) \mathbb{I}\left[N_t(x) < 0\right] + \left(N_t(x) + 1\right) \mathbb{I}\left[N_t(x) \geq 0\right]} \nabla_\theta N_t(x) \right\|_1 \\
=& \left\| \left( \mathbb{I}\left[P_t(x) < 0\right] + \frac{\mathbb{I}\left[P_t(x) \geq 0\right]}{\left(P_t(x) + 1\right)} \right) \nabla_\theta P_t(x) - \left( \mathbb{I}\left[N_t(x) < 0\right] + \frac{\mathbb{I}\left[N_t(x) \geq 0\right]}{\left(N_t(x) + 1\right)} \right) \nabla_\theta N_t(x) \right\|_1 \\
=& \underbrace{\left\| \nabla_\theta P_t(x) - \nabla_\theta N_t(x) \right\|_1 \mathbb{I}\left[P_t(x) < 0, N_t(x) < 0\right]}_{\text{II}_1} \\
& + \underbrace{\left\| \nabla_\theta P_t(x) - \frac{\nabla_\theta N_t(x)}{N_t(x) + 1} \right\|_1 \mathbb{I}\left[P_t(x) < 0, N_t(x) \geq 0\right]}_{\text{II}_2} \\
& + \underbrace{\left\| \frac{\nabla_\theta P_t(x)}{P_t(x) + 1} - \nabla_\theta N_t(x) \right\|_1 \mathbb{I}\left[P_t(x) \geq 0, N_t(x) < 0\right]}_{\text{II}_3} \\
& + \underbrace{\left\| \frac{\nabla_\theta P_t(x)}{P_t(x) + 1} - \frac{\nabla_\theta N_t(x)}{N_t(x) + 1} \right\|_1 \mathbb{I}\left[P_t(x) \geq 0, N_t(x) \geq 0\right]}_{\text{II}_4}
\end{aligned}
$$

On simplifying $\text{II}_2$, we get

$$
\begin{aligned}
\text{II}_2 \leq & \left( \left| \frac{1}{N_t(x) + 1} \right| \left\| \nabla_\theta P_t(x) - \nabla_\theta N_t(x) \right\|_1 + \left| \frac{N_t(x)}{1 + N_t(x)} \right| \left\| \nabla_\theta P_t(x) \right\|_1 \right) \mathbb{I}\left[P_t(x) < 0, N_t(x) \geq 0\right] \\
\leq & \left( \left\| \nabla_\theta P_t(x) - \nabla_\theta N_t(x) \right\|_1 + \Delta_{np}^t \left\| \nabla_\theta P_t(x) \right\|_1 \right) \mathbb{I}\left[P_t(x) < 0, N_t(x) \geq 0\right] \qquad (\text{C.13})
\end{aligned}
$$

Similarly, on simplifying $\text{II}_3$, we get

$$
\begin{aligned}
\text{II}_3 \leq & \left( \left| \frac{1}{P_t(x) + 1} \right| \left\| \nabla_\theta P_t(x) - \nabla_\theta N_t(x) \right\|_1 + \left| \frac{P_t(x)}{1 + P_t(x)} \right| \left\| \nabla_\theta N_t(x) \right\|_1 \right) \mathbb{I}\left[P_t(x) \geq 0, N_t(x) < 0\right] \\
\leq & \left( \left\| \nabla_\theta P_t(x) - \nabla_\theta N_t(x) \right\|_1 + \Delta_{np}^t \left\| \nabla_\theta N_t(x) \right\|_1 \right) \mathbb{I}\left[P_t(x) \geq 0, N_t(x) < 0\right] \qquad (\text{C.14})
\end{aligned}
$$

On simplifying $\text{II}_4$, we get

$$
\begin{aligned}
\text{II}_4 \leq & \left( \left\| \frac{\nabla_\theta P_t(x)}{P_t(x) + 1} - \frac{\nabla_\theta N_t(x)}{P_t(x) + 1} \right\|_1 + \left\| \frac{\nabla_\theta N_t(x)}{P_t(x) + 1} - \frac{\nabla_\theta N_t(x)}{N_t(x) + 1} \right\|_1 \right) \mathbb{I}\left[P_t(x) \geq 0, N_t(x) \geq 0\right] \\
\leq & \left( \frac{1}{P_t(x) + 1} \left\| \nabla_\theta P_t(x) - \nabla_\theta N_t(x) \right\|_1 + \frac{\left\| \nabla_\theta N_t(x) \right\|_1 \Delta_{np}^t}{\left(P_t(x) + 1\right)\left(N_t(x) + 1\right)} \right) \mathbb{I}\left[P_t(x) \geq 0, N_t(x) \geq 0\right] \\
\leq & \left( \left\| \nabla_\theta P_t(x) - \nabla_\theta N_t(x) \right\|_1 + \left\| \nabla_\theta N_t(x) \right\|_1 \Delta_{np}^t \right) \mathbb{I}\left[P_t(x) \geq 0, N_t(x) \geq 0\right] \qquad (\text{C.15})
\end{aligned}
$$

Using Eq.(C.13), Eq.(C.14) and Eq.(C.15), we get

$$
\begin{aligned}
\mathrm{II} &= \left\| \frac{\nabla_\theta g'_t(x)}{g'_t(x)} - \frac{\nabla_\theta f'_t(x)}{f'_t(x)} \right\|_1 \\
&\leq \left\| \nabla_\theta P_t(x) - \nabla_\theta N_t(x) \right\|_1 + \left\| \nabla_\theta N_t(x) \right\|_1 \Delta_{np}^t \mathbb{I}\left[P_t(x) \geq 0\right] \\
&\quad + \Delta_{np}^t \left\| \nabla_\theta P_t(x) \right\|_1 \mathbb{I}\left[P_t(x) < 0, N_t(x) \geq 0\right]
\end{aligned}
$$

Using Eq.(C.12), we get

$$
\begin{aligned}
\mathrm{II} &\leq 8c_1\epsilon_a \left|\mathcal{H}_t^c\right| \sqrt{2\log m} + \Delta_{np}^t \left( \left\| \nabla_\theta N_t(x) \right\|_1 + \left\| \nabla_\theta P_t(x) \right\|_1 \right) \\
&\leq 8c_1\epsilon_a \left|\mathcal{H}_t^c\right| \sqrt{2\log m} + \Delta_{np}^t \Bigg( \left\| (A_0, A_0) \odot (\mathbf{1}x, \mathbf{1}) \odot \left( \mathbb{I}\left[W_0 x + B_0 \geq 0\right], \mathbb{I}\left[W_0 x + B_0 \geq 0\right] \right) \right\|_1 \\
&\quad + \left\| (A_0, A_0) \odot (\mathbf{1}x, \mathbf{1}) \odot \left( \mathbb{I}\left[(W_0 + W^t)x + B_0 + B^t \geq 0\right], \mathbb{I}\left[(W_0 + W^t)x + B_0 + B^t \geq 0\right] \right) \right\|_1 \Bigg) \\
&\leq 8c_1\epsilon_a \left|\mathcal{H}_t^c\right| \sqrt{2\log m} + \Delta_{np}^t \left( 8c_1\epsilon_a m \sqrt{2\log m} \right) \\
&= 8c_1\epsilon_a \left( \left|\mathcal{H}_t^c\right| + m\Delta_{np}^t \right) \sqrt{2\log m} \qquad\qquad\qquad (\mathrm{C.16})
\end{aligned}
$$

Combining bounds on I and II, we get

$$
\begin{aligned}
\left\| \nabla_\theta \hat{L}(f'_t, x) - \nabla_\theta \hat{L}(g'_t, x) \right\|_1 &\leq 8c_1\epsilon_a \left( m\Delta_{np}^t + 2\left|\mathcal{H}_t^c\right| \right) \sqrt{2\log m} \\
&\quad + 8c_1\epsilon_a \left( \left|\mathcal{H}_t^c\right| + m\Delta_{np}^t \right) \sqrt{2\log m} \\
&\leq 8c_1\epsilon_a \left( 2m\Delta_{np}^t + 3\left|\mathcal{H}_t^c\right| \right) \sqrt{2\log m}
\end{aligned}
$$

Using Lemma C.1, with at least $1 - \frac{1}{c_1} - \exp\left( -\frac{64(c_2-1)^2\eta^2 m^2 \bar{\Delta}^2 t^2}{\pi} \right)$ probability, we get

$$
\left\| \nabla_\theta \hat{L}(f'_t, x) - \nabla_\theta \hat{L}(g'_t, x) \right\|_1 \leq \frac{192\eta m^{1.5}\bar{\Delta}c_1 c_2 \epsilon_a t\sqrt{\log m}}{\sqrt{\pi}} + 16c_1\epsilon_a m\Delta_{np}^t \sqrt{2\log m}
$$

Define $\Gamma$ as the upper bound on $\left\| \nabla_\theta \hat{L}(f'_t, x) - \nabla_\theta \hat{L}(g'_t, x) \right\|_1$.

$$
\Gamma = \frac{192\eta m^{1.5}\bar{\Delta}c_1 c_2 \epsilon_a t\sqrt{\log m}}{\sqrt{\pi}} + 16c_1\epsilon_a m\Delta_{np}^t \sqrt{2\log m} \qquad\qquad (\mathrm{C.17})
$$

$\square$

## D  OPTIMIZATION PROOF

This section shows that gradient-based optimization of the loss for the target function can be closely approximated by the gradient-based optimization of the pseudo network. Since the loss function of the pseudo network is convex in its parameters, we get global optimization.

**Lemma D.1.** *(Convexity of loss function of pseudo network) The loss function for pseudo network is convex with respect to parameters of neural network.*

*Proof.* The loss function for pseudo network is

$$
\hat{L}(g'_t, x) = \sum_{i=1}^{Q} \Delta_x g'_t(\tau_i(x)) - \log\left(g'_t(x)\right)
$$

Dividing the loss function in 2 parts,

$$
\hat{L}(g'_t, x) = \hat{L}_1(g'_t, x) + \hat{L}_2(g'_t, x)
$$

where

$$\hat{L}_1(g'_t, x) = \sum_{i=1}^{Q} \Delta_x g'_t(\tau_i(x))$$

$$\hat{L}_2(g'_t, x) = -\log(g'_t(x))$$

We will prove convexity of both $\hat{L}_1(g'_t, x)$ and $\hat{L}_2(g'_t, x)$. To prove convexity of $\hat{L}_1(g'_t, x)$ as a function of parameters $\theta$, we will prove that Hessian of $\hat{L}_1(g'_t, x)$ is positive semidefinite.

$$\nabla_\theta \hat{L}_1(g'_t, x) = \sum_{i=1}^{Q} \Delta_x \nabla_\theta g'_t(\tau_i(x)) = \sum_{i=1}^{Q} \Delta_x \phi'\left(P_t(\tau_i(x))\right) \nabla_\theta P_t(\tau_i(x))$$

$$\nabla_\theta^2 \hat{L}_1(g'_t, x) = \sum_{i=1}^{Q} \Delta_x \nabla_\theta^2 g'_t(\tau_i(x))$$

$$= \sum_{i=1}^{Q} \Delta_x \phi''(P_t(\tau_i(x))) \nabla_\theta P_t(\tau_i(x)) \nabla_\theta P_t(\tau_i(x))^T$$

$$+ \sum_{i=1}^{Q} \Delta_x \phi'(P_t(\tau_i(x))) \nabla_\theta^2 P_t(\tau_i(x))$$

$$= \sum_{i=1}^{Q} \Delta_x \phi''(P_t(\tau_i(x))) \nabla_\theta P_t(\tau_i(x)) \nabla_\theta P_t(\tau_i(x))^T$$

The first term of the Hessian matrix is sum of Gram matrix and the second term of the Hessian matrix is Gram matrix. Hence, the Hessian of $\hat{L}_1(g'_t, x)$ is positive semidefinite. For second term,

$$\hat{L}_2(g'_t, x) = -\log\left(\exp\left(P_t(x)\right) \mathbb{I}\left[P_t(x) \le 0\right] + (P_t(x) + 1)\mathbb{I}\left[P_t(x) > 0\right]\right)$$
$$= -P_t(x)\mathbb{I}\left[P_t(x) \le 0\right] - \log\left(P_t(x) + 1\right)\mathbb{I}\left[P_t(x) > 0\right]$$

Note that $\hat{L}_2(g'_t, x)$ is convex in $P_t(x)$ and $P_t(x)$ is linear in $\theta$. Composition of convex and linear function is convex therefore, $\hat{L}_2(g'_t, x)$ is convex in $\theta$. As sum of 2 convex functions is convex, $\hat{L}(g'_t, x)$ is convex. $\qquad\square$

**Remark D.1.** *If we use base distribution as standard Gaussian distribution, then loss function will have following term.*

$$\hat{L}_1(g'_t, x) = \left(\sum_{i=1}^{Q} \Delta_x g'_t(\tau_i(x))\right)^2$$

*If we find Hessian of $\hat{L}_1$, then we get*

$$\nabla_\theta \hat{L}_1(g'_t, x) = \left(\sum_{i=1}^{Q} \Delta_x g'_t(\tau_i(x))\right)\left(\sum_{i=1}^{Q} \Delta_x \phi'\left(P_t(\tau_i(x))\right)\nabla_\theta P_t(\tau_i(x))\right)$$

$$\nabla_\theta^2 \hat{L}_1(g'_t, x) = \left(\sum_{i=1}^{Q} \Delta_x g'_t(\tau_i(x))\right)\left(\sum_{i=1}^{Q} \Delta_x \phi''(P_t(\tau_i(x)))\nabla_\theta P_t(\tau_i(x))\nabla_\theta P_t(\tau_i(x))^T\right)$$

$$+ \left(\sum_{i=1}^{Q} \Delta_x \nabla_\theta g'_t(\tau_i(x))\right)\left(\sum_{i=1}^{Q} \Delta_x \nabla_\theta g'_t(\tau_i(x))\right)^T$$

*If base distribution is standard Gaussian distribution, then $'_t$ has to be negative for some points therefore the first term in the Hessian won't remain positive semi-definite therefore, the loss function won't remain convex in parameters of neural network $\theta$ if we use standard Gaussian distribution as base distribution. At points with negative values of $\tilde{g}$, Hessian of $\hat{L}_1(g'_t, x)$ with respect to $\theta$ can be negative semidefinie and $\hat{L}_1(g'_t, x)$ can be non-convex in $\theta$.*

**Lemma D.2.** *(Approximated loss is close to optimal loss) For every $\epsilon \in (0,1)$, there exist $m > poly\left(U_h, \frac{1}{\epsilon}\right), \eta = \tilde{O}\left(\frac{1}{m\epsilon}\right)$ and $T = O\left(\frac{U_h^2 \log m}{\epsilon^2}\right)$ such that, with at least $0.95 - \exp\left(-\frac{\epsilon^2 m}{128 c_1^2 U_h^2 \log m}\right)$ probability, we get*

$$\frac{1}{T}\sum_{t=0}^{T-1} \mathbb{E}_{sgd}[\hat{L}(f'_t, \mathcal{X})] - \hat{L}(F^{*\prime}, \mathcal{X}) \leq O(\epsilon')$$

*Proof.* For set of examples $\mathcal{X}$, define

$$\hat{L}(f'_t, \mathcal{X}) = \frac{1}{|\mathcal{X}|}\sum_{x\in\mathcal{X}}\hat{L}(f'_t, x)$$

From Lemma D.1, we know that $\hat{L}(g'_t, \mathcal{X})$ is convex in parameters $\theta$. Using convexity of $\hat{L}(g'_t, \mathcal{X})$ wrt $\theta$,

$$\begin{aligned}
\hat{L}(g'_t, \mathcal{X}) - \hat{L}(g^{*\prime}, \mathcal{X}) &\leq \langle \nabla_\theta \hat{L}(g'_t, \mathcal{X}), \theta^t - \theta^* \rangle \\
&\leq \|\nabla_\theta \hat{L}(g'_t, \mathcal{X}) - \nabla_\theta \hat{L}(f'_t, \mathcal{X})\|_1 \|\theta^t - \theta^*\|_\infty \\
&\quad + \langle \nabla_\theta \hat{L}(f'_t, \mathcal{X}), \theta^t - \theta^* \rangle
\end{aligned} \tag{D.1}$$

where $\|.\|_1$ and $\|.\|_\infty$ denotes $l_1$ and $l_\infty$ norm respectively. The stochastic gradient descent updates the parameters using $x^t$ at time $t$. $g^{*\prime}$ is defined as following.

$$g^{*\prime}(x) = \sum_{r=1}^m a_{r0}\sigma(w_{r0}x + b_{r0}) + \sum_{r=1}^m a_{r0}\sigma(w_{r0}x + b_{r0})(w_r^* x + b_r^*)$$

For stochastic gradient descent, we get

$$\begin{aligned}
\|\theta^{t+1} - \theta^*\|_2^2 &= \|\theta^t - \eta\nabla_\theta\hat{L}(f'_t, x^t) - \theta^*\|_2^2 \\
&= \|\theta^t - \theta^*\|_2^2 + \eta^2\|\nabla_\theta\hat{L}(f'_t, x^t)\|_2^2 - 2\eta\langle\theta^t - \theta^*, \nabla_\theta\hat{L}(f'_t, x^t)\rangle
\end{aligned}$$

Taking expectation wrt $x_t$,

$$\mathbb{E}_{x^t}\left[\|\theta^{t+1} - \theta^*\|_2^2\right] = \|\theta^t - \theta^*\|_2^2 + \eta^2\mathbb{E}_{x^t}\left[\|\nabla_\theta\hat{L}(f'_t, x^t)\|_2^2\right] - 2\eta\langle\nabla_\theta\hat{L}(f'_t, \mathcal{X}), \theta^t - \theta^*\rangle \tag{D.2}$$

Using Eq.(D.2) and Eq.(D.1),

$$\begin{aligned}
\hat{L}(g'_t, \mathcal{X}) - \hat{L}(g'^*, \mathcal{X}) &\leq \left\|\nabla_\theta\hat{L}(g'_t, \mathcal{X}) - \nabla_\theta\hat{L}(f'_t, x)\right\|_1 \|\theta^t - \theta^*\|_\infty \\
&\quad + \frac{\|\theta^t - \theta^*\|_2^2 - \mathbb{E}_{x^t}\left[\|\theta^{t+1} - \theta^*\|_2^2\right]}{2\eta} \\
&\quad + \frac{\eta}{2}\mathbb{E}_{x^t}\left[\|\nabla_\theta\hat{L}(f'_t, x^t)\|_2^2\right]
\end{aligned}$$

Using Eq.(C.4) and Eq.(C.5), we get

$$\left\|\nabla_\theta\hat{L}(f'_t, x^t)\right\|_2^2 \leq 2m\bar{\Delta}^2$$

Averaging from $t = 0$ to $T - 1$, we get

$$\begin{aligned}
\frac{1}{T}\sum_{t=0}^{T-1}\mathbb{E}_{sgd}[\hat{L}(g'_t, \mathcal{X})] - \hat{L}(g'^*, \mathcal{X}) &\leq \Gamma\left(\sup_{t\in[T]}\|\theta_t\|_\infty + \|\theta^*\|_\infty\right) + \frac{\|\theta^0 - \theta^*\|_2^2}{2\eta T} + \eta m\bar{\Delta}^2 \\
&\overset{(i)}{=} \Gamma\left(\sup_{t\in[T]}\|\theta_t\|_\infty + \|\theta^*\|_\infty\right) + \frac{\|\theta^*\|_2^2}{2\eta T} + \eta m\bar{\Delta}^2 \tag{D.3}
\end{aligned}$$

Note that $\bar{\Delta}$ and $\Gamma$ are defined in Eq.(C.3) and Eq.(C.17). Inequality (i) follows from the fact that $\theta^0 = (0, 0, \ldots, 0) \in \mathbb{R}^{2m}$. Using Lemma B.3 and Lemma C.3, with at least $0.99 - \frac{1}{c_1} - $

$\sum_{t=1}^{T} \exp\left(-\frac{64(c_2-1)^2 \eta^2 m^2 \bar{\Delta}^2 t^2}{\pi}\right) - \frac{1}{c_6} - \frac{1}{c_7} - \exp\left(-\frac{\epsilon^2 m}{128 c_1^2 U_h^2 \log m}\right)$ probability, we get

$$\frac{1}{T}\sum_{t=0}^{T-1} \mathbb{E}_{sgd}[\hat{L}(g_t', \mathcal{X})] - \hat{L}(g'^*, \mathcal{X}) \leq \Gamma\left(\sup_{t\in[T]} \|\theta_t\|_\infty + \|\theta^*\|_\infty\right) + \frac{\|\theta^*\|_2^2}{2\eta T} + \eta m \bar{\Delta}^2$$

$$\frac{1}{T}\sum_{t=0}^{T-1} \mathbb{E}_{sgd}[\hat{L}(f_t', \mathcal{X})] - \hat{L}(g'^*, \mathcal{X}) \overset{(i)}{\leq} \Gamma\left(\sup_{t\in[T]} \|\theta_t\|_\infty + \|\theta^*\|_\infty\right) + \frac{\|\theta^*\|_2^2}{2\eta T} + \eta m \bar{\Delta}^2 + 3\Delta_{np}^t$$

$$\frac{1}{T}\sum_{t=0}^{T-1} \mathbb{E}_{sgd}[\hat{L}(f_t', \mathcal{X})] - \hat{L}(F^{*\prime}, \mathcal{X}) \overset{(ii)}{\leq} \Gamma\left(\sup_{t\in[T]} \|\theta_t\|_\infty + \|\theta^*\|_\infty\right) + \frac{\|\theta^*\|_2^2}{2\eta T} + \eta m \bar{\Delta}^2$$
$$+ 3\Delta_{np}^t + 3\left(16 c_1 (c_6 + c_7) \epsilon_a \log m + \omega_{\phi^{-1}(F^{*\prime})}(\delta) + \epsilon\right)$$

Note that inequality (i) and inequality (ii) uses Lemma B.3 and Lemma C.3 respectively. We choose following values/relations of $\eta, T$.

$$
\begin{aligned}
\eta &= \frac{\epsilon}{m\bar{\Delta}^2} \\
&= \frac{\epsilon}{m\left(6 c_1 \epsilon_a \sqrt{2\log m}\right)^2} \\
&= \frac{\epsilon}{72 c_1^2 m \epsilon_a^2 \log m} \\
T &= \frac{\|\theta^*\|_2^2}{2\eta\epsilon} \\
&= \frac{U_h^2 \pi}{2m\epsilon_a^2} \frac{72 c_1^2 m \epsilon_a^2 \log m}{2\epsilon^2} \\
&= \frac{18\pi c_1^2 \log m \, U_h^2}{\epsilon^2}
\end{aligned}
\tag{D.4}
$$

We can choose $\delta$ such that $\omega_{\phi^{-1}(F^{*\prime})}(\delta) = \epsilon$. Using above inequalities, we get following equalities.

$$\frac{\|\theta^*\|_2^2}{2\eta T} = \frac{\|\theta^*\|_2^2}{2\eta}\frac{2\eta\epsilon}{\|\theta^*\|_2^2} = \epsilon$$
$$\eta m \bar{\Delta}^2 = \frac{\epsilon}{m\bar{\Delta}^2}m\bar{\Delta}^2 = \epsilon$$
$$3\left(16 c_1 (c_6 + c_7) \epsilon_a \log m + \omega_{\phi^{-1}(F^{*\prime})}(\delta) + \epsilon\right) \leq 3\left(16 c_1 (c_6 + c_7) \epsilon_a \log m + 2\epsilon\right)$$

Using Corollary B.1, we get

$$\|\theta^*\|_\infty \leq \frac{U_h\sqrt{\pi}}{\sqrt{2}m\epsilon_a}$$
$$\|\theta^*\|_2 \leq \sqrt{m}\|\theta^*\|_\infty \leq \frac{U_h\sqrt{\pi}}{\sqrt{2}\sqrt{m}\epsilon_a}$$

To get value of $m$,

$$
\sup_{t\in[T]} \|\theta^t\|_\infty = \sup_{t\in[T]} \eta\bar{\Delta}t = \eta\bar{\Delta}T = \frac{\|\theta^*\|_2^2 \bar{\Delta}}{2\epsilon} \leq \frac{U_h^2 \pi}{2m\epsilon_a^2} \frac{\left(6c_1\epsilon_a\sqrt{2\log m}\right)}{2\epsilon}
$$

$$
= \frac{3\pi U_h^2 c_1 \sqrt{\log m}}{\sqrt{2}m\epsilon_a\epsilon}
$$

$$
\sup_{t\in[T]} \|\theta^t\|_\infty + \|\theta^*\|_\infty \leq \frac{\pi U_h^2 \left(1 + 3c_1\right) \sqrt{\log m}}{\sqrt{2}m\epsilon_a\epsilon}
$$

$$
\Gamma = \frac{192\eta m^{1.5}\bar{\Delta}c_1 c_2\epsilon_a t\sqrt{\log m}}{\sqrt{\pi}} + 16c_1\epsilon_a m\Delta_{np}^t\sqrt{2\log m}
$$

$$
\leq \frac{192\eta m^{1.5}\bar{\Delta}c_1 c_2\epsilon_a t\sqrt{\log m}}{\sqrt{\pi}} + 16c_1\epsilon_a m\sqrt{2\log m}\left(\frac{192\eta^2 m^{1.5}\bar{\Delta}^2 c_1 c_2\epsilon_a t^2\sqrt{\log m}}{\sqrt{\pi}}\right)
$$

$$
\leq \frac{192\eta m^{1.5}\bar{\Delta}c_1 c_2\epsilon_a t\sqrt{\log m}}{\sqrt{\pi}} + \frac{3072\sqrt{2}c_1^2 c_2\epsilon_a^2 \eta^2 t^2 m^{2.5}\log m\bar{\Delta}^2}{\sqrt{\pi}}
$$

$$
\leq \frac{192 m^{1.5} c_1 c_2\epsilon_a \sqrt{\log m}}{\sqrt{\pi}}\left(\frac{U_h^2 \pi}{4m\epsilon_a^2\epsilon}\right)\left(6c_1\epsilon_a\sqrt{2\log m}\right)
$$

$$
+ \frac{3072\sqrt{2}c_1^2 c_2\epsilon_a^2 m^{2.5}\log m}{\sqrt{\pi}}\left(\frac{U_h^2 \pi}{4m\epsilon_a^2\epsilon}\right)^2\left(6c_1\epsilon_a\sqrt{2\log m}\right)^2
$$

$$
\leq \frac{288\sqrt{2\pi}\sqrt{m}\log m c_1^2 c_2 U_h^2}{\epsilon} + \frac{13824\sqrt{2\pi^3}c_1^4 c_2\sqrt{m}\left(\log m\right)^2 U_h^4}{\epsilon^2}
$$

$$
\leq \frac{14112\sqrt{2\pi^3}c_1^4 c_2\sqrt{m}\left(\log m\right)^2 U_h^4}{\epsilon^2}
$$

Multiplication of $\Gamma$ and $\left(\sup_{t\in[T]} \|\theta^t\|_\infty + \|\theta^*\|_\infty\right)$ will be

$$
\Gamma\left(\sup_{t\in[T]} \|\theta^t\|_\infty + \|\theta^*\|_\infty\right) \leq \frac{14112\sqrt{2\pi^3}c_1^4 c_2\sqrt{m}\left(\log m\right)^2 U_h^4}{\epsilon^2}\left(\frac{\pi U_h^2 \left(1 + 3c_1\right)\sqrt{\log m}}{\sqrt{2}m\epsilon_a\epsilon}\right)
$$

$$
= \frac{14112\pi^{2.5}c_1^4 c_2 \left(1 + 3c_1\right)\left(\log m\right)^{2.5} U_h^6}{\sqrt{m}\epsilon_a\epsilon^3}
$$

Taking $m$ as

$$
m \geq \Omega\left(\frac{c_1^8 c_2^2 \left(1 + 3c_1\right)^2 U_h^{12}}{\epsilon_a^2\epsilon^8}\right) \tag{D.5}
$$

Choosing $m$ which satisfies above inequality will give us the following inequality.

$$
\Gamma\left(\sup_{t\in[T]} \|\theta^t\|_\infty + \|\theta^*\|_\infty\right) \leq \epsilon
$$

Using Eq.(C.11), we get

$$
\Delta_{np}^t = \left(\frac{192\eta^2 m^{1.5}\bar{\Delta}^2 c_1 c_2\epsilon_a t^2\sqrt{\log m}}{\sqrt{\pi}}\right)
$$

$$
\leq \left(\frac{192 m^{1.5} c_1 c_2\epsilon_a \sqrt{\log m}}{\sqrt{\pi}}\right)\left(\frac{U_h^2 \pi}{4m\epsilon_a^2\epsilon}\right)^2\left(6c_1\epsilon_a\sqrt{2\log m}\right)^2
$$

$$
= \left(\frac{864\pi^{1.5}c_1^3 c_2 \left(\log m\right)^{1.5} U_h^4}{\sqrt{m}\epsilon^2\epsilon_a}\right) \tag{D.6}
$$

Using sufficiently high $m$, we get

$$\Delta_{np}^t \leq \left( \frac{864\pi^{1.5}c_1^3c_2\left(\log m\right)^{1.5}U_h^4}{\sqrt{m}\epsilon^2\epsilon_a} \right)$$

$$\leq O\left( \frac{c_1^3c_2\left(\log m\right)^{1.5}U_h^4\epsilon_a\epsilon^4}{\epsilon^2\epsilon_a c_1^4 c_2\left(1+3c_1\right)U_h^6} \right)$$

$$= O\left( \frac{\epsilon^2\left(\log m\right)^{1.5}}{c_1\left(1+3c_1\right)U_h^2} \right)$$

$$\leq O\left(\epsilon\right)$$

Using Eq.(D.4) and Eq.(D.5), with at least $0.99 - \frac{1}{c_1} - \frac{1}{c_6} - \frac{1}{c_7} - \sum_{t=1}^{T}\exp\left(-\frac{64(c_2-1)^2\eta^2m^2\bar{\Delta}^2t^2}{\pi}\right) - \exp\left(-\frac{\epsilon^2 m}{128c_1^2U_h^2\log m}\right)$ probability, we get

$$\frac{1}{T}\sum_{t=0}^{T-1}\mathbb{E}_{sgd}[\hat{L}(f_t',\mathcal{X})] - \hat{L}(F^{*\prime},\mathcal{X}) \leq \Gamma\left( \sup_{t\in[T]}\|\theta_t\|_\infty + \|\theta^*\|_\infty \right) + \frac{\|\theta^*\|_2^2}{2\eta T} + \eta m\bar{\Delta}^2$$

$$+ \frac{2592\pi^{1.5}c_1^3c_2\left(\log m\right)^{1.5}U_h^4}{\sqrt{m}\epsilon^2\epsilon_a} + 3\left(16c_1\left(c_6+c_7\right)\epsilon_a\log m + 2\epsilon\right)$$

$$\leq 3\epsilon + \frac{2592\pi^{1.5}c_1^3c_2\left(\log m\right)^{1.5}U_h^4}{\sqrt{m}\epsilon^2\epsilon_a} + 3\left(16c_1\left(c_6+c_7\right)\epsilon_a\log m + 2\epsilon\right)$$

Taking $c_1 = 100, c_2 = 2, c_5 = 1000, c_6 = 100, c_7 = 100, \epsilon_a = \frac{\epsilon}{6000\log m} \leq \epsilon$, with at least $0.95 - \sum_{t=1}^{T}\exp\left(-\frac{64\eta^2m^2\bar{\Delta}^2t^2}{\pi}\right) - \exp\left(-\frac{\epsilon^2 m}{128c_1^2U_h^2\log m}\right)$ probability,

$$\frac{1}{T}\sum_{t=0}^{T-1}\mathbb{E}_{sgd}[\hat{L}(f_t',\mathcal{X})] - \hat{L}(F^{*\prime},\mathcal{X}) \leq 3\epsilon + \frac{2592\pi^{1.5}c_1^3c_2\left(\log m\right)^{1.5}U_h^4}{\sqrt{m}\epsilon^2\epsilon_a} + O\left(\epsilon\right)$$

$$\leq 3\epsilon + O\left(\frac{\epsilon^2}{U_h^2}\right) + O\left(\epsilon\right)$$

For any $\epsilon \in [0,1]$, with probability at least $0.96 - \sum_{t=1}^{T}\exp\left(-\frac{64\eta^2m^2\bar{\Delta}^2t^2}{\pi}\right) - \exp\left(-\frac{\epsilon^2 m}{128c_1^2U_h^2\log m}\right)$ probability,

$$\frac{1}{T}\sum_{t=0}^{T-1}\mathbb{E}_{sgd}[\hat{L}(f_t',\mathcal{X})] - \hat{L}(F^{*\prime},\mathcal{X}) \leq O(\epsilon)$$

To find lower bound on probability, we use $\sum_{t=1}^{T}\frac{1}{t^2} \leq \sum_{t=1}^{\infty}\frac{1}{t^2} \leq 2$.

$$\sum_{t=1}^{T}\exp\left(-\frac{64\eta^2m^2\bar{\Delta}^2t^2}{\pi}\right) \overset{(i)}{\leq} \sum_{t=1}^{T}\frac{\pi}{64\eta^2m^2\bar{\Delta}^2t^2}$$

$$\leq \frac{\pi\left(m\bar{\Delta}^2\right)^2}{32\epsilon^2m^2\bar{\Delta}^2}$$

$$\leq \frac{\pi\bar{\Delta}^2}{32\epsilon^2}$$

$$\leq \frac{\pi}{3200}$$

$$\leq 0.01$$

where inequality (i) follows from $\exp(-x) \leq \frac{1}{x}$ for all $x \geq 0$. Finally, with at least $0.95 - \exp\left(-\frac{\epsilon^2 m}{128 c_1^2 U_h^2 \log m}\right)$ probability,

$$\frac{1}{T} \sum_{t=0}^{T-1} \mathbb{E}_{sgd}[\hat{L}(f_t', \mathcal{X})] - \hat{L}(F^{*\prime}, \mathcal{X}) \leq O(\epsilon)$$

□

## E  GENERALIZATION

In this section, we prove generalization guarantees to complement our optimization result, and complete the proof of our main theorem (Theorem E.1) about efficiently learning distributions using univariate normalizing flows. The proof in this section can be divided broadly in two parts. First, we prove that empirical average of $\hat{L}(f_t', x)$ and $\hat{L}(F^{*\prime}, x)$ on training examples are close to expectation of $\hat{L}(f_t', x)$ and $\hat{L}(F^{*\prime}, x)$ with respect to underlying data distribution, respectively. The similar argument is also used in Allen-Zhu et al. (2019). Second, we prove that $\hat{L}(f_t', x)$ and $\hat{L}(F^{*\prime}, x)$ are close to $L(f_t, x)$ and $L(F^*, x)$, respectively.

Recall that the approximate loss function $\hat{L}$ is given by

$$\hat{L}(f_t', x) = \sum_{i=1}^{Q} \Delta_x \phi(N_t(\tau_i(x))) - \log(\phi(N_t(x)))$$

where

$$N_t(x) = \sum_{r=1}^{m} a_{r0} \sigma\left((w_{r0} + w_r^t)x + b_{r0} + b_r^t\right)$$

**Lemma E.1.** *(Empirical Rademacher complexity for two-layer neural network) For every $B > 0$, for every $n \geq 1$, with at least $1 - \frac{1}{c_1}$ probability over random initialization, the empirical Rademacher complexity is bounded by*

$$\frac{1}{n} \mathbb{E}_{\xi \in \{\pm 1\}^n} \left[ \sup_{\max_{r \in [m]} |w_r|, |b_r| \leq B} \sum_{i=1}^{n} \xi_i N(x_i) \right] \leq \frac{8 c_1 \epsilon_a B m \sqrt{2 \log m}}{\sqrt{n}}$$

*Proof.* Using part a of Lemma H.5, we get that $\{x \mapsto w_r x + b_r \mid |w_r| \leq B, |b_r| \leq B\}$ has Rademacher complexity $\frac{2B}{\sqrt{n}}$. Using part b of Lemma H.5, we get that $\{x \mapsto ((w_{r0} + w_r)x + (b_{r0} + b_r)) \mid |w_r| \leq B, |b_r| \leq B, w_{r0}, b_{r0} \sim \mathcal{N}(0, \frac{1}{m})\}$ has Rademacher complexity $\frac{2B}{\sqrt{n}}$. Using part c of Lemma H.5, we get that class of functions in $\mathcal{F} = \{x \mapsto N(x) \mid \max_{r \in [m]} |w_r| \leq B, \max_{r \in [m]} |b_r| \leq B\}$ has Rademacher complexity

$$\hat{\mathcal{R}}(\mathcal{X}; \mathcal{F}) \leq 2\|\mathbf{a}\|_1 \frac{2B}{\sqrt{n}} \overset{(i)}{\leq} \frac{8 c_1 \epsilon_a B m \sqrt{2 \log m}}{\sqrt{n}}$$

where inequality (i) follows from Lemma H.2 with at least $1 - \frac{1}{c_1}$ probability over random initialization.

□

Define upper bound on maximum and lower bound on minimum value of loss function $\hat{L}$ is given by

$$\sup_x \hat{L}(F^{*\prime}, x) = \sup_x \sum_{i=1}^{Q} \Delta_x F^{*\prime}(\tau_i(x)) - \log(F^{*\prime}(x)) \leq 2 M_{F^{*\prime}} - \log(m_{F^{*\prime}}) := M_{\hat{L}} \quad \text{(E.1)}$$

$$\inf_x \hat{L}(F^{*\prime}, x) = \inf_x \sum_{i=1}^{Q} \Delta_x F^{*\prime}(\tau_i(x)) - \log(F^{*\prime}(x)) \geq 2 m_{F^{*\prime}} - \log(M_{F^{*\prime}}) := m_{\hat{L}} \quad \text{(E.2)}$$

**Lemma E.2.** *Suppose $n$ is sufficiently high such that it satisfies following condition.*

$$n \geq O\left(\frac{\left(M_{\hat{L}} - m_{\hat{L}}\right)^2 (Q+1)^2 U_h^4 (\log m)^2}{\epsilon^2}\right)$$

*If $n$ satisfies above condition, then with at least $0.98$ probability over random initialization, population loss of any functions of set $\{x \mapsto N(x) \mid |w_r| \leq \eta\bar{\Delta}T, \ |b_r| \leq \eta\bar{\Delta}T \ \forall r \in [m]\}$ is close to empirical loss i.e.*

$$\sup_{N \in \mathcal{F}} \left| \mathbb{E}_{x \in \mathcal{D}}\left[\hat{L}\left(f_t', x\right)\right] - \frac{1}{n}\sum_{i=1}^{n} \hat{L}\left(f_t', x\right) \right| \leq \epsilon$$

*Proof.* Note that $\hat{L}(f_t', x)$ depends on neural network $N_t(x)$ through $(N_t(\tau_1(x)), N_t(\tau_2(x)), \ldots, N_t(\tau_Q(x)), N_t(x))$ vector. Using Fact H.8, with at least $1 - \delta$ probability, we get

$$\sup_{N \in \mathcal{F}} \left| \mathbb{E}_{x \sim \mathcal{D}}\left[\hat{L}\left(f_t', x\right)\right] - \frac{1}{n}\sum_{i=1}^{n} \hat{L}\left(f_t', x\right) \right| \leq 2\sqrt{2}L_s(Q+1)\hat{\mathcal{R}}(\mathcal{X};\mathcal{F}) + b\sqrt{\frac{\log\frac{1}{\delta}}{2n}} \quad \text{(E.3)}$$

where $\mathcal{F} = \{x \mapsto N(x) \mid |w_r| \leq \eta\bar{\Delta}T, \ |b_r| \leq \eta\bar{\Delta}T \ \forall r \in [m]\}$. We get coordinate wise Lipschitz continuity of loss $\hat{L}$ function as following.

$$L_j \leq \sup_{N \in \mathcal{F}, |x| \leq 1} |\Delta_x \phi'(N(\tau_j(x)))|$$

$$\leq \sup_{N \in \mathcal{F}, |x| \leq 1} \frac{1}{Q}|\phi'(N(\tau_j(x)))|$$

$$\leq \frac{2}{Q} \quad \forall i \in [Q]$$

$$L_{Q+1} \leq \sup_{N \in \mathcal{F}, |x| \leq 1} \frac{\phi'(N(x))}{\phi(N(x))} = \sup_{N \in \mathcal{F}, |x| \leq 1} \frac{\exp(N(x))\,\mathbb{I}[N(x) \leq 0] + \mathbb{I}[N_t(x) \geq 0]}{\exp(N(x))\,\mathbb{I}[N(x) \leq 0] + (N(x)+1)\,\mathbb{I}[N(x) \geq 0]}$$

$$\leq \sup_{N \in \mathcal{F}, |x| \leq 1} \mathbb{I}[N(x) \leq 0] + \frac{1}{N(x)+1}\mathbb{I}[N(x) \geq 0]$$

$$\leq 1$$

Using Lemma H.4, standard Lipschitz constant of $\hat{L}$ is

$$L_s \leq \sqrt{\sum_{i=1}^{Q+1} L_i^2} \leq \sqrt{\frac{4}{Q}+1} \leq 2 \quad \text{(E.4)}$$

To get upper bound on $\hat{L}$, we use Lipschitz property of $\hat{L}$.

$$\left| \hat{L}(f_t', x) - \hat{L}\left(\tilde{f}_t', x\right) \right| \leq \sum_{i=1}^{Q} \Delta_x |N(\tau_i(x))| + |N(x)| \quad \text{(E.5)}$$

Note that $\hat{L}(f_t', x)$ depends upon $(N(\tau_1(x)), N(\tau_2(x)), \ldots, N(\tau_Q(x)), N(x))$ vector and similarly, $\hat{L}(\tilde{f}_t', x)$ depends upon $(0, 0, 0, \ldots, 0, 0)$ Finding upper bound $N(x)$ for all $x \in [-1, 1]$,

$$\sup_{N \in \mathcal{F}, x \in [-1,1]} N(x) \leq \sup_{|w_r| \leq \eta \bar{\Delta} T, |b_r| \leq \eta \bar{\Delta} T, x \in [-1,1]} P_t(x) + \Delta_{np}^T$$

$$\leq \sup_{|w_r| \leq \eta \bar{\Delta} T, |b_r| \leq \eta \bar{\Delta} T, x \in [-1,1]} \sum_{r=1}^{m} a_{r0} \sigma(w_{r0} x + b_{r0}) + \sum_{r=1}^{m} a_{r0} (w_r x + b_r) \sigma(w_{r0} x + b_{r0}) + \Delta_{np}^T$$

$$\overset{(i)}{\leq} 16 c_1 (c_6 + c_7) \epsilon_a \log m + m \left(2 c_1 \epsilon_a \sqrt{2 \log m}\right) \left(2 \eta \bar{\Delta} T\right) + \Delta_{np}^T$$

$$\leq 16 c_1 (c_6 + c_7) \epsilon_a \log m + m \left(2 c_1 \epsilon_a \sqrt{2 \log m}\right) \left(2 \eta \bar{\Delta} T\right) + \Delta_{np}^T$$

$$\overset{(ii)}{\leq} 16 c_1 (c_6 + c_7) \epsilon_a \log m + m \left(48 c_1^2 \epsilon_a^2 \log m \left(\frac{U_h^2 \pi}{4 m \epsilon_a^2 \epsilon}\right)\right) + \Delta_{np}^T$$

$$\leq O\left(\frac{U_h^2 \log m}{\epsilon}\right)$$

where inequality (i) uses Eq. (B.9), Lemma H.2 and Eq.(C.6). The inequality (ii) uses our choices of $\eta$ and $T$ from Eq.(D.4) and lower bound on $m$ from Eq.(D.5). Define $K$ as upper bound on $\sup_{N \in \mathcal{F}, x \in [-1,1]} N(x)$.

$$K := O\left(\frac{U_h^2 \log m}{\epsilon}\right) \tag{E.6}$$

Using upper bound on $\sup_{N \in \mathcal{F}, x \in [-1,1]} N(x)$ and Eq.(E.5), we get upper bound on $\hat{L}(b)$.

$$b = K + K + \hat{L}(0, 0, ..., 0, 0) \leq 2K + 2$$

Using value of $b$ in Eq.(E.3) and Lemma E.1, with at least $1 - \delta - \frac{1}{c_1}$ probability, we get

$$\sup_{N \in \mathcal{F}} \left| \mathbb{E}_{x \in \mathcal{D}} \left[ \hat{L}(f_t', x) \right] - \frac{1}{n} \sum_{i=1}^{n} \hat{L}(f_t', x_i) \right|$$

$$\leq 4\sqrt{2}(Q + 1) \frac{8 c_1 \epsilon_a \eta \bar{\Delta} T m \sqrt{2 \log m}}{\sqrt{n}} + (2K + 2) \sqrt{\frac{\log \frac{1}{\delta}}{2n}}$$

We use $\delta = 0.01$ and choose $n$ which satisfies following condition.

$$n \geq O\left(\frac{(M_{\hat{L}} - m_{\hat{L}})^2 (Q + 1)^2 U_h^4 (\log m)^2}{\epsilon^2}\right) \tag{E.7}$$

Using above $n$, with at least $0.98$ probability, we get

$$\sup_{N \in \mathcal{F}} \left| \mathbb{E}_{x \in \mathcal{D}} \left[ \hat{L}(f_t', x) \right] - \frac{1}{n} \sum_{i=1}^{n} \hat{L}(f_t', x_i) \right| \leq \epsilon$$

$\square$

**Lemma E.3.** *(Concentration on approximated loss of target function) Suppose $n$ is sufficiently high such that it satisfies following condition.*

$$n \geq O\left(\frac{(M_{\hat{L}} - m_{\hat{L}})^2 (Q + 1)^2 U_h^4 (\log m)^2}{\epsilon^2}\right)$$

*If $n$ satisfies above condition, then with at least $0.9999$ probability, population loss of target function $F^{*'}$ is close to empirical loss i.e.*

$$\left| \mathbb{E}_{x \sim \mathcal{D}} \left[ \hat{L}(F^{*'}, x) \right] - \hat{L}(F^{*'}, \mathcal{X}) \right| \leq \epsilon$$

*Proof.* Finding minimum value $(m_{\hat{L}})$ and maximum value $(M_{\hat{L}})$ of loss function $\hat{L}$,

$$\sup_x \hat{L}\left(F^{*\prime}, x\right) = \sup_x \sum_{i=1}^{Q} \Delta_x F^{*\prime}\left(\tau_i\left(x\right)\right) - \log(F^{*\prime}(x)) \leq 2M_{F^{*\prime}} - \log\left(m_{F^{*\prime}}\right) = M_{\hat{L}}$$

$$\inf_x \hat{L}\left(F^{*\prime}, x\right) = \inf_x \sum_{i=1}^{Q} \Delta_x F^{*\prime}\left(\tau_i\left(x\right)\right) - \log(F^{*\prime}(x)) \leq 2m_{F^{*\prime}} - \log\left(M_{F^{*\prime}}\right) = m_{\hat{L}}$$

where $M_{F^{*\prime}} = \max_{x \in [-1,1]} F^{*\prime}(x)$ and $m_{F^{*\prime}} = \max_{x \in [-1,1]} F^{*\prime}(x)$. Using Hoeffding's inequality,

$$\Pr\left(\left|\mathbb{E}_{x \sim \mathcal{D}}\left[\hat{L}\left(F^{*\prime}, \mathcal{X}\right)\right] - \hat{L}\left(F^{*\prime}, \mathcal{X}\right)\right| \geq \epsilon\right) \leq \exp\left(-\frac{2n\epsilon^2}{\left(M_{\hat{L}} - m_{\hat{L}}\right)^2}\right)$$

Taking $n$ as

$$n \geq O\left(\frac{\left(M_{\hat{L}} - m_{\hat{L}}\right)^2 (Q+1)^2 U_h^4 (\log m)^2}{\epsilon^2}\right)$$

With at least probability $1 - \exp\left(-\frac{2n\epsilon^2}{\left(M_{\hat{L}} - m_{\hat{L}}\right)^2}\right)$,

$$\left|\mathbb{E}_{x \sim \mathcal{D}}\left[\hat{L}\left(F^{*\prime}, x\right)\right] - \hat{L}\left(F^{*\prime}, \mathcal{X}\right)\right| \leq \epsilon \tag{E.8}$$

$\square$

**Corollary E.1.** *Under same setting as Lemma D.2 and*

$$n \geq O\left(\frac{\left(M_{\hat{L}} - m_{\hat{L}}\right)^2 (Q+1)^2 U_h^4 (\log m)^2}{\epsilon^2}\right)$$

*then with at least $0.92 - 2\exp\left(-\frac{m}{8\epsilon_a^2}\right) - \exp\left(-\frac{\epsilon^2 m}{128c_1^2 U_h^2 \log m}\right) - \exp\left(-\frac{2n\epsilon^2}{\left(M_{\hat{L}} - m_{\hat{L}}\right)^2}\right)$ probability, we get*

$$\mathbb{E}_{sgd}\left[\frac{1}{T}\sum_{t=0}^{T-1} \mathbb{E}_{x \sim \mathcal{D}}\left[\hat{L}(f_t', x)\right]\right] - \mathbb{E}_{x \sim \mathcal{D}}\left[\hat{L}(F^{*\prime}, x)\right] \leq O(\epsilon)$$

*Proof.* Using Lemma D.2, Lemma E.2 and Lemma E.3, with at least $0.92 - 2\exp\left(-\frac{m}{8\epsilon_a^2}\right) - \exp\left(-\frac{\epsilon^2 m}{128c_1^2 U_h^2 \log m}\right) - \exp\left(-\frac{2n\epsilon^2}{\left(M_{\hat{L}} - m_{\hat{L}}\right)^2}\right)$ probability, we get

$$\mathbb{E}_{sgd}\left[\frac{1}{T}\sum_{t=0}^{T-1} \mathbb{E}_{x \sim \mathcal{D}}\left[\hat{L}(f_t', x)\right]\right] - \mathbb{E}_{x \sim \mathcal{D}}\left[\hat{L}(F^{*\prime}, x)\right] \leq O(\epsilon)$$

$\square$

**Theorem E.1.** *(loss function is close to optimal) For every $\epsilon \in (0,1)$, there exist $m > \text{poly}\left(U_h, \frac{1}{\epsilon}\right)$, $\eta = \tilde{O}\left(\frac{1}{m\epsilon}\right)$ and $T = O\left(\frac{U_h^2 \log m}{\epsilon^2}\right)$, for any target function $F^{*\prime}$ with finite second order derivative and number of quadrature points $Q \geq \frac{4M_{F^{*\prime\prime}} + 4K_2}{\epsilon}$ and number of training points $n \geq O\left(\frac{\left(M_{\hat{L}} - m_{\hat{L}}\right)^2 (Q+1)^2 U_h^4 (\log m)^2}{\epsilon^2}\right)$, with at least $0.92 - 2\exp\left(-\frac{m}{8\epsilon_a^2}\right) - \exp\left(-\frac{\epsilon^2 m}{128c_1^2 U_h^2 \log m}\right) - \exp\left(-\frac{2n\epsilon^2}{\left(M_{\hat{L}} - m_{\hat{L}}\right)^2}\right)$ probability, we have*

$$\mathbb{E}_{sgd}\left[\frac{1}{T}\sum_{t=0}^{T-1} \mathbb{E}_{x \sim \mathcal{D}}\left[L(f_t, x)\right]\right] - \mathbb{E}_{x \sim \mathcal{D}}\left[L(F^*, x)\right] \leq O(\epsilon)$$

*where $U_h$ is the complexity of target function defined in B.1 and*

$$M_{F^{*\prime\prime}} = \sup_{x\in[-1,1]} F^{*\prime\prime}(x)$$

$$M_{F^{*\prime}} = \sup_{x\in[-1,1]} F^{*\prime}(x)$$

$$m_{F^{*\prime}} = \inf_{x\in[-1,1]} F^{*\prime}(x)$$

$$M_{\hat{L}} = 2M_{F^{*\prime}} - \log(m_{F^{*\prime}})$$

$$m_{\hat{L}} = 2m_{F^{*\prime}} - \log(M_{F^{*\prime}})$$

$$K_2 = O\left(\frac{U_h^2 \left(\log m\right)^{1.5}}{\epsilon}\right)$$

*Proof.* First, we will try to bound

$$\left|\hat{L}(F^{*\prime}, x) - L(F^*, x)\right| \leq \left|\sum_{i=1}^{Q} \Delta_x F^{*\prime}\left(\tau_i\left(x\right)\right) - F^*(x)\right|$$

$$\leq \frac{2M_{F^{*\prime\prime}}}{Q}$$

Similarly, bounding error for $f_t'$, we will get

$$\left|\hat{L}(f_t', x) - L(f_t, x)\right| \leq \left|\sum_{i=1}^{Q} \Delta_x f_t'\left(\tau_i\left(x\right)\right) - f_t(x)\right|$$

$$\leq \frac{2\left(\sup_x f_t''(x)\right)}{Q}$$

To get $\sup_x f_t''(x)$, we will use Eq.(E.6).

$$\sup_x f_t''(x) \leq \sup_x |N_t'(x)|$$

$$\leq \sup_x \sum_{r=1}^{m} a_{r0}\sigma'\left((w_{r0} + w_r^t)x + b_{r0} + b_r^t\right)\left(w_{r0} + w_r^t\right)$$

$$\leq \sup_x \sum_{r=1}^{m} a_{r0}\left(w_{r0} + w_r^t\right)\mathbb{I}\left[(w_{r0} + w_r^t)x + b_{r0} + b_r^t \geq 0\right]$$

$$\leq \sup_x \sum_{r\in\mathcal{H}} a_{r0}\left(w_{r0} + w_r^t\right)\mathbb{I}\left[w_{r0}x + b_{r0} \geq 0\right] + \sum_{r\notin\mathcal{H}} a_{r0}\left(w_{r0} + w_r^t\right)\mathbb{I}\left[(w_{r0} + w_r^t)x + b_{r0} + b_r^t \geq 0\right]$$

$$\leq \sup_x \sum_{r\in\mathcal{H}} a_{r0}w_{r0}\mathbb{I}\left[w_{r0}x + b_{r0} \geq 0\right] + \sum_{r\in\mathcal{H}} a_{r0}w_r^t\mathbb{I}\left[w_{r0}x + b_{r0} \geq 0\right]$$

$$+ \sum_{r\notin\mathcal{H}} a_{r0}w_{r0}\mathbb{I}\left[(w_{r0} + w_r^t)x + b_{r0} + b_r^t \geq 0\right] + \sum_{r\notin\mathcal{H}} a_{r0}w_r^t\mathbb{I}\left[(w_{r0} + w_r^t)x + b_{r0} + b_r^t \geq 0\right]$$

$$\overset{(i)}{\leq} 16c_1c_6\epsilon_a\left(\log m\right) + m\left(2c_1\epsilon_a\sqrt{2\log m}\right)\left(\eta\bar{\Delta}T\right)$$

$$+ \left(c_2\frac{4\sqrt{2}\eta m\sqrt{m}\bar{\Delta}t}{\sqrt{\pi}}\right)\left(2c_1\epsilon_a\sqrt{2\log m}\right)\left(\frac{2c_6\sqrt{2\log m}}{\sqrt{m}}\right)$$

$$+ \left(c_2\frac{4\sqrt{2}\eta m\sqrt{m}\bar{\Delta}t}{\sqrt{\pi}}\right)\left(2c_1\epsilon_a\sqrt{2\log m}\right)\left(\eta\bar{\Delta}T\right)$$

$$\leq O\left(\epsilon\right) + m\left(2c_1\epsilon_a\sqrt{2\log m}\right)\left(\bar{\Delta}\frac{U_h^2\pi}{4m\epsilon_a^2\epsilon}\right)$$

$$+ \left(c_2\frac{4\sqrt{2}m\sqrt{m}\bar{\Delta}}{\sqrt{\pi}}\right)\left(2c_1\epsilon_a\sqrt{2\log m}\right)\left(\frac{2c_6\sqrt{2\log m}}{\sqrt{m}}\right)\frac{U_h^2\pi}{4m\epsilon_a^2\epsilon}$$

$$+ \left(c_2\frac{4\sqrt{2}m\sqrt{m}\bar{\Delta}}{\sqrt{\pi}}\right)\left(2c_1\epsilon_a\sqrt{2\log m}\right)\left(\bar{\Delta}\left(\frac{U_h^2\pi}{4m\epsilon_a^2\epsilon}\right)^2\right)$$

$$\leq O\left(\epsilon\right) + O\left(\frac{U_h^2\log m}{\epsilon}\right) + O\left(\frac{U_h^2\left(\log m\right)^{1.5}}{\epsilon}\right) + O\left(\epsilon\right)$$

$$\leq O\left(\frac{U_h^2\left(\log m\right)^{1.5}}{\epsilon}\right)$$

Define $K_2$ as upper bound on $\sup_x f_t''(x)$,

$$K_2 = O\left(\frac{U_h^2\left(\log m\right)^{1.5}}{\epsilon}\right)$$

Taking $Q$ as

$$Q \geq \frac{2M_{F^{*''}} + 2K_2}{\epsilon} \tag{E.9}$$

Using given value of $Q$, we get that

$$\left|\hat{L}(F^{*'}, x) - L(F^*, x)\right| \leq \epsilon \tag{E.10}$$

$$\left|\hat{L}(f_t', x) - L(f_t, x)\right| \leq \epsilon \tag{E.11}$$

Using these relations, we get

$$\mathbb{E}_{sgd}\left[\frac{1}{T}\sum_{t=0}^{T-1}\mathbb{E}_{x\sim\mathcal{D}}\left[L(f_t, x)\right]\right] - \mathbb{E}_{x\sim\mathcal{D}}\left[L(F^*, x)\right] \leq O(\epsilon)$$

By definition of KL divergence, we get

$$\mathbb{E}_{sgd} \left[ \frac{1}{T} \sum_{t=0}^{T-1} \mathrm{KL}\left(p_{F^*,Z} || p_{f_t,Z}\right) \right] \leq O(\epsilon)$$

$\square$

## F    PROBLEM IN TRAINING OF CONSTRAINED NORMALIZING FLOW

In this section, we provide reasons and details why changing initialization will not solve the problem (described in section 2.1 ) in the training of Constrained Normalizing Flow. The neural network in Constrained Normalizing Flow (CNF) is defined as

$$N(x) = \tau \sum_{r=1}^{m} a_{r0} \, \tanh\left(\left(w_{r0} + w_r\right)x + \left(b_r + b_{r0}\right)\right), \text{ with constraints } w_{r0} + w_r \geq \epsilon, \text{ for all } r.$$

Here, $\epsilon > 0$ is a small constant and $\tau$ is a normalization constant which only depends on $m$. The pseudo network for this neural network will be

$$P(x) = \tau \sum_{r=1}^{m} a_{r0} \left(\tanh(w_{r0}x + b_{r0}) + \tanh'(w_{r0}x + b_{r0})\left(w_r x + b_r\right)\right),$$

with constraints $w_{r0} + w_r \geq \epsilon$, for all $r$. We decompose $P(x)$ into two parts: $P(x) = P_c(x) + P_\ell(x)$, where

$$P_c(x) = \tau \sum_{r=1}^{m} a_{r0} \left(\tanh(w_{r0}x + b_{r0})\right) \quad \text{and} \quad P_\ell(x) = \tau \sum_{r=1}^{m} a_{r0} \left(\tanh'(w_{r0}x + b_{r0})\left(w_r x + b_r\right)\right).$$

Note that $P_c(x)$ only depends upon initialization and does not depend on $w_r$ and $b_r$. Hence, it can not approximate the target function after the training, therefore $P_\ell(x)$ needs to approximate target function with $P_c(x)$ subtracted. Note that normalization constant $\tau$ is necessary to keep $P_\ell(x)$ as same order of $F^{*\prime}(x)$ with $P_c(x)$ subtracted. Note that Now, we will show that $P_\ell(x)$ can not approximate "sufficiently non-linear" functions. Using half-normal distribution initialization of $w_{r0}$ with (mean, variance) of normal distribution as $\left(0, \frac{1}{m}\right)$ and normal distribution initialization of $b_{r0}$ with (mean, variance) as $\left(0, \frac{1}{m}\right)$, $w_{r0}$ and $|b_{r0}|$ are $O\left(\frac{\sqrt{\log m}}{\sqrt{m}}\right)$ with high probability; therefore, $|w_{r0}x + b_{r0}|$ is $O\left(\frac{\sqrt{\log m}}{\sqrt{m}}\right)$. Using the fact that $\tanh'(y) \approx 1$ for small $y$, we get that $\tanh'(w_{r0}x + b_{r0}) \approx 1$ for sufficient large $m$. In such cases, $P_\ell(x)$ becomes linear function in $x$ and won't be able to approximate sufficiently non-linear function. Using other variance of initializations leads to problem in other parts of the proof.

**Remark F.1.** *Using different variance in initialization of $w_{r0}$ and $b_{r0}$ will not solve the problem in the training of CNFs (mentioned in section 2.1).*

*Informal Proof:* As described above, if we use variance of initialization of $w_{r0}$ and $b_{r0}$ as $\frac{1}{m}$, then $P_\ell(x)$ cannot approximate "sufficiently non-linear" target function. The same problem will remain if variance of initialization of $w_{r0}$ and $b_{r0}$ is $o\left(\frac{1}{\log m}\right)$. For variance of initialization of $w_{r0}$ and $b_{r0}$ is $\Omega\left(\frac{1}{\log m}\right)$ and $O(1)$, we will prove by contradiction that there doesn't exist any pseudo network with sufficiently small norm on $\|\theta^*\|_2$ for which the pseudo network can approximate the target function and for bigger norm on $\|\theta^*\|_2$, pseudo network $P(x)$ during the training does not stay close to $N(x)$.

$\square$

**Remark F.2.** *There doesn't exist any pseudo network with sufficiently small norm on $\|\theta^*\|_2$ for which the pseudo network can approximate the target function and for bigger norm on $\|\theta^*\|_2$, pseudo network $P(x)$ during the training does not stay close to $N(x)$.*

*Informal Proof:* Suppose there exist $P^*(x)$ function which approximates the target function $F^{*\prime}(x)$. As described earlier, $P_c^*(x)$ only depends upon initialization and does not depend on $w_r$ and $b_r$. Hence,

it can not approximate the target function after the training, therefore $P_\ell^*(x)$ needs to approximate target function with $P_c^*(x)$ subtracted. Hence, $P_c^*(x)$, $P_\ell^*(x)$ and $F^{*\prime}(x)$ should be $\Theta(1)$. From the condition that $P_c^*(x)$ needs to be $O(1)$, we get that $\tau$ will be $o\left(\frac{1}{\sigma_{wb}m}\right)$ for the considered range of variance of $w_{r0}$ and $b_{r0}$ where $\sigma_{wb}$ is variance of $w_{r0}$ and $b_{r0}$. We denote variance of $a_{r0}$ as $\sigma_a$. From the condition that $P_\ell^*(x)$ needs to be $\Theta(1)$, we need

$$\tau \sum_{r=1}^{m} a_{r0} \left(\tanh'(w_{r0}x + b_{r0})(w_r^*x + b_r^*)\right) = \Theta(1)$$

$$\implies \tau m \sigma_a \sqrt{\log m} \|\theta^*\|_\infty = \Theta(1) \implies \|\theta^*\|_\infty = \Theta\left(\frac{1}{\tau m \sigma_a \sqrt{\log m}}\right)$$

Using norm equivalance, we need

$$\|\theta^*\|_2 = \Theta\left(\frac{1}{\tau \sqrt{m} \sigma_a \sqrt{\log m}}\right)$$

Doing similar calculation as in Lemma D.2 for the loss of normalizing flow $L_G$, we will get Eq.(D.3) and from Eq.(D.3), to get small training error, we need

$$\frac{\|\theta^*\|_2^2}{\eta T} = O(\epsilon)$$

$$\implies \eta T = O\left(\frac{\|\theta^*\|_2^2}{\epsilon}\right)$$

$$\implies \eta T = O\left(\frac{1}{\epsilon m \tau^2 \sigma_a^2 \log m}\right) \tag{F.1}$$

Now, to find coupling of function $N(x)$ and $P(x)$ as well as derivative of function $N'(x)$ and $P'(x)$, we find upper bound on the derivative of loss function and $w_r$ and $b_r$.

$$\frac{\partial L_G(f^t, x)}{\partial w_r} = \tau N_t(x)(a_{r0}\sigma'((w_{r0} + w_r^t)x + b_{r0} + b_r^t))$$
$$- \frac{\tau}{N'(x)}(a_{r0}(\sigma'((w_{r0} + w_r^t)x + b_{r0} + b_r^t) + (w_{r0} + w_r^t)x\sigma''((w_{r0} + w_r^t)x + b_{r0} + b_r^t)))$$

We assume that $L_G(f_t, x)$ is $\tilde{L}_1$-lipschitz continuous wrt $N$ and $\tilde{L}_2$-lipschitz continuous wrt $N'$. Assuming $|\sigma'(.)| \leq 1$ and $|x| \leq 1$,

$$\left|\frac{\partial L_G(f_t, x)}{\partial w_r}\right| \leq \tau \tilde{L}_1 a_{r0} + \tau \tilde{L}_2 a_{r0} \left(1 + |w_r^t + w_{r0}||\sigma''((w_{r0} + w_r^t)x + b_{r0} + b_r^t)|\right)$$

Assuming $|\sigma''(.)| \leq C$,

$$\left|\frac{\partial L_G(f_t, x)}{\partial w_r}\right| \leq \tau \tilde{L}_1 a_{r0} + \tau \tilde{L}_2 a_{r0} \left(1 + C(|w_r^t| + w_{r0})\right)$$

Using Lemma H.2 for $a_{r0}$ and $w_{r0}$, with at least $1 - \frac{1}{c_1} - \frac{1}{c_2}$ probability,

$$\left|\frac{\partial L_G(f_t, x)}{\partial w_r}\right| \leq \left(2c_1\sigma_a\tau\sqrt{2\log m}\right)\left(\tilde{L}_1 + \tilde{L}_2\left(1 + C\left(|w_r^t| + 2c_2\sigma_{wb}\sqrt{2\log m}\right)\right)\right) \tag{F.2}$$

For projected gradient descent,

$$|w_r^t| \leq \eta \sum_{i=0}^{t-1} \left|\frac{\partial L_G(f_t, x^i)}{\partial w_r}\right|$$

$$\leq \eta \sum_{i=0}^{t-1} \left(\left(2c_1\sigma_a\tau\sqrt{2\log m}\right)\left(\tilde{L}_1 + \tilde{L}_2 + 2c_2C\sigma_{wb}\tilde{L}_2\sqrt{2\log m}\right) + \left(2c_1\sigma_a\tau\sqrt{2\log m}\right)\tilde{L}_2C|w_r^i|\right)$$

$$\leq \left(2\eta c_1\sigma_a\tau\sqrt{2\log m}\right)\left(\tilde{L}_1 + \tilde{L}_2 + 2c_2C\tilde{L}_2\sigma_{wb}\sqrt{2\log m}\right)t + \left(2\eta c_1\sigma_a\tau\sqrt{2\log m}\tilde{L}_2C\right)\left(\sum_{i=0}^{t-1}|w_r^i|\right)$$

Define $\alpha$ and $\beta$ as

$$\alpha = \left(2\eta c_1 \tau \sigma_a \sqrt{2\log m}\right)\left(\tilde{L}_1 + \tilde{L}_2 + 2c_2 C \tilde{L}_2 \sigma_{wb}\sqrt{2\log m}\right)$$

$$\beta = \left(2\eta c_1 \sigma_a \tau \sqrt{2\log m} \tilde{L}_2 C\right)$$

Using $\alpha$ and $\beta$,

$$|w_r^t| \le \alpha t + \beta \left(\sum_{i=0}^{t-1} |w_r^i|\right)$$

$$\text{where} \quad \sum_{i=0}^{t-1} |w_r^i| \le \alpha(t-1) + (1+\beta)\left(\sum_{i=0}^{t-2} |w_r^i|\right)$$

$$\le \alpha\left((t-1) + (1+\beta)(t-2)\right) + (1+\beta)^2 \left(\sum_{i=0}^{t-3} |w_r^i|\right)$$

$$\le \alpha\left((t-1) + (1+\beta)(t-2) + (1+\beta)^2(t-3)\right) + (1+\beta)^3 \left(\sum_{i=0}^{t-4} |w_r^i|\right)$$

In general, we can write

$$\sum_{i=0}^{t-1} |w_r^i| \le \alpha \left(\sum_{i=1}^{t-t'-1} (1+\beta)^{i-1}(t-i)\right) + (1+\beta)^{(t-t'-1)}\left(\sum_{i=0}^{t'} |w_r^i|\right)$$

Taking $t' = 0$,

$$\sum_{i=0}^{t-1} |w_r^i| \le \alpha \left(\sum_{i=1}^{t-1} (1+\beta)^{i-1}(t-i)\right)$$

$\sum_{i=1}^{t-1} (1+\beta)^{(i-1)}(t-i)$ is sum of an arithmetic-geometric progression (AGP). Using Fact H.6,

$$\sum_{i=0}^{t-1} |w_r^i| \le \alpha \left(\sum_{i=1}^{t-1} (1+\beta)^{i-1}(t-i)\right)$$

$$= \alpha \left(\frac{(t-1) - (1+\beta)^{t-1}}{-\beta} - \frac{(1+\beta)\left(1 - (1+\beta)^{t-2}\right)}{\beta^2}\right)$$

$$= \alpha \left(\frac{\beta(1+\beta)^{t-1} - \beta(t-1) - (1+\beta) + (1+\beta)^{t-1}}{\beta^2}\right)$$

$$= \alpha \left(\frac{(1+\beta)^t - (1+\beta t)}{\beta^2}\right) \tag{F.3}$$

Using Eq.(F.3) to bound $|w_r^t|$,

$$|w_r^t| \le \alpha \left(t + \beta \left(\frac{(1+\beta)^t - (1+\beta t)}{\beta^2}\right)\right)$$

$$= \alpha \left(\frac{(1+\beta)^t - 1}{\beta}\right)$$

$$= \left(\frac{\tilde{L}_1 + \tilde{L}_2 + 2c_2 C \tilde{L}_2 \sigma_{wb}\sqrt{2\log m}}{\tilde{L}_2 C}\right)\left(\left(1 + 2\eta c_1 \sigma_a \tau \sqrt{2\log m}\tilde{L}_2 C\right)^t - 1\right) = \Delta_w^t$$

Similarly, we

$$\frac{\partial L_G(f_t, x)}{\partial b_r} = N_t(x)\tau \left(a_{r0}\sigma'((w_{r0} + w_r^t)x + b_{r0} + b_r^t)\right) - \frac{\tau}{N_t'(x)}\left(a_{r0}(w_{r0} + w_r^t)\sigma''((w_{r0} + w_r^t)x + b_{r0} + b_r^t)\right)$$

We assume that $L_G(f_t, x)$ is $\tilde{L}_1$-lipschitz wrt $N_t(x)$ and $\tilde{L}_2$-lipschitz wrt $N'_t(x)$. Additionaly, assuming that $|\sigma'(.)| \leq 1$ and $|\sigma''(.)| \leq C$,

$$\left| \frac{\partial L_G(f_t, x)}{\partial b_r} \right| \leq \tilde{L}_1 a_{r0} \tau + \tilde{L}_2 a_{r0} C \tau \left( w_{r0} + |w_r^t| \right)$$

Using Lemma H.2 for $a_{r0}$ and $w_{r0}$, with at least $1 - \frac{1}{c_1} - \frac{1}{c_2}$ probability,

$$\left| \frac{\partial L_G(f_t, x)}{\partial b_r} \right| \leq \left( 2c_1\sigma_a\sqrt{2\log m} \right) \tau \left( \tilde{L}_1 + \tilde{L}_2 C|w_r| + 2c_2 C\tilde{L}_2\sigma_{wb}\sqrt{2\log m} \right) \tag{F.4}$$

For projected gradient descent,

$$|b_r^t| \leq \eta \sum_{i=0}^{t-1} \left| \frac{\partial L_G(f_t, x^i)}{\partial b_r} \right|$$

$$= 2\eta c_1\sigma_a\tau\sqrt{2\log m} \left( \tilde{L}_1 + 2c_2\tilde{L}_2 C\sigma_{wb}\sqrt{2\log m} \right) t + 2\eta c_1\sigma_a\tau\tilde{L}_2 C\sqrt{2\log m} \left( \sum_{i=0}^{t-1} |w_r^i| \right)$$

Using Eq.(F.3),

$$|b_r^t| \leq 2\eta c_1\sigma_a\tau\sqrt{2\log m} \left( \tilde{L}_1 + 2c_2\tilde{L}_2 C\sigma_{wb}\sqrt{2\log m} \right) t$$
$$+ 2\eta c_1\sigma_a\tilde{L}_2 C\sqrt{2\log m} \left( \frac{\tilde{L}_1 + \tilde{L}_2 + 2c_2 C\tilde{L}_2\sigma_{wb}\sqrt{2\log m}}{2\eta c_1\sigma_a\sqrt{2\log m}\tilde{L}_2^2 C^2} \right) \left( \left( 1 + 2\eta c_1\tau\sigma_a\sqrt{2\log m}\tilde{L}_2 C \right)^t - 1 \right)$$
$$= 2\eta c_1\sigma_a\tau\sqrt{2\log m} \left( \tilde{L}_1 + 2c_2\tilde{L}_2 C\sigma_{wb}\sqrt{2\log m} \right) t$$
$$+ \left( \frac{\tilde{L}_1 + \tilde{L}_2 + 2c_2 C\tilde{L}_2\sigma_{wb}\sqrt{2\log m}}{\tilde{L}_2 C} \right) \left( \left( 1 + 2\eta c_1\sigma_a\tau\sqrt{2\log m}\tilde{L}_2 C \right)^t - 1 \right) = \Delta_b^t$$

Finding lower bound on $\Delta_b^t$, we get

$$\Delta_b^t \geq 2\eta c_1\sigma_a\tau\sqrt{2\log m} \left( \tilde{L}_1 + 2c_2\tilde{L}_2 C\sigma_{wb}\sqrt{2\log m} \right) t$$
$$+ \left( \frac{\tilde{L}_1 + \tilde{L}_2 + 2c_2 C\tilde{L}_2\sigma_{wb}\sqrt{2\log m}}{\tilde{L}_2 C} \right) \left( \left( 1 + 2\eta t c_1\sigma_a\tau\sqrt{2\log m}\tilde{L}_2 C \right) - 1 \right)$$
$$= 2\eta c_1\sigma_a\tau\sqrt{2\log m} \left( \tilde{L}_1 + 2c_2\tilde{L}_2 C\sigma_{wb}\sqrt{2\log m} \right) t$$
$$+ \left( \frac{\tilde{L}_1 + \tilde{L}_2 + 2c_2 C\tilde{L}_2\sigma_{wb}\sqrt{2\log m}}{\tilde{L}_2 C} \right) \left( 2\eta t c_1\sigma_a\tau\sqrt{2\log m}\tilde{L}_2 C \right)$$
$$= \Omega \left( \eta t\sigma_a\tau\sigma_{wb}\log m \right)$$

We get similar lower bound on $\Delta_w^t$.

$$\Delta_w^t = \Omega \left( \eta t\sigma_a\tau\sigma_{wb}\log m \right)$$

Now, we show coupling between $N_t'(x)$ and $P_t'(x)$. Assuming that $\sigma'$ is C-lipschitz continuous,

$$|N_t'(x) - P_t'(x)| = \left| \tau \sum_{r=1}^{m} a_{r0} \big( (w_r^t + w_{r0}) \left( \sigma'((w_{r0} + w_r^t)x + b_{r0} + b_r^t) - \sigma'(w_{r0}x + b_{r0}) \right) \right.$$

$$\left. - \sigma''(w_{r0}x + b_{r0}) \left( w_{r0}(w_r^t x + b_r^t) \right) \big) \right|$$

$$\leq C\tau \sum_{r=1}^{m} 2a_{r0}|w_r^t + w_{r0}| + C\tau \sum_{r=1}^{m} a_{r0}w_{r0}|w_r^t x + b_r^t|$$

$$= C\tau \sum_{r=1}^{m} a_{r0} \left( 2|w_r^t + w_{r0}| + w_{r0}|w_r^t x + b_r^t| \right)$$

$$\overset{(i)}{\leq} C\tau \left( 2c_1\sigma_a\sqrt{2\log m} \right) \left( \|W^t\|_1 + \|W_0\|_1 + 2c_2\sigma_{wb}\sqrt{2\log m} \left( \|W^t\|_1 + \|B^t\|_1 \right) \right)$$

$$= O\left( \tau\sigma_a\sigma_{wb}m\log m \left( \Delta_w^t + \Delta_b^t \right) \right)$$

where inequality $(i)$ follows from Fact H.2. Now, using $t = T$ to find upper bound on $|N_T'(x) - P_T'(x)|$, we get

$$|N_T'(x) - P_T'(x)| = O\left( \tau\sigma_a\sigma_{wb}m\log m \left( \Delta_w^T + \Delta_b^T \right) \right)$$

$$= O\left( \tau\sigma_a\sigma_{wb}m\log m \left( \Delta_w^T + \Delta_b^T \right) \right)$$

$$= O\left( \tau\sigma_a\sigma_{wb}m\log m \left( \Delta_w^T + \Delta_b^T \right) \right)$$

$$= O\left( \tau\sigma_a\sigma_{wb}m\log m \left( \eta T\sigma_a\tau\sigma_{wb}\log m \right) \right)$$

$$= O\left( \eta T\tau^2\sigma_a^2\sigma_{wb}^2 m \left( \log m \right)^2 \right)$$

$$= O\left( \frac{\tau^2\sigma_a^2\sigma_{wb}^2 m \left( \log m \right)^2}{\epsilon m\tau^2\sigma_a^2\log m} \right)$$

$$= O\left( \frac{\sigma_{wb}^2 \left( \log m \right)}{\epsilon} \right)$$

For $\sigma_{wb}$ in range $\Omega\left( \frac{1}{\log m} \right)$ and $O(1)$, $|N_T'(x) - P_T'(x)|$ can become very high and in those cases, $N_T'(x)$ and $P_T'(x)$ willnot remain close. $\qquad \square$

## G    ADDITIONAL EXPERIMENTS

In this section, we show experimental results on synthetic 1D data to support our theoretical findings. We use the same architecture and initialization as described in Sec. 2 for both constrained and unconstrained normalizing flows. In all our experiments, we fix the weights of the output layer and train the weights and biases of the hidden layer. For training, we use mini-batch SGD with batch size 32. We use 2 datasets each with 10,000 data points. One of the dataset is mixture of 2 Gaussians and other one is mixture of 3 beta distributions. All results are averaged over 3 different iterations.

### G.1    RESULTS FOR UNCONSTRAINED NORMALIZING FLOW

In Fig. 2, we show comparison of data distribution and generated data distribution for unconstrained normalizing flow. Unconstrained normalizing flow with exponential distribution as a base distribution learns the data distribution well. We study the effect of overparameterization on L2-norm of $W^t$ and $B^t$ and convergence speed. To reproduce situation similar to the theoretical analyses for unconstrained normalizing flow, we choose learning rate as $\frac{c}{m}$ where $c$ is a constant. The first row of Figure 3 contains results for mixture of Gaussians dataset and the second row contains results for mixtures of beta distributions dataset. From Fig. 3, we see that L2-norm of $W^t$ and $B^t$ decreases with increasing $m$. Moreover, the change is proportional to $1/\sqrt{m}$ which is similar to the bound in theoretical result. From the last column of Fig. 3, we see that the training speed for different values of $m$ remains almost constant. Our choice of $T$ in theoretical analysis also poly-logarithmically depends upon $m$. We obtained similar results for Gaussian distribution as well.

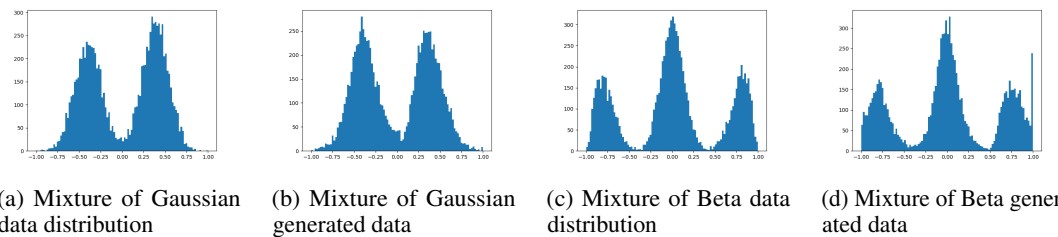

(a) Mixture of Gaussian data distribution

(b) Mixture of Gaussian generated data

(c) Mixture of Beta data distribution

(d) Mixture of Beta generated data

Figure 2: Comparison of data distribution and generated data for mixture of Gaussian and beta distributions

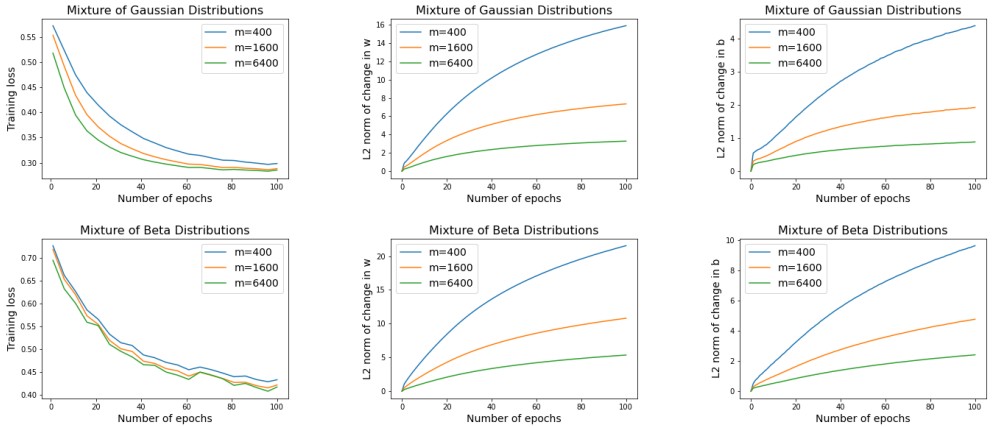

Figure 3: Effect of over-parameterization on training of unconstrained normalizing flow on mixture of Gaussian and mixture of beta distributions

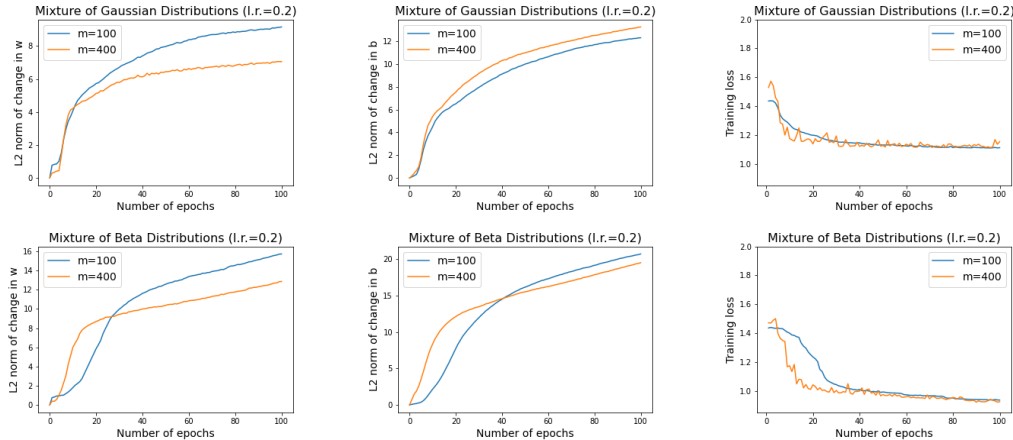

Figure 4: Effect of over-parameterization on training of small sized constrained normalizing flow

## G.2 RESULTS FOR CONSTRAINED NORMALIZING FLOW

In Sec. 2.1, we suggested that high overparameterization may adversely affect training for constrained normalizing flows. We now give experimental evidence for this in Figs 4, 5 and 6. We use Gaussian distribution as a base distribution for all our experiments of constrained normalizing flow. In Fig.4, we see that for neural network with $m = 100$ and $m = 400$, the training loss decreases stably. In Figs. 5 and 6, we see that as we increase the learning rate, training becomes more stable for larger $m$. Note that for learning rate equal to 0.025 and 0.0125, constrained normalizing flow with $m = 1600$ doesn't learn anything due to small learning rate. We observe that the $L2$-norms of $W^t$ and $B^t$ for $m = 6400$ are at least at large as those of $m = 1600$. On both datasets, as we increase the learning rate, L2-norm of $B^t$ increases (except for learning rate=0.05 of mixture of beta distribution) and learning of constrained normalizing flow becomes more and more stable. We also experimented with more number of epochs (1000 epochs) for mixture of Gaussian dataset for number of hidden layers $m = 1600, 6400$ (see Fig.7). These observations support our claim in Sec.2.1 that for learning and approximation of overparameterized constrained normalizing flow, neural networks need large $L2$-norms of $W^t$ and $B^t$.

## H USEFUL FACTS

**Lemma H.1.** *Suppose $Z_k \sim \mathcal{N}(0, \sigma^2)$ and $Y = \sum_{k=1}^{n} Z_k^2$ is chi-squared distribution with following property for all $t \in (0, 1)$.*

$$Pr \left[ \left| \frac{1}{n} \sum_{k=1}^{n} Z_k^2 - \sigma^2 \right| \geq t \right] \leq 2 \exp \left( -\frac{nt^2}{8\sigma^4} \right)$$

*Proof.* From example 2.11 from Wainwright (2019), for $Z'_k \sim \mathcal{N}(0, 1)$ and $Y = \sum_{k=1}^{n} Z'^2_k$ is chi-squared distribution with following property for all $t \in (0, 1)$.

$$\Pr \left[ \left| \frac{1}{n} \sum_{k=1}^{n} Z'^2_k - 1 \right| \geq t \right] \leq 2 \exp \left( -\frac{nt^2}{8} \right)$$

Using above equation for $\frac{Z_k}{\sigma}$,

$$\Pr \left[ \left| \frac{1}{n} \sum_{k=1}^{n} \frac{Z_k^2}{\sigma^2} - 1 \right| \geq \frac{t}{\sigma^2} \right] \leq 2 \exp \left( -\frac{nt^2}{8\sigma^4} \right)$$

$$\Pr \left[ \left| \frac{1}{n} \sum_{k=1}^{n} Z_k^2 - \sigma^2 \right| \geq t \right] \leq 2 \exp \left( -\frac{nt^2}{8\sigma^4} \right)$$

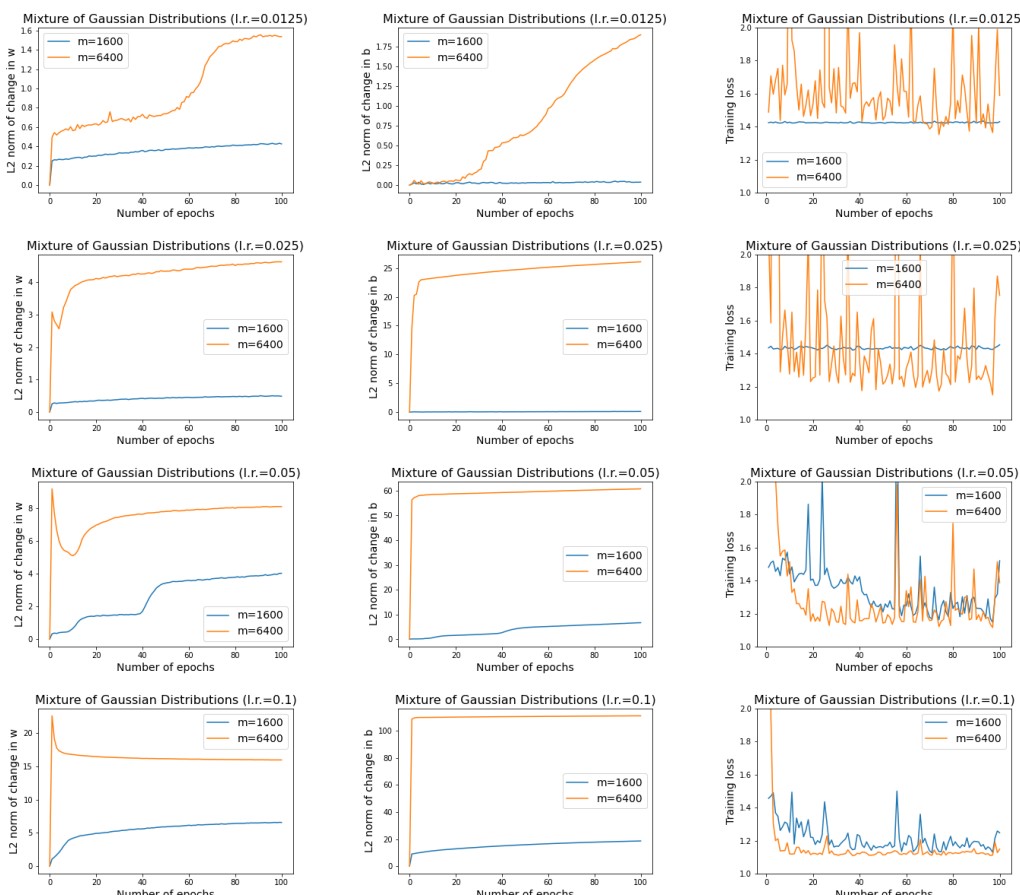

Figure 5: Effect of over-parameterization on training of constrained normalizing flow on mixture of Gaussian dataset for number of hidden layers $m = 1600, 6400$

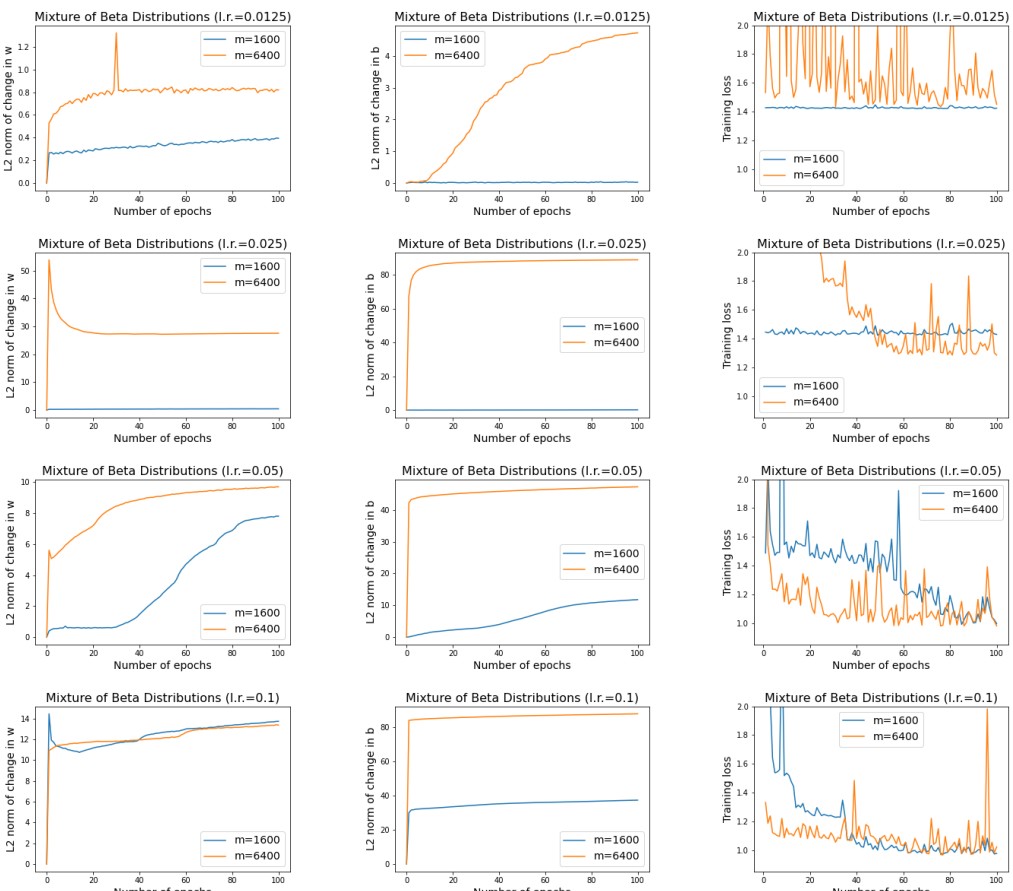

Figure 6: Effect of over-parameterization on training of constrained normalizing flow on mixture of beta distribution dataset for number of hidden layers $m = 1600, 6400$

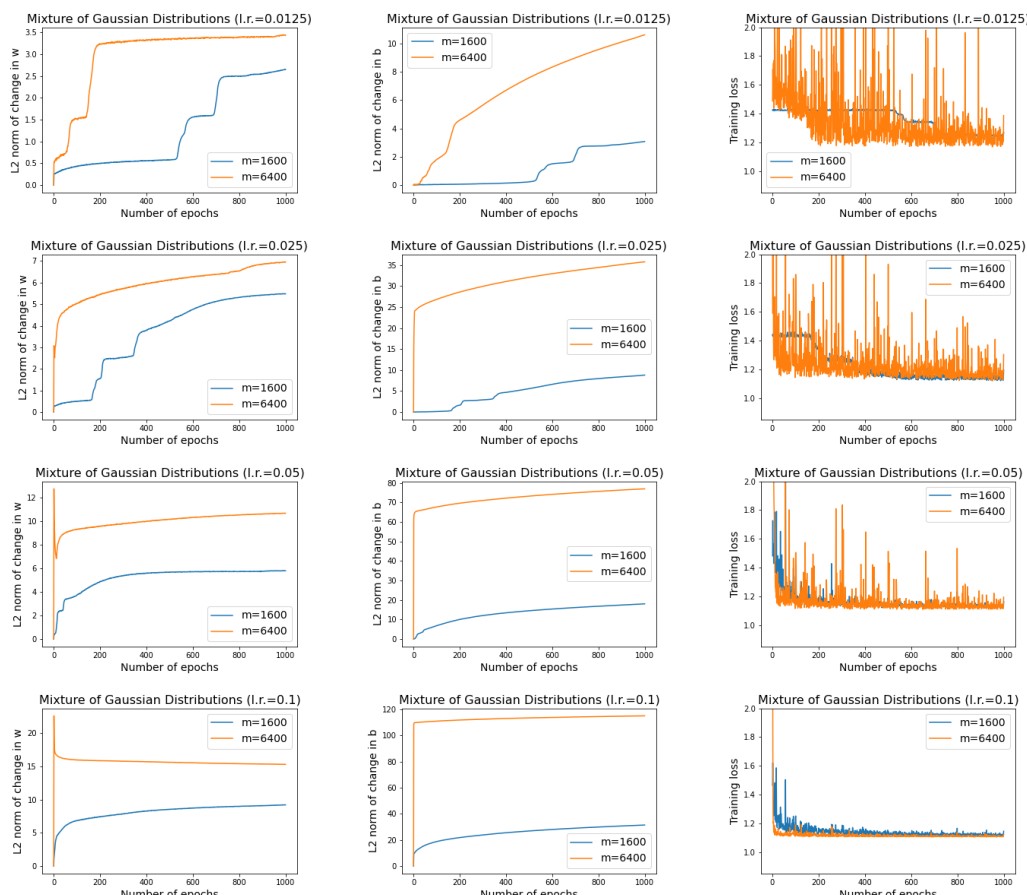

Figure 7: Effect of over-parameterization on training of constrained normalizing flow on mixture of Gaussian dataset for epochs=1000 and number of hidden layers $m = 1600, 6400$

$\square$

**Lemma H.2.** *Let $X_1, X_2, ..., X_n$ be independent random variables from $\mathcal{N}(0, \sigma^2)$, then with at least $1 - \frac{1}{c_1}$ probability, following holds.*

$$\max_{i \in \{1,2,...,n\}} |X_i| \leq 2c_1 \sigma \sqrt{2 \log n}$$

*Proof.* From Romberg (2012),

$$\mathbb{E}\left[\max_{i \in \{1,2,...,n\}} |X_i|\right] \leq \sigma \left(\sqrt{2 \log n} + 1\right) \leq 2\sigma \left(\sqrt{2 \log n}\right)$$

Assuming $n \geq 2$, the last inequality follows. Using Markov's inequality,

$$\Pr\left(\max_{i \in \{1,2,...,n\}} |X_i| \geq 2c_1\sigma\left(\sqrt{2 \log n}\right)\right) \leq \frac{1}{c_1}$$

$$\Pr\left(\max_{i \in \{1,2,...,n\}} |X_i| \leq 2c_1\sigma\left(\sqrt{2 \log n}\right)\right) \geq 1 - \frac{1}{c_1}$$

$$s$$

$\square$

**Lemma H.3.** *For standard Gaussian random variable $X$ from $\mathcal{N}(0, \sigma^2)$, following anti-concentration inequality holds.*

$$\Pr\left(|X| \leq R\right) \leq \frac{2R}{\sigma\sqrt{2\pi}}$$

*Proof.* (From Du et al. (2018)) For standard Gaussian random variable $\frac{X}{\sigma}$,

$$\Pr\left(\left|\frac{X}{\sigma}\right| \leq R\right) \leq \frac{2R}{\sqrt{2\pi}}$$

Using $R = \frac{R'}{\sigma}$, we get the required result. $\square$

**Lemma H.4.** *Suppose function $f : \mathbb{R}^d \to \mathbb{R}$ is $L_g$-Lipschitz continuous and $L_i$-coordinate wise Lipschitz continuous i.e.*

$$|f(\mathbf{a}) - f(\mathbf{b})| \leq L_g \|\mathbf{a} - \mathbf{b}\|$$
$$\forall \mathbf{a}, \mathbf{b} \in \mathbb{R}^d \quad \text{(Standard Lipschitz continuity)}$$
$$|f(a_1, a_2, ..., a_i, ..., a_d) - f(a_1, a_2, ..., b_i, ..., a_d)| \leq L_i |a_i - b_i|$$
$$\forall a_1, a_2, ..., a_i, ..., a_d, b_i \in \mathbb{R} \text{ and } \forall i \in [d] \quad \text{(Coordinate-wise Lipschitz continuity)}$$

*If a function $f$ satisfies $L_i$-coordinate wise Lipschitz continuity for all $i$, then function $f$ follows following inequality.*

$$|f(a_1, a_2, ..., a_d) - f(b_1, b_2, ..., b_d)| \leq \sum_{i=1}^{n} L_i |a_i - b_i|$$

*Moreover, the function $f$ also satisfies standard Lipschitz continuity with $L_g$ Lipschitz constant where inequality between $L_g$ and $L_i$ is as follows.*

$$L_g \leq \sqrt{\sum_{i=1}^{d} L_i^2}$$

*Proof.* Define $\mathbf{a} = (a_1, a_2, ..., a_d)$ and $\mathbf{b} = (b_1, b_2, ..., b_d)$.

$$
\begin{aligned}
|f(a_1, a_2, ..., a_d) - f(b_1, b_2, ..., b_d)| \leq & |f(a_1, a_2, ..., a_d) - f(b_1, a_2, ..., a_d)| \\
& + |f(b_1, a_2, a_3, ..., a_d) - f(b_1, b_2, a_3, ..., a_d)| \\
& + |f(b_1, b_2, a_3, ..., a_d) - f(b_1, b_2, b_3, ..., a_d)| \\
& + ... + |f(b_1, b_2, ..., b_{d-1}, a_d) - f(b_1, b_2, b_3, ..., b_d)| \\
\leq & L_1 |a_1 - b_1| + L_2 |a_2 - b_2| + ... + L_d |a_d - b_d| \\
\leq & \sqrt{\sum_{i=1}^{d} L_i^2} \|\mathbf{a} - \mathbf{b}\|_2
\end{aligned}
$$

where last inequality follows from Cauchy-Schwarz inequality. $\square$

**Fact H.1.** *(Hoeffding's inequality on Binomial random variable) If we have a binomial random variable with parameters $n$ (total number of trials) and $p$ (probability of success). For number successful trial $k \geq np$, following inequality holds.*

$$
Pr(X \geq k) \leq \exp\left(-2n\left(\frac{k}{n} - p\right)^2\right)
$$

**Fact H.2.** *(Half-normal distribution) If $X$ follows a normal distribution with with mean 0 and variance $\sigma^2$, $\mathcal{N}\left(0, \sigma^2\right)$, then $Y = |X| = X\,sign(X)$ follows a half-normal distribution with mean $\mathbb{E}[Y] = \frac{\sigma\sqrt{2}}{\sqrt{\pi}}$.*

**Fact H.3.** *For a gaussian random variable $X \sim \mathcal{N}(0, \sigma^2)$, $\forall t \in (0, \sigma)$, we have*

$$
Pr(|X| \geq t) \geq 1 - \frac{4t}{5\sigma}
$$

**Fact H.4.** *The sum of reciprocals of the squares of the natural numbers is given by*

$$
\sum_{n=1}^{\infty} \frac{1}{n^2} = \frac{\pi^2}{6} \leq 2
$$

**Fact H.5.** *(Theorem $3.1(r_5')$ of Li & Yeh (2013)) For any $\alpha > 1$ and $x \in \left[0, \frac{1}{\alpha-1}\right)$,*

$$
(1 + x)^{\alpha} \leq \frac{1}{1 - \frac{\alpha x}{1+x}} = 1 + \frac{\alpha x}{1 - (\alpha - 1) x}
$$

**Fact H.6.** *If Arithmetic-Geometric Progression(AGP) is as follows.*

$$
a, (a + d)r, (a + 2d)r^2, (a + 3d)r^3, ...., [a + (n - 1)d] r^{n-1}
$$

*where $a$ is the initial term, $d$ is the common difference and $r$ is the common ratio. The sum of the first $n$ terms of the AGP ($S_n$) is given by*

$$
S_n = \frac{a - [a + (n - 1)d] r^n}{1 - r} + \frac{dr\left(1 - r^{n-1}\right)}{(1 - r)^2}
$$

**Fact H.7.** *( Lemma A.3 from Ji et al. (2020) ) The Fourier transform of $f$ is defined as*

$$
\hat{f}(w) = \int f(x)e^{2\pi i w x} dx
$$

*The polar decomposition of the Fourier transform $\hat{f}$ is $\hat{f}(w) = \left|\hat{f}(w)\right| e^{2\pi i \theta_f(w)}$ with $|\theta_f(w)| \leq 1$. The Fourier transform $\hat{f}$ follows below properties.*

1. *$\left|\hat{f}(w)\right| \leq \|f\|_{L_1}$ for any real number $w$.*

2. *Let $\alpha > 0$ be given and define $\beta := \frac{1}{2\pi\alpha}$. $G_\alpha$ is Gaussian with coordinate-wise variance $\alpha^2$. Then $\left|\hat{G}_\alpha\right| = \hat{G}_\alpha$ (meaning $\hat{G}_\alpha$ has no radial component) and*

$$\hat{G}_\alpha(w) = \frac{1}{\sqrt{2\pi\alpha^2}}G_\beta(w) = \sqrt{2\pi\beta^2}G_\beta(w) = \sqrt{2\pi}G\left(w/\beta\right)$$

**Definition H.1.** *Let $\mathcal{F}$ be a set of functions $\mathbb{R}^d \to \mathbb{R}$ and $\mathcal{X} = (x_1, x_2, ..., x_n)$ be a finite set of samples. The empirical Rademacher complexity of $\mathcal{F}$ with respect to $\mathcal{X}$ is defined by*

$$\hat{\mathcal{R}}\left(\mathcal{X}; \mathcal{F}\right) = \mathbb{E}_{\xi \sim \{\pm 1\}^n}\left[\sup_{f \in \mathcal{F}}\frac{1}{n}\sum_{i=1}^{n}\xi_i f(x_i)\right]$$

**Lemma H.5.** *The Rademacher complexity have following properties.*

a. *Suppose $|x| \leq 1$ for all $\mathcal{X}$. The class $\mathcal{F} = \{x \to wx + b \mid |w| \leq B, |b| \leq B\}$ has Rademacher complexity $\hat{\mathcal{R}}\left(\mathcal{X}, \mathcal{F}\right) \leq \frac{2B}{\sqrt{n}}$*

b. *Given $\mathcal{F}_1, \mathcal{F}_2$ classes of functions, then $\hat{\mathcal{R}}\left(\mathcal{X}; \mathcal{F}_1 + \mathcal{F}_2\right) = \hat{\mathcal{R}}\left(\mathcal{X}; \mathcal{F}_1\right) + \hat{\mathcal{R}}\left(\mathcal{X}; \mathcal{F}_2\right)$*

c. *Given $\mathcal{F}_1, \mathcal{F}_2, ..., \mathcal{F}_m$ classes of functions from $\mathcal{X} \to \mathbb{R}$ and suppose $w \in \mathbb{R}^m$ is a fixed vector, then $\mathcal{F}' = \{x \to \sum_{r=1}^{m}w_r\sigma\left(f_r(x)\right) \mid f_r \in \mathcal{F}_r\}$ satisfies $\hat{\mathcal{R}}\left(\mathcal{X}; \mathcal{F}'\right) \leq 2\|w\|_1 \max_{r \in [m]}\hat{\mathcal{R}}\left(\mathcal{X}; \mathcal{F}_r\right)$ where $\sigma$ is 1-Lipschitz continuous function.*

*Proof.* The b and c parts of the proposition are from Allen-Zhu et al. (2019) . Proof of the a part is as following.

$$\hat{\mathcal{R}}\left(\mathcal{X}, \mathcal{F}\right) = \frac{1}{n}\mathbb{E}_{\xi \sim \{\pm 1\}^n}\left[\sup_{f \in \mathcal{F}}w\left(\sum_{i=1}^{n}\xi_i x_i\right) + b\left(\sum_{i=1}^{n}\xi_i\right)\right]$$

$$= \frac{B}{n}\mathbb{E}_{\xi \sim \{\pm 1\}^n}\left[\left|\sum_{i=1}^{n}\xi_i x_i\right| + \left|\sum_{i=1}^{n}\xi_i\right|\right]$$

Using Jensen's inequality, we get

$$\mathbb{E}_{\xi \sim \{\pm 1\}^n}\left[\left|\sum_{i=1}^{n}\xi_i x_i\right|\right] = \mathbb{E}_{\xi}\left[\left(\left|\sum_{i=1}^{n}\xi_i x_i\right|^2\right)^{0.5}\right] \leq \left(\mathbb{E}_{\xi}\left[\left|\sum_{i=1}^{n}\xi_i x_i\right|^2\right]\right)^{0.5}$$

Using independence of $\xi_i$ for all $i \in [n]$, we get

$$\mathbb{E}_{\xi}\left[\left(\sum_{i=1}^{n}\xi_i x_i\right)^2\right] = \mathbb{E}_{\xi}\left[\sum_{i,j}\xi_i\xi_j x_i x_j\right]$$

$$= \sum_{i=1}^{n}|x_i|^2 \leq n$$

Using same technique, we will get same bound for $\mathbb{E}_{\xi}\left[|\sum_{i=1}^{n}\xi_i|\right]$.

$$\hat{\mathcal{R}}\left(\mathcal{X}, \mathcal{F}\right) \leq \frac{2B}{\sqrt{n}}$$

$\square$

**Fact H.8.** *(Rademacher Complexity) If $\mathcal{F}_1, \mathcal{F}_2, ..., \mathcal{F}_k$ are $k$ classes of functions $\mathbb{R}^d \to \mathbb{R}$ and $L_x : \mathbb{R}^d \to [-b, b]$ is $L_g$-Lipschitz continuous function for any $x \sim \mathcal{D}$, then*

$$\sup_{f_1 \in \mathcal{F}_1, ..., f_k \in \mathcal{F}_k}\left|\mathbb{E}_{x \in \mathcal{D}}\left[L_x\left(f_1(x), ..., f_k(x)\right)\right] - \frac{1}{n}\sum_{i=1}^{n}L_x\left(f_1(x_i), ..., f_k(x_i)\right)\right| \leq 2\hat{\mathcal{R}}\left(\mathcal{X}; \mathcal{L}\right) + b\sqrt{\frac{\log\frac{1}{\delta}}{2n}}$$

*where $\mathcal{L}$ is set of all functions $L_x$. Using vector contraction inequality from Maurer (2016), we get*

$$\sup_{f_1 \in \mathcal{F}_1, \dots, f_k \in \mathcal{F}_k} \left| \mathbb{E}_{x \in \mathcal{D}} \left[ L_x \left( f_1(x), \dots, f_k(x) \right) \right] - \frac{1}{n} \sum_{i=1}^{n} L_x \left( f_1(x_i), \dots, f_k(x_i) \right) \right|$$

$$\leq 2\sqrt{2} L_g \left( \sum_{i=1}^{k} \hat{\mathcal{R}} \left( \mathcal{X}; \mathcal{F}_i \right) \right) + b \sqrt{\frac{\log \frac{1}{\delta}}{2n}}$$

