# OpenReview forum: "Learning and Generalization in Univariate Overparameterized Normalizing Flows"
_ICLR.cc/2021/Conference — Reject_

### Official Review · AnonReviewer2 · 2020-10-27
**Review for `'Learning and Generalization in Univariate Overparameterized Normalizing Flows'**

**Rating:** 5
**Confidence:** 3

**Review:**

This paper studies overparameterization over unsupervised learning. In detail, it uses constrained normalizing flows (CNF) and unconstrained normalizing flows (UNF) to learn the underlying unknown one-dimensional distribution, which can be parameterized by a two-layer neural network. The authors propose theoretical results for UNF and suggest that by selecting wide enough neural networks, a great number of random samples and number of quadrature points, a two-layer neural network is able to learn the true UNF up to small error. Experiment results are presented for both CNF and UNF, which back up their claim.

Here are my detailed comments.







- The presentation of theoretical results can be further improved. For instance:
-- In Lemma 1, what does $(\phi^{-1}(F’)){\|\delta}$ mean?
-- In Theorem 2, it seems that to derive a finite-sample analysis, the second-order derivative of $F^*$ should be finite. The authors may want to add such a claim in the statement of Theorem 1.
-- What is the definition of  $\tilde w_i$ in Lemma 3?
-- What are the definitions of $M_{\hat L}$ and $m_{\hat L}$ in Lemma 12? Are they related to $n$?
-- Second line in Page 24, eq.() is a typo.
- My main concern is the scalability issue. The main theorem suggests that it is possible to use a neural network to approximate the first-order derivative of the unknown distribution transformation $f$, and to use the neural network to construct the original function $f$ with sufficient quadrature points. However, just as Theorem 1 suggests, the number of quadrature points is of the order $O(1/\epsilon)$, where $\epsilon$ is the approximation error. Thus, it seems that for a $d$-dimension case, the number of quadrature points may be the order of $O(1/\epsilon^d)$. Such an exponential dependence is unacceptable in terms of the scalability. Can the authors explain more about the high-dimension case?
- In Section 2.1, the authors suggest that overparameterization may hurt the overall performance of CNF by showing experiment results of tanh activation function. Is it true that the failure is actually due to the gradient explosion caused by tanh activation function rather than overparameterization? Meanwhile, it seems that the reason for the authors to use tanh is because they want activations with continuous derivatives and convexity for loss function. Why do we need such a convexity? Is it due to some theoretical concerns (like to make the derivation go through) or concerns from practice?
- The authors may want to discuss existing results about optimization and generalization of overparameterized deep neural networks [1-3], which are related to this work. Besides, this work relies on the idea of the existence of a pseudo network which approximates the target function well, which may be related to [4-6]. Can the authors discuss and show the relations between these works?

[1] Zou, Difan, et al. "Gradient descent optimizes over-parameterized deep ReLU networks." Machine Learning 109.3 (2020): 467-492.

[2] Du, Simon, et al. "Gradient descent finds global minima of deep neural networks." International Conference on Machine Learning. 2019.

[3] Allen-Zhu, Zeyuan, Yuanzhi Li, and Zhao Song. "A convergence theory for deep learning via over-parameterization." International Conference on Machine Learning. PMLR, 2019.

[4] Allen-Zhu, Zeyuan, and Yuanzhi Li. "What Can ResNet Learn Efficiently, Going Beyond Kernels?." Advances in Neural Information Processing Systems. 2019.

[5] Chen, Zixiang, et al. "How Much Over-parameterization Is Sufficient to Learn Deep ReLU Networks?." arXiv preprint arXiv:1911.12360 (2019).

[6] Ji, Ziwei, and Matus Telgarsky. "Polylogarithmic width suffices for gradient descent to achieve arbitrarily small test error with shallow relu networks." arXiv preprint arXiv:1909.12292 (2019).

---

> ### Author Response · Authors · 2020-11-25
> **Response (part 1/2)**
>
> Thank you for reading our paper and for your thoughtful comments.
>
> *The presentation of theoretical results can be further improved.*
>
> Thank you for detailed comments. We will incorporate them where appropriate.
>
> *In Lemma 1, what does $(\phi^{−1}(F^′))_{|δ}$ mean?*
>
> Please see the second line of Result 1 on page 12 in the original submission.
>
> *In Theorem 2, it seems that to derive a finite-sample analysis, the second-order derivative of $F^∗$ should be finite. The authors may want to add such a claim in the statement of Theorem 1.*
>
> You are right and we have rectified this in the revised version.
>
> *What is the definition of $\tilde{w}_i$ in Lemma 3?*
>
> $\tilde{w}_i$ denotes weight of function value at $i^{th}$ quadrature point in the quadrature sum in the original version of the paper. To make it simpler and more readable, we have changed this notation to $\Delta_x$ in the new version because weights of function value at all quadrature points in the sum are same.
>
> *What are the definitions of $M_{\hat{L}}$ and $m_{\hat{L}}$ in Lemma 12? Are they related to $n$?*
>
> Thank you for pointing out this omission. We have included both definitions in Eq.(E.1) and Eq.(E.2). No, $M_{\hat{L}}$ and $m_{\hat{L}}$ doesn't depend upon $n$.
>
> *Second line in Page 24, eq.() is a typo.*
>
> Corrected; thanks.
>
> *My main concern is the scalability issue. The main theorem suggests that it is possible to use a neural network to approximate the first-order derivative of the unknown distribution transformation $f$, and to use the neural network to construct the original function $f$ with sufficient quadrature points. However, just as Theorem 1 suggests, the number of quadrature points is of the order $O(1/\epsilon)$, where $\epsilon$ is the approximation error. Thus, it seems that for a $d$-dimension case, the number of quadrature points may be the order of $O(1/ \epsilon^d )$. Such an exponential dependence is unacceptable in terms of the scalability. Can the authors explain more about the high-dimension case?*
>
> No, the number of quadrature points will not be exponential in $d$ (i.e. $O(1/\epsilon^d)$) for the $d$-dimensional problem as the UNF network of Wehenkel and Louppe does not perform integration over the $d$-dimensional space. Note that in the univariate case, we need $O(1/\epsilon)$ number of quadrature points to approximate one integral with finite sum within $\epsilon$-error. In the $d$-dimensional case, we need to approximate $d$ integrals with finite sum (See Eq. (8) in Wehenkel and Louppe https://arxiv.org/abs/1908.05164). To approximate each integral by finite sum within $\epsilon$-error, we will need $O(1/\epsilon)$ number of quadrature points therefore, combining all of them, we will need $O(poly(d) / \epsilon)$ number of quadrature points to approximate the original function value.
>
>
> *In Section 2.1, the authors suggest that overparameterization may hurt the overall performance of CNF by showing experiment results of tanh activation function. Is it true that the failure is actually due to the gradient explosion caused by tanh activation function rather than overparameterization?*
>
> We believe that it is not due to gradient explosion because in Figure 1, we see that when the learning rate is large, L2 norm of change in $w$ and $b$ is large and training with large learning rate becomes more stable than training with smaller learning rate case but if it was happening due to gradient explosion then larger learning rate should have made learning more unstable.
>
> *Meanwhile, it seems that the reason for the authors to use tanh is because they want activations with continuous derivatives and convexity for loss function. Why do we need such a convexity? Is it due to some theoretical concerns (like to make the derivation go through) or concerns from practice?*
>
> The properties that are necessary for the activation for CNFs are (1) continuous first derivative,  (2) strictly monotonically increasing, (3) non-convex. These properties (satisfied by tanh) are essential for CNFs to work even in practice as discussed in the beginning of Sec. 2.1.
>
> For theoretical arguments, we additionally want the loss function for the pseudo network to be convex.

---

> ### Author Response · Authors · 2020-11-25
> **Response (part 2/2)**
>
> *The authors may want to discuss existing results about optimization and generalization of overparameterized deep neural networks [1-3], which are related to this work. Besides, this work relies on the idea of the existence of a pseudo network which approximates the target function well, which may be related to [4-6]. Can the authors discuss and show the relations between these works?*
>
> Thank you for the references. This area has a very large number of papers. We have cited papers that are most directly relevant to our work. Some of the references you provide are somewhat less closely related. In particular, some focus on training and do not treat generalization or are concerned with orthogonal considerations such as Resnets. We will include the ones that are directly relevant after looking at them in more detail.

---

### Official Review · AnonReviewer3 · 2020-10-28
**Interesting but somewhat limited problem setting with lack of novelty in proof techniques**

**Rating:** 4
**Confidence:** 4

**Review:**

Summary:

This paper proved that for a certain modified version of sufficiently-overparametrized univariate normalizing flows where the underlying neural network has only one hidden layer, with high probability it can learn a distribution that is close enough to the target distribution where the distance can be measured in, e.g., KL divergence. The width of the network, number of samples, and the number of quadrature points are required to be at least polynomial in inverse of error rate and complexity measure of the target distribution. The authors also provided theoretical evidence and did experiments on synthetic Gaussian mixture datasets to show that another variation of the normalizing flow model does not benefit from overparametrization under this one-hidden-layer univariate setting.

Pros:

1. Understanding why normalizing flow works and the learning process of the underlying neural networks is an important problem, and the idea of using overparametrization to explain this is interesting.

2. The experimental methodologies and theoretical computations appear to be correct.

3. The intuitions behind the problem setting (including the modifications to the algorithm) and the ideas behind the proof are explained in detail and easy to understand. The limitations of this paper are also discussed.

Cons:

1. This particular setting of normalizing flow models used in this paper might be a bit limited. The authors only analyzed the univariate case, which is far from the high-dimensional case in practice. It is possible that these two cases work in very different regimes due to the differences between high-dimensional and low-dimensional probabilities. The authors also made two important modifications to the unconstrained normalizing flow: changing the base distribution to standard exponential distribution and changing the quadrature to simple rectangle quadrature. These modifications can also make the model work in very different ways from practice, and the authors did not provide enough theoretical or experimental justifications for the modifications.

2. The techniques used in this paper mainly come from [1], and the proof framework and results are roughly the same. The authors made modifications to the setting so that the optimization becomes convex, and this seems to be the only justification for these modifications. The proof lies in the NTK/lazy training regime, which is hard to generalize to non-convex settings such as moderate width or large learning rate.

3. The structure of this paper may need some improvements. The introduction section is a bit too long with perhaps too much background knowledge for normalizing flows. The contribution and related work parts can also be shortened. Section 2 is also a bit too long, and it may be better to re-organize this section and separate it into preliminaries/main results/proof sketch/discussions to make it easier for the readers to understand.

4. The experiments in this paper are a bit simple, i.e., the authors only did experiments on synthetic datasets like Gaussian mixture with models whose underlying neural networks have only one hidden layer. This setting is far from empirical settings, which makes the conclusions of the experiments not so convincing.

[1] Allen-Zhu, Zeyuan, Yuanzhi Li, and Yingyu Liang. "Learning and generalization in overparameterized neural networks, going beyond two layers." Advances in neural information processing systems. 2019.

Recommendation:

I vote for rejecting this paper. As mentioned in "Cons", my major concern is the limitations of the problem setting in this paper and the novelty of the proof. The setting for normalizing flows in this paper is a bit far from practice, and the theoretical proof highly depends on the convexity of the optimization process, making the theoretical claims hard to generalize to more practical settings.

Supporting arguments for recommendation:

See "Cons", especially points 1 and 2 there.

Questions for the authors:

1. Please address the cons mentioned above.

2. The authors use "generalization" in the title, but I am a bit confused because I do not know what generalization means in this normalizing flow setting. Does this mean something like the KL divergence between the learned distribution and the target distribution?

3. In the last paragraph of page 5, the authors said that convex activations "cannot be used" because then N(x) would also be convex. Does N(x) have to be non-convex?

Additional feedback:

1. It may be better to explain "quadrature" by figures or examples in the introduction because this seems to be an important difference between normalizing flow models and normal neural networks but it is not explained in detail in the introduction. Explaining this earlier and clearer can help the readers better understand this paper.

2. Typos: In the abstract, "On the other" -> "On the other hand". In the paragraph just before Section 3, "Lemmas 1" -> "Lemma 1".

---

> ### Author Response · Authors · 2020-11-25
> **Response (part 1/2)**
>
> Thank you for reading our paper and for your thoughtful comments.
>
> *This particular setting of normalizing flow models used in this paper might be a bit limited. The authors only analyzed the univariate case, which is far from the high-dimensional case in practice. It is possible that these two cases work in very different regimes due to the differences between high-dimensional and low-dimensional probabilities.*
>
> We agree that the case of multivariate NFs is indeed what we ultimately want to solve. Please see our common response for why we believe the univariate result is interesting and what it says about the multivariate case.
>
>
> *The authors also made two important modifications to the unconstrained normalizing flow: changing the base distribution to standard exponential distribution and changing the quadrature to simple rectangle quadrature. These modifications can also make the model work in very different ways from practice, and the authors did not provide enough theoretical or experimental justifications for the modifications.
> The authors made modifications to the setting so that the optimization becomes convex, and this seems to be the only justification for these modifications.*
>
> In our paper we provide both theoretical and experimental justifications for our modifications and we restate them here: Both of these modifications are motivated by the need to make the optimization problem for the pseudo network convex. In our experiments, the use of the exponential base distributions does not cause any appreciable change. And the use of rectangle quadrature requires about twice as many points and about twice as much time as required for the Clenshaw--Curtis quadrature. It is also pertinent that our modifications are arguably natural.
>
> There's nothing canonical about using the Gaussian distribution as a base distribution. It does have some properties that are useful for a base distribution to have (easy sampling and log probability density estimation) but Gaussian is far from the only distribution to have them and the widespread use of Gaussian is probably just a historical accident. We remark that the modification to exponential base distribution is not an obvious modification even though it may appear so in retrospect and we consider it as a contribution of our work.
> Indeed, in connection with this rebuttal we checked if any previous work has considered base distributions other than Gaussian. We found an open problem in the survey https://arxiv.org/pdf/1908.09257 and the paper https://arxiv.org/pdf/1907.04481.pdf  mentioned there (contributions here are unrelated to ours) , and a recent arXiv posting (appearing after our work) showing that base distributions other than the Gaussian could be more effective for some purposes: https://arxiv.org/abs/2010.12059.
>
> On related note, normalizing flows architectures are evolving and there is no single standard architecture.
>
>
>
> *The techniques used in this paper mainly come from [1], and the proof framework and results are roughly the same.*
>
> As discussed in detail in our paper, our proof for 1D UNFs builds upon [1]. However, it's completely incorrect to say "the proof framework and results are roughly the same." The concrete challenges, both technical and conceptual, that our work needs to overcome are mentioned after the statement of Theorem 1. We would appreciate more concrete details if the reviewer is suggesting that our results can be proved directly using [1] without significant additional work and ideas. Even identifying UNF as a tractable version of NFs took substantial work on our part.
>
> We remind the reviewer of our other contributions as mentioned in the paper:
> * Identification of architectural variants of UNFs that admit analysis via overparametrization.
>
> * Identification of “barriers” to the analysis of CNFs.
>
>
> *The proof lies in the NTK/lazy training regime, which is hard to generalize to non-convex settings such as moderate width or large learning rate.*
>
> This is true. It must be noted, however, that this is also true for most (all?) of the unconditional theoretical results on deep learning.
>
> *The structure of this paper may need some improvements. The introduction section is a bit too long with perhaps too much background knowledge for normalizing flows. The contribution and related work parts can also be shortened. Section 2 is also a bit too long, and it may be better to re-organize this section and separate it into preliminaries/main results/proof sketch/discussions to make it easier for the readers to understand.*
>
> Thank you. We will revisit the structure of the paper.

---

> ### Author Response · Authors · 2020-11-25
> **Response (part 2/2)**
>
> *The experiments in this paper are a bit simple, i.e., the authors only did experiments on synthetic datasets like Gaussian mixture with models whose underlying neural networks have only one hidden layer. This setting is far from empirical settings, which makes the conclusions of the experiments not so convincing.*
>
> We are not sure if we fully understand this comment. Our experiments are for supporting our theory and not to provide new models in practical settings. As for the mixtures of Gaussian distribution, we note that it is well-known (we don't know of a standard reference but see e.g. https://ieeexplore.ieee.org/document/1100034, https://arxiv.org/abs/0805.3795) that mixtures of Gaussians are universal approximators, i.e. they can approximate any continuous probability density on compact sets if one allows sufficiently many Gaussians and the smoother the distribution the smaller the number of required Gaussians. We agree that more datasets could be tried, and indeed we have done that and have obtained similar results. However, there's no small set of datasets that would "cover" the space of all 1D probability distributions. Thus experiments can only provide evidence and not a proof that the training works in all cases.
>
>
> *The authors use "generalization" in the title, but I am a bit confused because I do not know what generalization means in this normalizing flow setting. Does this mean something like the KL divergence between the learned distribution and the target distribution?*
>
> Yes, this is what we mean. We want the learned distribution and the data distributions to be close. Our Theorem 1 shows this for KL-divergence, and as noted in the paper, this also implies that the two distributions are close in total variation distance.
>
>
> *In the last paragraph of page 5, the authors said that convex activations "cannot be used" because then $N(x)$ would also be convex. Does $N(x)$ have to be non-convex?*
>
> This is the CNF setting where we use $N(x)$ to fit $f(x)$. In general, all we know about $f(x)$ is that it is continuous strictly monotonically increasing and need not be convex. In such a situation, at the end of a successful training, $N(x)$ must be non-convex.

---

### Official Review · AnonReviewer4 · 2020-10-28
**Interesting model but the paper is confusing**

**Rating:** 4
**Confidence:** 4

**Review:**

**Summary**

The paper studies the problem of learning univariate normalizing flows with single-layer neural networks. The paper studies two models of normalizing flows: constrained normalizing flows (CNFs) and unconstrained normalizing flows(UNFs). For UNFs, the paper gives finite-sample results for UNFs in Theorem 1.

**Positives**

The paper studies two models of using neural networks for learning normalizing flows. For CNFs, paper identifies issues with the Taylor expansion in the parameter space. For UNFs, Theorem 1 shows that running SGD with a suitable learning rate leads to a neural network with small error. A theoretical study of normalizing flows looks like a promising research direction.


**Negatives**

The paper is very difficult to follow because of numerous grammatical issues and lax notations. In particular, parenthetical commas are incorrectly used throughout the paper.The paper should also be reorganized: Theorem 1, which is the main result, appears on Page 7. Please see the comments below for more details.

**Score**

I recommend rejection of this paper. The paper is not well-written and difficult to follow. The results on CNF are unsatisfactory (Section 2.1) and it is difficult to parse  the results in UNFs (Theorem 1).  The paper should go through major revisions for clarity. Please see the comments below for more details.

**Major comments**

1. I am confused by the term "constrained normalizing flows (CNFs)" for $a^2, w^2$ instead of $a$ and $w$. After this re-parameterization, the parameters are no longer constrained.
1. As the main result is Theorem 1, UNFs should be discussed earlier and CNFs should be discussed later.
3. Theorem 1 is too informal, and the statement of Theorem 2 should be explained better. The complexity measures $C_1, C_2,$ and  $C_3$ should be mentioned in theorem statements and discussed in the main text.
1. Should $\rho$ be a monotonically strictly increasing function or simply non-decreasing? (See the line after Eq. (4)) If so, why are ReLU networks considered throughout the paper. The first line in Section 2.1 should also be clarified.
2. The notation $L(f,x)$ is over-loaded in different sections: sometimes it is used with $f$ and sometimes with $f_t'$. This is extremely confusing.
4. ** Note that by the initialization, $|w_{r0}|$ and $|b_{r0}$ are O(\sqrt{\log m / m})**
What is the initialization distribution, and why can we not change the initialization distribution?
5. On page 5, the last line of the first paragraph: $L(N_t,x_t)$ is defined as the squared loss, but it was defined earlier in Eq. (2).
6.Please provide more details for experiments in Section 3. What were the base distribution, target distribution, and training set size? Since 1D distributions are easy to visualize, how does the estimated distribution compare with the target distribution?
7. In the top right image of Figure 1, I don't see any benefit of large $m$ --- the training curve is too unstable?


**Minor comments**


1. *Recent work in supervised learning attempts to provide theoretical justification for
why overparameterized neural networks can train and generalize efficiently in the above sense* Add a citation.
2. *We will only train the wr, br, and the ar0 will remain frozen to their initial value*
What about $w_{r0}$ and $b_{r0}$?
3. Some terms are defined but they are not used ever again, for example, $L_G$ for the Gaussian distribution.

---

> ### Author Response · Authors · 2020-11-25
> **Response (part 1/2)**
>
> Thank you for reading our paper and for your thoughtful comments.
>
> *The paper is very difficult to follow because of numerous grammatical issues and lax notations. In particular, parenthetical commas are incorrectly used throughout the paper. The paper should also be reorganized: Theorem 1, which is the main result, appears on Page 7. Please see the comments below for more details.*
>
> Thank you for detailed comments about writing. We will incorporate them in our revision where appropriate.
>
> *The results on CNF are unsatisfactory (Section 2.1)*
>
> We would appreciate more details here. In the paper, we have identified significant barriers for the analysis of CNFs, namely, one would need to analyze moderately-sized neural networks which remains an outstanding open problem.
>
> *I am confused by the term "constrained normalizing flows (CNFs)" for $a^2,w^2$ instead of $a$ and $w$. After this re-parameterization, the parameters are no longer constrained.*
>
> While we understand your concern, we think our terminology is reasonable and descriptive: as explained on page 2, the term Constrained NF refers to the general class of NFs where we model $f(x)$ as a neural network $N(x)$. Since f is strictly monotonically increasing, for this class of models, $N(x)$ is required to have the same property for all possible parameter values. This can be achieved either by re-parametrization or by explicit constraints. The main point here is that we constrain the parameters of a standard neural network in some way to achieve monotonicity (as opposed to UNFs). It is true that after re-parametrization the parameters are no longer constrained but in the original standard network they are.
>
> *As the main result is Theorem 1, UNFs should be discussed earlier and CNFs should be discussed later.*
>
> We think the current sequence of results is natural: We have three main contributions and Theorem 1 is one of them. While Theorem 1 takes up the bulk of technical work, our observations about CNFs (which are the dominant NF model) naturally lead up to consideration of UNFs and Theorem 1. To our knowledge, there's only one work on UNFs and all the other NFs fall under CNFs which lends extra significance to CNF results.
>
> *Theorem 1 is too informal, and the statement of Theorem 2 should be explained better. The complexity measures $C_1,C_2$ and $C_3$ should be mentioned in theorem statements and discussed in the main text.*
>
> We haven't included technical details in the main paper such as the complexity measures $C_1,C_2$ and $C_3$  in Theorem 1 because including technical details will  reduce readability of the main paper. Moreover more space would be needed than is available. For $p: R \to R$, complexity measures $C_1(p),C_2(p)$ and $C_3(p)$ all measure how fast the function $p$ varies. The faster the $p$ varies, the larger the measures.
>
>
> *Should $\rho$ be a monotonically strictly increasing function or simply non-decreasing? (See the line after Eq. (4)) If so, why are ReLU networks considered throughout the paper.*
>
> $\rho$ needs to be strictly increasing. For UNFs we are considering ReLU networks. For CNFs, we have explained in Sec. 2.1 that ReLU activation cannot be used.
>
> *The notation $L(f,x)$ is over-loaded in different sections*
>
> Theorem 1 statement had a typo: it should have had $L(f_t, x)$ instead of $L(f'_t, x)$.
>
> *```Note that by the initialization, $|w_{r0}|$ and $|b_{r0}|$ are $O(\sqrt{\log m / m})$. What is the initialization distribution, and why can we not change the initialization distribution?*
>
> This was an inadvertent omission. The initialization distribution for $w_{r0}$ is "half-normal" distribution with zero-mean and variance=$1/m$ of normal distribution, i.e. $w_{r0}=|X|$ where $X$ has normal distribution with the same parameters. The bias term $b_{r0}$ follows normal distribution with 0 mean and $1/m$ variance. This corresponds to the usual initialization.
> More generally, for the same initialization as above but variance $\sigma^2$ at most one, similar approximation issues arise. Details are in Appendix F of the revised version.
>
>
>
> *On page 5, the last line of the first paragraph: $L(N_t,x_t)$ is defined as the squared loss, but it was defined earlier in Eq. (2).*
>
> The first paragraph of page 5, which had $L(N_t, x_t)$ for square loss indeed had bad overloading. We have changed that notation to $L_S(N_t, x_t)$ to denote the square loss.
>
>
> *Please provide more details for experiments in Section 3. What were the base distribution, target distribution, and training set size? Since 1D distributions are easy to visualize, how does the estimated distribution compare with the target distribution?*
>
> Most of the details were mentioned in the Appendix E in the older version. For the UNF experiments, we used exponential base distribution and for CNF experiments, we used Gaussian base distribution.

---

> ### Author Response · Authors · 2020-11-25
> **Response (part 2/2)**
>
> *In the top right image of Figure 1, I don't see any benefit of large $m$ --- the training curve is too unstable?*
>
> This image supports our theoretical observation that overparametrization hurts training of CNF models.
>
> *We will only train the $w_r$, $b_r$, and the $a_{r0}$ will remain frozen to their initial value What about $w_{r0}$ and $b_{r0}$?*
>
> $w_{r0}$ and $b_{r0}$ are the initial random values of the weights (and thus are numerical values and not parameters) and $w_r$ and $b_r$ are the offsets to them and are parameters. Since $a$ is not trained, the offset $a_r$ remains 0 and the parameter remains frozen at $a_{r0}+a_r=a_{r0}$.

---

### Official Review · AnonReviewer1 · 2020-10-29

**Rating:** 6
**Confidence:** 3

**Review:**

The paper studies the role of overparameterization in learning normalizing flow models. More specifically, the authors analyze the optimization and generalization of such a model when the transport map f is parameterized by a two-layer neural network with potentially many hidden units (or highly over-parameterized). Importantly, the focus is on univariate data distributions.

First, the authors argue that overparameterization hurts the learning of constrained normalizing flows (CNFs) that impose positivity of weights though either projected gradient descent (PGD) or quadratic parameterization. Second, the authors prove that unconstrained NFs (UNFs) by modeling the gradient function f’ rather than f itself can learn the data distribution.

I definitely think this work makes some interesting contributions in terms of provable results for learning over-parameterized NFs. This is given by the fact that the problem is less well-understood compared to supervised learning. However, I am not sure about the impacts of the contribution to the general multivariate/high dimensional setting. Also, I have some other questions:

+) The first result on the failure of PGD/quadratic parameterization in the constrained case is interesting, theoretically. But I wonder if there is any artifact in the proof framework using pseudo networks or linear approximation.

+) Would you see the same observation in Figure 1 with more number of epochs and other activations, says ReLU. Please clarify “Gradient-based optimization algorithms are not applicable to problems with discontinuous objectives” around the end of page 5.

+) What are the difficulties of the Gaussian base distribution?

---

> ### Author Response · Authors · 2020-11-25
> **Response**
>
> Thank you for reading our paper and for your thoughtful comments.
>
> *However, I am not sure about the impacts of the contribution to the general multivariate/high dimensional setting.*
>
> We agree that the case of multivariate NFs is indeed what we ultimately want to solve. Please see our common response for why we believe the univariate result is interesting and what it says about the multivariate case.
>
> *+) The first result on the failure of PGD/quadratic parameterization in the constrained case is interesting, theoretically. But I wonder if there is any artifact in the proof framework using pseudo networks or linear approximation.*
>
> This is not an artifact of the proof technique: our results in Fig. 1 show that experiments bear out our theoretical observations about CNFs.
>
> *+) Would you see the same observation in Figure 1 with more number of epochs and other activations, says ReLU.*
>
> More epochs: We experimented with 1000 epochs and large change in the parameters is required in all cases. New experiments are included in the appendix.
> Other activations: Among the standard activations tanh and sigmoid are the only ones that we know to be admissible
> (i.e. non-convex functions with continuous derivative).  We have experimented with one more admissible activation that we designed. The activation is $(e^x - 1) I(x  \leq 0 ) + (1 - e^{-x}) I(x > 0)$. It gives results similar to tanh.
>
> *Please clarify “Gradient-based optimization algorithms are not applicable to problems with discontinuous objectives” around the end of page 5.*
>
> This statement means that if the objective function is discontinuous then gradient-based methods such as SGD are not applicable because the function may not have (sub-)gradient at some points and the function may fail to have small Lipschitz constant (the function value can change by a large amount for a small change in the argument value) around the points of discontinuity. Gradient-based methods can fail on functions with large Lipschitz constant. In particular, SGD on discontinuous functions may not converge to a stationary point. As remarked in the paper, this is not just a theoretical issue and we observed it in our experiment with CNF models with ReLU activation.
>
> *+) What are the difficulties of the Gaussian base distribution?*
>
> As pointed out in Remark 1 in Appendix C in the original submission, the main problem in using Gaussian base distribution is that it is not clear whether the loss of normalizing flow using pseudo network will remain convex throughout the training. Moreover, it is also not clear that Lipschitz constant of the loss function with the Gaussian base distribution will remain bounded by an absolute constant and hence independent of the complexity of the target function, $\epsilon$, etc. throughout the training. Lack of such boundedness property brings up additional challenges for the current analysis. (This last point is also affects the analysis of Allen-Zhu et al. where it was assumed that the loss function was 1-Lipschitz. But this assumption does not seem to be verified in their analysis.)

---

### Author Response · Authors · 2020-11-25
**Common Response**

[We have updated our submission with some minor notational and other improvements.]

We thank all the reviewers for their comments. One common concern was that the paper focuses on the univariate case which may be too limited compared to the high-dimensional settings which is usually what one is interested in practice. That the applications of NFs generally involve high-dimensional probability distributions is no doubt correct. Then why study the univariate case? Several reasons:

• For a hard problem, one often needs to first find simpler but illuminating cases that one can solve. Within deep learning theory, there have been multiple recent works on univariate neural networks even in the supervised settings, some of them cited in the paper.

• The univariate case is of special interest for normalizing flows even when one is concerned only with the high dimensional distributions. (This is not the case with other generative models like GANs.)
In the multidimensional case, the flow maps $(x_1, …, x_d)$ to $(z_1, …, z_d)$. More specifically, one has d neural nets $N_1, …, N_d$ with $N_i$ mapping $(x_1, …, x_i)$ to $z_i$. Neural net $N_1$ is a monotone univariate neural net mapping $x_1$ to $z_1$. For $i>1$, for any fixed $(x_1, …, x_i)$, the network $N_i$ is a univariate monotone network mapping $x_i$ to $z_i$.

• Our work on the univariate NFs identifies specific barriers for proving results for NFs modeling higher dimensional probability distributions. This was briefly mentioned at the end of the conclusion; we explain it in a little more detail below:

First, let us note that despite extensive work, even in the supervised setting the only case that's reasonably understood is when the neural network has one hidden layer and is very wide (also referred to as the NTK regime). Thus, to our knowledge, the following problems remain open (even in 1D):
(1) Analysis of training and generalization of moderately sized neural networks with one hidden layer (non-NTK regime).
(2) Analysis of neural networks with more than one hidden layer. This remains open for both moderately-wide as well as very wide networks.

An analysis of NFs for non-univariate (i.e., $d>1$) distributions will have to solve one of the above two problems:
	• For CNFs, as explained above, the neural net taking $x_1$ to $z_1$ is a univariate monotone neural net. We have shown in the paper that for univariate CNFs, there's strong evidence that one must work with moderately sized neural nets. Thus, one needs to solve problem (1) above to analyze CNFs.
	• To our knowledge, for UNFs there is no construction of the corresponding neural net with only one hidden layer if $d>1$. Thus, one needs to make progress on problem (2) above to analyze UNFs for $d>1$.

A different difficulty in the high-dimensional NTK regime for supervised learning with smooth activations like tanh and sigmoid is that the linear span of the dataset has small dimension then training can become extremely slow (https://arxiv.org/abs/1908.05660). This does not happen for activations like ReLU. As we have shown, the activations for neural net underlying CNFs must satisfy certain admissibility conditions and the only standard activations we know to be admissible are tanh and sigmoid. Thus one cannot expect a general theorem for such activations and one must work with a non-standard activation.

---

### Decision · Program_Chairs · 2021-01-07
**Final Decision**

**Decision:**

Reject

**Comment:**

Motivated by the fact that the benefit of overparameterization in unsupervised learning is not well understood than supervised learning, this paper analyzes normalized flow (NF) when the underlying neural network is one hidden layer overparameterized network and proves that for a certain class of NFs, one can efficiently learn any reasonable data distribution under minimal assumptions. The paper is very well motivated. However, the main concerns from the reviewers include (1) the writing quality and presentation are poor, even after revision during the author’s response; and (2) the analysis is limited in the neural tangent kernel (NTK) regime, which makes the results less significant. I agree with the reviewers’ evaluation and I think the first concern can be addressed by a careful revision, while the second concern needs additional nontrivial effort. Thus, I recommend rejection.